# Cross-modality mapping using image varifolds to align tissue-scale atlases to molecular-scale measures with application to 2D brain sections

Kaitlin M. Stouffer [1,2,3] ✉, Alain Trouvé [3], Laurent Younes [4], Michael Kunst[5], Lydia Ng[5], Hongkui Zeng [5], Manjari Anant[1], Jean Fan [1], Yongsoo Kim [6], Xiaoyin Chen [5], Mara Rue [5] & Michael I. Miller [1,2] ✉

This paper explicates a solution to building correspondences between molecular-scale transcriptomics and tissue-scale atlases. This problem arises in atlas construction and cross-specimen/technology alignment where specimens per emerging technology remain sparse and conventional image representations cannot efficiently model the high dimensions from subcellular detection of thousands of genes. We address these challenges by representing spatial transcriptomics data as generalized functions encoding position and high-dimensional feature (gene, cell type) identity. We map onto low-dimensional atlas ontologies by modeling regions as homogeneous random fields with unknown transcriptomic feature distribution. We solve simultaneously for the minimizing geodesic diffeomorphism of coordinates through LDDMM and for these latent feature densities. We map tissue-scale mouse brain atlases to gene-based and cell-based transcriptomics data from MERFISH and BARseq technologies and to histopathology and cross-species atlases to illustrate integration of diverse molecular and cellular datasets into a single coordinate system as a means of comparison and further atlas construction.

Since the 17th century, scientists have seen living organisms as a hierarchy of biological mechanisms at work across scales. To understand the interplay of these mechanisms, reference atlases that incorporate genetic, cellular, and connectivity measures into a single coordinate space have been constructed and which aim to summarize the mass of data across scales through a set of discrete partitions. An instance of the more general segmentation problem in computer vision, atlas construction relies on the underlying assumption of homogeneity within each region. The optimal partitioning assigns a label to each region based on this homogeneity and the presence of sharp changes at the boundaries between regions. In biology, this label frequently reflects behavior or function, as seen in two of the most common mouse brain atlases: the Allen Common Coordinate Framework (CCFv3)[1] and the Franklin and Paxinos Atlas[2]. Together, the common coordinate framework an atlas provides in addition to its ontology facilitates the comparison of different types or replicates of data in a single coordinate system and hones efforts of study to particular regions relevant to each unique investigation.

Hence, atlas construction and segmentation of new data with defined atlases comprise two classes of widespread problems across domains of biology. These are particularly relevant in the context of emerging datasets in spatial transcriptomics, where atlases have still

[1]Department of Biomedical Engineering, Johns Hopkins University, Baltimore, MD, USA. [2]Kavli Neuroscience Discovery Institute, Johns Hopkins University, Baltimore, MD, USA. [3]Centre Borelli, ENS Paris-Saclay, Gif-sur-yvette, France. [4]Department of Applied Mathematics and Statistics, Johns Hopkins University, Baltimore, MD, USA. [5]Allen Institute for Brain Science, Seattle, WA, USA. [6]Department of Neural and Behavioral Sciences, Penn State University, College of Medicine, State College, PA, USA. ✉e-mail: kstouff4@jhmi.edu; mim@jhu.edu

yet to be constructed[3] and where the different technologies used number quite high, thus emphasizing the need for mapping data across samples and technologies into a common coordinate framework for comparison. A central challenge in both atlas construction and segmentation, however, is in aligning in a single coordinate space data that often exists at different scales (e.g. coarse scale atlases to fine-scale datasets) and is fundamentally of different functional modalities (e.g. tissue regions to molecular images). These differences preclude an obvious definition of similarity for consequently optimizing alignment.

In the classical image setting, a large family of diffeomorphism-based methods[4] has been developed within the field of Computational Anatomy (CA)[5,6] for transforming coordinate systems at the tissue scales. The smoothness, invertibility, and non-rigid nature of diffeomorphisms are particularly relevant in medical image alignment as they reflect the mechanics/dynamics of soft tissue[7]. Consequently, methods rooted in diffeomorphic registration come particularly from multiple labs in the magnetic resonance imaging (MRI) community, with variations developed in the contexts of both atlas construction and segmentation at these tissue scales[7–16]. Large Deformation Diffeomorphic Metric Mapping (LDDMM)[17], on which many such methods are built, specifically equips diffeomorphisms with a metric that allows them to be reduced to a low-dimensional representation that conveniently identifies shape. Consequently, it has been harnessed in the setting of image classification and the study of atrophy in diseased versus control populations[17–19]. While successful at aligning images of the same modality, most of these diffeomorphism-based methods focus exclusively on generating geometric transformations (e.g. diffeomorphisms). The challenge of crossing functional modalities has instead been addressed in a classical imaging setting by coupling such methods to variations in matching cost or additional transformations. These include matching based on analytical methods using cross-correlation[13] or localized texture features[20], and methods for transforming one range space to another in crossing modalities and scales based on polynomial transformations[21], scattering transforms[22], graphical models[23], and neural networks[24–26].

In the molecular setting, however, both the diversity and magnitude of data measured by spatial transcriptomics technologies[27] often prohibit the representation of such data as classical continuous images discretized as regular grids, and consequently, the direct use of these image alignment methods at the molecular scales. As seen in repositories generated in the BRAIN Initiative Cell Census Network (BICCN) and archived at the Brain Image Library (BIL), these datasets are already on the order of terabytes and will only continue to increase as technologies shift from mouse to human measurements. Hence, while some methods in deep learning have been applied to align single-cell datasets, modeled as regular grid images, both to atlases at the histological scale[28] as well as reference transcriptional atlases that are beginning to emerge[29], most image-based tactics are limited in their ability to represent this high-dimensional and memory-intensive data. Furthermore, many learning-based schemes in the context of classical images have relied on the use of extensive numbers (typically on the order of hundreds) of images from different subjects for extracting features useful for cross-modalities and mapping[18,26]. In the setting of spatial transcriptomics technologies that have been developed only over the last few years, the acquisition of such large training datasets is typically prohibitive for some of these learning-based approaches.

Consequently, an independent class of methods has been developing for aligning spatially resolved transcriptional datasets at these molecular scales. With influence from image-based methods, some of these including GPSA[3] and PASTE[30] focus exclusively on spot-based data in which gene expression is measured in a neighborhood of each spot for a regular array of spots, analogous to a voxel grid in an image. In contrast, image-based technologies such as STARmap, BARseq, and MERFISH, which take point measures of individual mRNA molecules or cells, cover measurements irregularly sampled over space according to tissue architecture and dynamics. Furthermore, natural fluctuation in gene expression over time and space coupled to the dynamics of each spatial transcriptomics technology leads each tissue section, at the molecular/cellular (0.3–100 micron) scales, to have a varying number of such particles with no natural ordering of particles consistently apparent between sections. Consequently, landmark-based methods[31] that assume direct permutation correspondence between particles are not applicable. Methods aimed at generalizing to allow alignment within and across these additional technologies are typically rooted in different data representations, such as graphs, in the case of SLAT[32] that aims to find correspondence between cells or groups of cells in an atlas and target or CAST[33], a deep graph-based neural network (GNN) algorithm which learns a graph representation of single-cell resolution omics data and subsequently aims to align datasets via these graph representations. Additionally, while many methods have showcased success at aligning different replicates within a single technology for both human and mouse samples, cross-technology alignment has typically relied on a nonempty intersection of feature sets across these technologies[28,32,33], limiting extension of these methods to the type of cross-modality and cross-scale mapping required for integrating tissue-scale atlases and molecular scale data.

Here, we build on both of these lineages in presenting a model equally equipped at representing tissue-scale atlases and molecular-scale data, and an associated cross-modality registration mechanism that addresses both challenges of crossing scales and functional modalities by estimating simultaneously geometric and functional transformations to align each dataset to the other. Specifically, we harness the generalizability of image varifolds, which have emerged in molecular CA[34], for simultaneously modeling molecular and tissue-scale data with both irregular and gridlike sampling schemes in a common framework (addressed in SLAT and CAST, for instance, by a graph-based representation in the setting of molecular and cellular data[32,33]). A subproblem covered by the image-varifold theory outlined in ref. 36 is the mapping of molecular scale data to atlas coordinate systems. As specified there, we estimate minimal energy diffeomorphic transformations as in image-based LDDMM[17], but where the action of diffeomorphisms on images has been adapted to the setting, here, of image varifolds in a consistent manner[34]. Additionally, we estimate in tandem a latent distribution over the molecular functional space for each atlas partition, without the need for the large datasets often used in deep learning approaches, but by instead relying on an assumption of spatial homogeneity in distribution across each atlas partition. We use this latent distribution to transform the functional space of each atlas (i.e., its ontology) to the molecular functional space. We refer to this method of jointly estimating diffeomorphisms and latent feature distributions, as Cross Image-varifold LDDMM (xIV-LDDMM), where the cross emphasizes its ability to map across scales and modalities[35]. We demonstrate this methodology with two common implementations of image varifolds (triangulated meshes and point clouds). First, we introduce the methodology through a demonstration of mapping both gene-based and cell-based MERFISH datasets to corresponding sections of the Allen Common Coordinate Framework (CCFv3)[1]. Second, we quantify accuracy in mapping specifically characterized cell types in BARseq data[36] to the CCFv3. Finally, we illustrate the method's generalizability to additional modalities and offer diverse examples of its use across the spans of biology in atlas construction, cross-replicate and cross-species comparison, and data segmentation.

## Results

### Data model and optimization problem: image varifolds and transformations for molecular scales based on varifold norms

To accommodate the high feature dimensionality and spatial irregularity of molecular datasets, as described in the Introduction, we harness the recent work that extends the theory of diffeomorphisms and

image-based LDDMM[17] to the setting of image varifolds and estimation of correspondences between them[34,35]. We first describe this framework of image varifolds to emphasize their capacity for modeling diverse types of tissue-, cellular-, and molecular-scale data. Image varifolds are geometric measures over the product space, $\mathbb{R}^2 \times \mathcal{F}$, therefore encompassing measures over both physical and feature spaces, $\mathbb{R}^2$ and $\mathcal{F}$, respectively. At the finest scale of capture in both classical imaging and image-based spatial transcriptomics technologies, we might model a set of pixels or detections (e.g. mRNA reads) as a discrete set of point measures (particles) with an elementary Dirac measure, centered over physical location and feature value for each detection. At coarser (e.g. cellular) scales or in spot-based technologies, however, point measurements taken at single physical locations may capture a range of individual detections. To generalize to the range of technologies, we model both tissue-scale atlases and molecular data with semi-discrete image varifolds where the physical arrangement of measurements is captured by a collection of discrete point (Dirac) measures, but where each such point measure is associated with a full distribution over feature values measured in its neighborhood. This collection of point measures is indexed by $i \in I$, with discrete measures, $\delta_{x_i}$, evaluating to 1 at locations $x_i \in \mathbb{R}^2$, and distributions over feature values modeled as weighted probability distributions: $w_i p_i$. Weight, $w_i$, is representative of total mass (e.g. total mRNA reads or total cells) measured at location $x_i$, thus enabling an estimate of density over physical space, and probability distribution, $p_i$, captures the proportions of each feature type within that mass.

The measure of the complete collection is denoted by the sum:

$$\mu \doteq \sum_{i \in I} w_i \delta_{x_i} \otimes p_i. \tag{1}$$

Feature spaces considered here are finite, yielding probability distributions over 10s–100s of different elements depending on the application:

- MERFISH gene sections: $w_i$ is the total mRNA at location $x_i$, and $p_i$ is the probability distribution on gene (~700).
- BARseq cell sections: $w_i$ is the total cells at location $x_i$ and $p_i$ is the probability distribution on cell type (~30).
- CCFv3 tissue sections: $w_i = 1$ for location $x_i$ in foreground tissue and $p_i$ is the probability distribution on ontology label (~700).

Within this established framework, we align image varifolds capturing tissue-scale atlases to those capturing molecular-scale data via the estimation of two types of correspondence: one between physical coordinates ($\varphi$) and the other between feature spaces ($\pi$). These correspondences act independently and in parallel on the physical and feature measure components of the image-varifold object. Consequently, while they are both applied in the setting of optimization to evaluate the alignment of the atlas to target (top panel, Fig. 1) they can be applied individually as relevant to specific applications including data segmentation, atlas construction, and cross-specimen comparison, as depicted in the middle panel in Fig. 1.

Regarding physical correspondence, in Computational Anatomy (CA), correspondence between tissue sections is computed using coordinate transformation between the sections by solving an optimization problem characterized by the set of possible transformations to optimize the image similarity function that specifies the alignment of the sections. These transformations are modeled as affine motions and diffeomorphisms, $\varphi$, which act to generate the space of all configurations. We model physical transformations in the setting of image varifolds similarly to diffeomorphisms, optimally solving for them with LDDMM[17]. However, while for classical images such as for MRI, LDDMM[17] uses the action of diffeomorphisms on images $I$ as classical

functions using function composition on the right with the inverse of the diffeomorphism: $\varphi \cdot I(x) = I \circ \varphi^{-1}(x)$ for $x \in R^d$, the action of diffeomorphisms on image varifolds is defined, as in[34,35]:

$$\varphi \cdot \mu \doteq \sum_{i \in I} |D\varphi|_{x_i} w_i \delta_{\varphi(x_i)} \otimes p_i , \tag{2}$$

for the form of the image varifold defined in (1). The determinant of the Jacobian, $|D\varphi|$, capturing the local expansion/contraction of physical space, is introduced to retain the given spatial density of the original object following physical transformation. Notably, the estimated diffeomorphism can be applied in both the forward and inverse directions, taking tissue-scale atlas to molecular coordinates or vice versa, as shown in the middle panel in Fig. 1. The bottom panel depicts the determinant of the Jacobian of the diffeomorphism taking CCFv3 section to MERFISH section (B,G) with areas of local expansion in red and local contraction in blue.

To define a similarity metric between the deformed atlas and the target, we also need to carry the feature component of tissue-scale atlases to the feature space of the molecular target. For this, we associate to each feature value in the atlas ontology ($\ell \in \mathcal{L}$), a distribution (denoted $\pi_\ell$) over target feature values, capturing the assumption of spatial homogeneity we make within each atlas region. Importantly, these distributions are not normalized, enabling the generation of both a measure of target mass (e.g. mRNA or cells) over physical space (given by $w'_i$) and conditional probability distribution ($p'_i$) over target features (e.g. gene or cell type). Both molecular mass and conditional probability distribution are associated with each point measured in the atlas through the mixture distribution:

$$w'_i p'_i = w_i \sum_{\ell \in \mathcal{L}} p_i(\ell) \pi_\ell , \tag{3}$$

with $w_i, p_i$ denoting the initial weight and probability distribution of the point measure over the atlas feature space and with

$$w'_i p'_i(f) = \begin{cases} \text{mass of cell type } f \text{ at location } x_i \text{ for } f \in \mathcal{F} = \{\text{cell types}\} \\ \text{mass of gene } f \text{ at location } x_i \text{ for } f \in \mathcal{F} = \{\text{genes}\} . \end{cases}$$

The bottom panel in Fig. 1c, h exhibits the estimated mRNA density ($w'_i$) for each location in the corresponding CCFv3 section while the middle column of the middle panel summarizes the estimated probability distribution over genes ($p'_i$) with the gene with the highest probability denoted for each location.

Finally, the estimation and application of both geometric transformations and latent feature distributions, $\varphi, \pi$, to the atlas image varifold take it to both the physical and feature space of the target as necessary to evaluate the similarity function in the optimization scheme (top panel, Fig. 1). In the image setting, the similarity function used is often a norm on functions, and solving the problem of minimization of the norm in the space of diffeomorphisms gives the metric theory of LDDMM for generating geodesic matching between exemplar anatomies[37,38]. Here, the similarity function is a norm on image varifolds (see "Molecular scale varifold norm" section), capturing proximity in both physical and feature spaces of the target. xIV-LDDMM jointly estimates optimal $\varphi$ and $\pi$ (top panel, Fig. 1) to minimize this normed difference in the space of diffeomorphisms through either simultaneous or alternating optimization algorithms using LDDMM (see "Alternating LDDMM and quadratic program algorithm for joint optimization" section), with additional regularization imposed in both settings on the estimated $\pi$ to ensure, for instance, positive values (see "Variational problems" section for explicit variational problems). Note that throughout we highlight mapping examples using

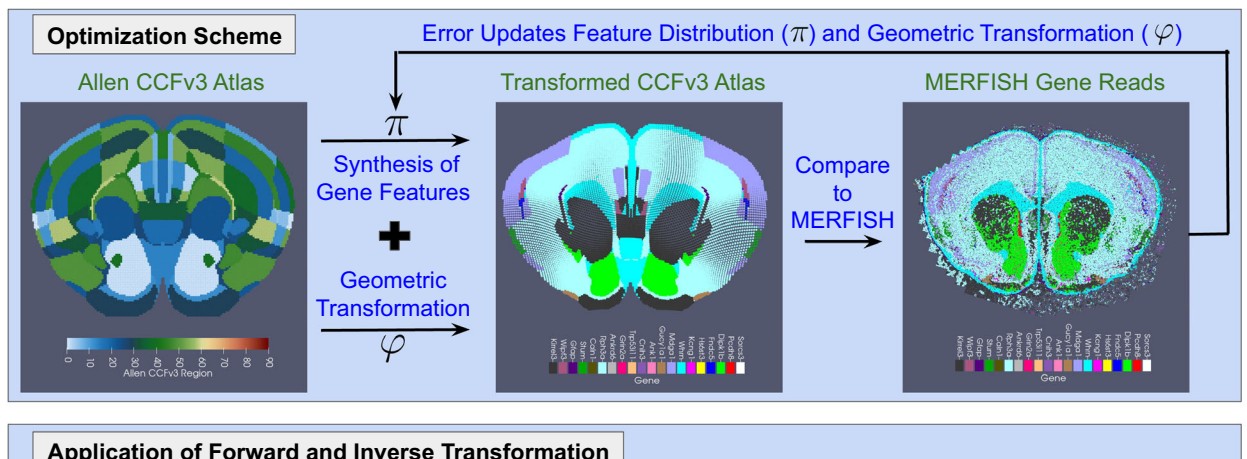

both mesh-based[35] and particle-based (point cloud)[34] implementations of image varifolds, with detailed construction, similarities, and differences of each covered in "Construction of image-varifold representation for different modalities" section. Supplementary Fig. 1 shows equivalence in these two implementations and in the corresponding mappings estimated in each case for the CCFv3 and MERFISH sections shown in Fig. 1.

## Generalizability of xIV-LDDMM to settings of complete and incomplete gene-based and cell-based data

Here, we emphasize the generalizability of xIV-LDDMM for computing mappings from tissue-scale atlases to datasets spanning a range of both gene-based and cell-based features. The bottom panel of Fig. 1 specifically summarizes the estimated latent distribution over genes for each atlas region with the sum over gene counts, exhibiting the

**Fig. 1 | Methodology and results of xIV-LDDMM for transforming CCFv3 sections to MERFISH spatial transcriptomic counts of 20 selected genes.** Top panel shows an iterative optimization scheme to estimate geometric transformation ($\varphi$) and latent feature distribution ($\pi$) by minimizing the normed difference (error) between the geometric and feature-transformed CCFv3 section to target MERFISH. Middle panel illustrates the application of estimated geometric transformation ($\varphi$) to deform the CCFv3 atlas to MERFISH coordinates (left); the application of latent feature distribution ($\pi$) to generate gene distributions on initial CCFv3 geometry (middle); and the application of inverse geometric transformation ($\varphi^{-1}$) to deform MERFISH genes to CCFv3 coordinates (right). Gene with the highest probability of expression at each location is shown as a MERFISH

feature. Bottom panel illustrates the results of mapping CCFv3 sections to corresponding MERFISH sections. **a, f** 10 μm atlas sections at $Z = 385$ and $Z = 485$ out of 1320 visually chosen to match MERFISH architecture (**e, j**) rendered as meshes at 100 μm. **e, j** MERFISH sections rendered as meshes at 50 μm, with mRNA density depicted as a feature. **b, g** Geometric mappings ($\varphi$) of CCFv3 sections to MERFISH coordinates with the approximate determinant of the Jacobian showing areas of contraction (blue) and expansion (red). **c, h** Estimated mRNA density per atlas region ($w_i'$ in (3)), as given by $\pi$ shown in CCFv3 coordinates. **d, i** Estimated mRNA density shown per atlas region following geometric deformation to MERFISH coordinates.

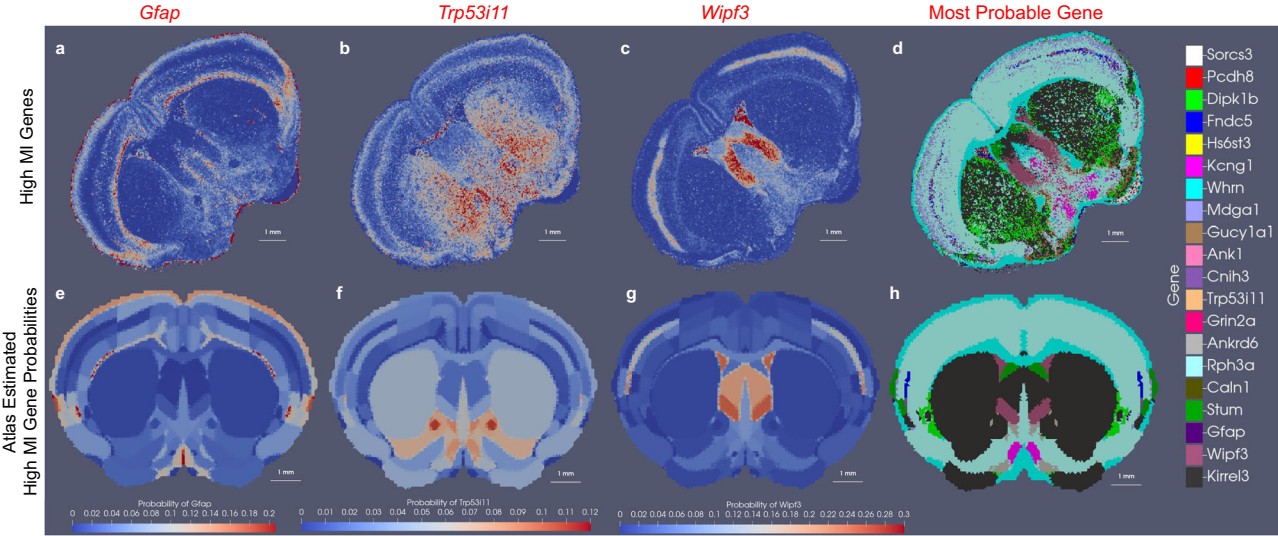

**Fig. 2 | Measured and predicted spatial patterns of gene expression for a subset of 20 genes measured by MERFISH. a–c** Relative expression on target section of MERFISH transcriptomics data for three genes (*Gfap* (**a**), *Trp53i11* (**b**), *Wipf3* (**c**)) out of a set of twenty (shown at the right), with demonstrated spatial variability according to a computed mutual information score. **e–g** Predicted expression for the same three genes (*Gfap* (**e**), *Trp53i11* (**f**), *Wipf3* (**g**)) in each region of the

CCFv3 section $Z = 485$ out of 1320, as part of the latent distribution over genes estimated in tandem with a geometric transformation to align the CCFv3 section to MERFISH section. **d** Gene with the highest probability in MERFISH target section. **h** Predicted gene with the highest probability in estimated latent distribution for the CCFv3 section.

estimated density of mRNA over physical space. However, as described in "Data model and optimization problem: image varifolds and transformations for molecular scales based on varifold norms" section, a full weighted probability distribution over feature values (e.g. gene type) is estimated via $\pi$ for each atlas location that not only contributes to achieving geometric alignment but can be visualized independently as reflective of mean distributions in target data. Figure 2a–c, for instance, shows the relative expression of three genes, (*Gfap, Trp53i11, Wipf3*, respectively) over space out of a total set of 20 in the target MERFISH section shown in Fig. 1. Figure 2e–g show the corresponding expression of each of these genes in each region of the CCFv3 section, as estimated through $\pi$, with notable similarity in probability magnitude and variation over space. Figure 2d, h showcase another comparative summary of the estimated feature distributions for the CCFv3 section versus MERFISH expression in depicting at each location the most probable gene type. The assumption of spatial homogeneity in distribution is evidenced here, particularly in large areas of the CCFv3, such as the striatum, where each gene carries a single probability of expression across the entire region and *Kirrel3* is uniformly the most probable gene type.

The previous result solved the mapping problem between CCFv3 and MERFISH sections based on the mRNA reads directly. Often, these raw mRNA reads are segmented into discrete cells as a mode of data reduction followed by downstream analyses clustering the cells into discrete cell types. We emphasize that there are various methods for solving the segmentation to cells and thereby dimension reduction as

determined by the specific imaging technology. Some of the methods are rooted in image-based segmentation schemes such as the Watershed algorithm, operating jointly on transcriptional data and immunofluorescence images such as DAPI stains[39], while others utilize learning-based methods[40] for accommodating often a wider diversity of cell shapes and sizes. In either case, the assignment of mRNA reads to specific cells introduces a layer of functional information at the micron scale, which can now be modeled in lieu of or in tandem with the functional information at the nanometer scale (e.g. raw mRNA reads) as the feature space of a target image varifold to which we wish to map sections of an atlas.

We demonstrate the efficacy of xIV-LDDMM for mapping tissue-scale atlases to cellular-scale data in mapping CCFv3 section $Z = 675$ to a section of cell-segmented MERFISH transcriptional data (courtesy of the JEFworks Lab, Johns Hopkins University) (Fig. 3). The total gene set measured is ≈500 genes, with each transcript assigned to a single cell. Transcriptional profiles per cell are clustered into 33 distinct clusters using Leiden graph-based clustering[41] and annotated as cell types based on known marker genes. This gives a cell-based dataset analogous to the transcript-based dataset used in the above where we now capture the spatial density of *cells*, and conditional probability distributions over cell types, which can be summarized via the depiction of the cell type with the highest probability at each location in space (Fig. 3a). The latent cell type distribution per each CCFv3 region, estimated in tandem with the geometric transformation taking CCFv3 section to MERFISH spatial coordinates exhibits similarity to the

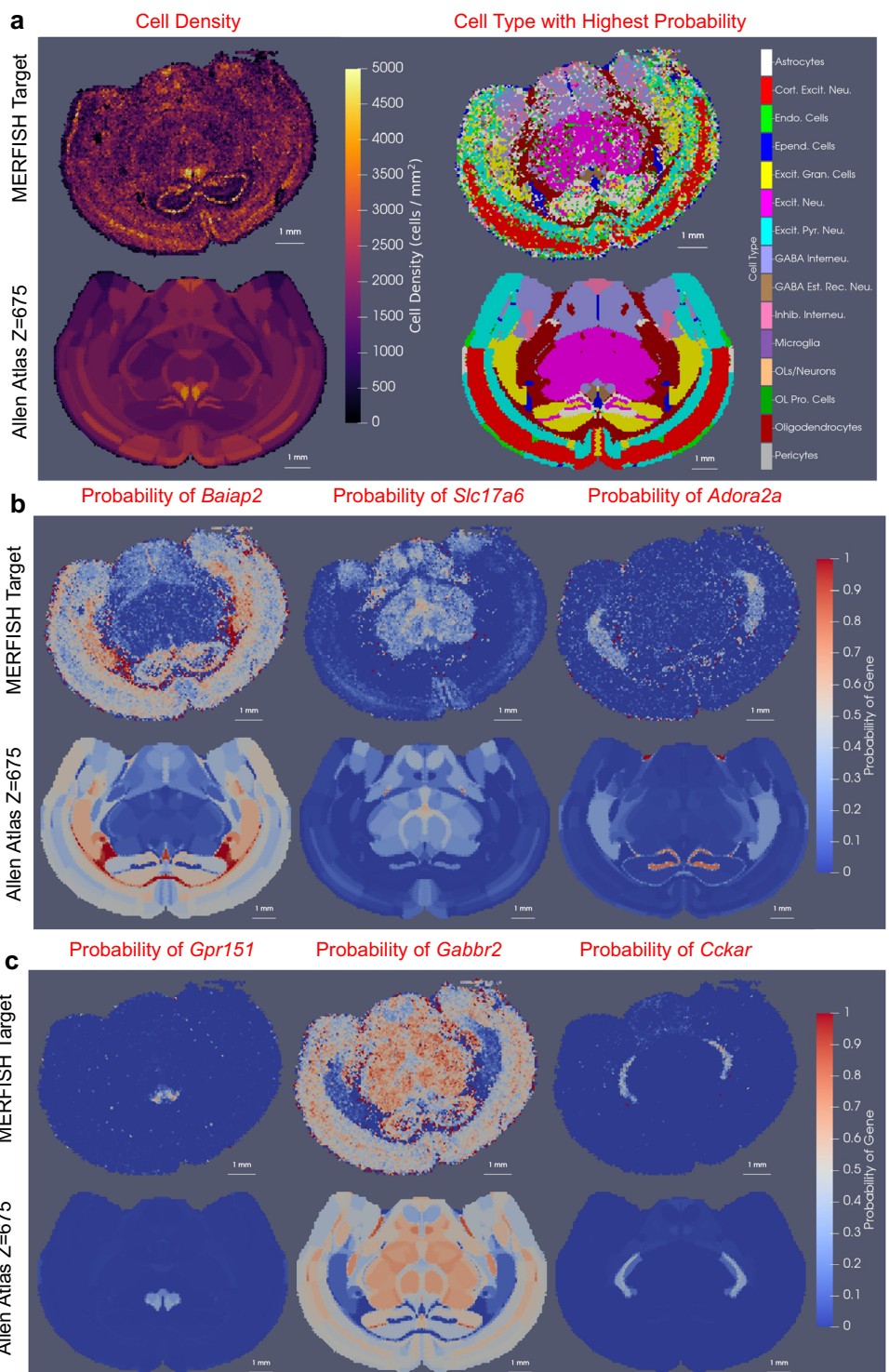

**Fig. 3 | Mapping of CCFv3 section *Z* = 675 out of 1320 to single MERFISH cell-based section with either cell type or gene feature spaces. a** Top row depicts MERFISH target rendered as a mesh with cell density (left) and cell type with the highest probability (right) used to summarize cell type distributions. Bottom shows predicted cell density (left) and cell type with the highest probability (right) for the latent feature distribution estimated for each CCFv3 region in the native CCFv3 coordinates. **b, c** Top row depicts the probability of expression for each gene out of a subset of 6 selected from a total measured set of ~500 as those with high spatial variance. Bottom row depicts the estimated probability of expression for each gene for the latent feature distribution estimated for each CCFv3 region in the native CCFv3 coordinates.

MERFISH target in both total cell density and cell type with highest probability predicted per CCFv3 region (Fig. 3a). Alternatively, each MERFISH section can be characterized with a feature space over genes, and a corresponding latent distribution over genes can be estimated via xIV-LDDMM in mapping the same CCFv3 section to MERFISH gene target (Fig. 3b, c). Here, we subsample the entire gene space by selecting a subset of six genes (*Baiap2, Slc17a6, Adora2a, Gpr151, Gabbr2, Cckar*) as those most spatially varying according to Moran's I score[42]. We combine both cell density and gene expression by aggregating the individual mRNA transcripts into an average gene

expression feature per cell and normalizing the total mRNA per cell to 1. The target and estimated probabilities for each gene out of this subset of six are shown following the estimation of an alternative geometric transformation and latent distribution, now, over genes rather than cell types. Correspondence is seen in both the absolute magnitude as well as the relative probability of each gene between those in the target MERFISH section and those resulting from the estimated feature laws, $\pi$. Two examples include the area of high *Baiap2* expression adjacent to the hippocampus (probability $\approx 0.95$) and the area of high *Slc17a6* expression in the rhomboid nucleus (probability $\approx 0.6$), both depicted in Fig. 3b. This correspondence serves to reinforce the validity in the estimated feature laws in xIV-LDDMM.

Notably, the assumed biological correlation between cell type and pattern of gene expression implies that signals of variation across cell types at the scale of microns should also exist across gene types at the scale of nanometers. Consequently, we might expect similar spatial deformation of a tissue-scale atlas in mapping onto the same geometric target, but with conditional feature distributions defined over either gene or cell types, with partition boundaries deforming to match regions of homogeneity that would be roughly consistent across genes and cells. Supplementary Fig. 2 shows the diffeomorphism estimated in the two cases (6 genes and cell type) in Fig. 3 together with that for a third feature space comprised of a second set of 7 different genes selected for high spatial variance using mutual information. We observe the global similarity in spatial pattern and magnitude of contraction and expansion occurring but with nuanced differences in areas where these feature spaces likely emphasize different boundaries. Hence, the manifest stability in the geometric mappings jointly estimated with the feature laws over three different feature spaces supports the stability of our method in the face of different numbers and types of features but also speaks to the stability of the biological organization across tissue, cellular, and molecular scales.

Finally, we emphasize the generalizability of our mapping methodology across particularly image-based spatial transcriptomics technologies by showcasing the alignment of a cell-segmented BARseq[36,43] partial coronal section to a corresponding full coronal CCFv3 section (Supplementary Fig. 3). Note that cells were typed according to the same procedure used for the full coronal section highlighted in "Quantitative and comparative evaluation of xIV-LDDMM" section. The costs involved with many of these emerging technologies coupled with their use in studying particular subregions or brain circuits of interest has caused the partial measurement of tissue areas to be incredibly prevalent across both image-based and spot-based technologies[44]. Consequently, this generates even greater variety in the scope and shape of tissue sections measured, only further emphasizing the need for aligning such partial captures to a common scaffold, such as an atlas coordinate system, where information can be merged across the intersection of these captures. The image-varifold representation, particularly in its semi-discrete form as used here, is amenable to representing data with regions of missing or disrupted capture as an extension of the irregularity in sampling that is assumed in the general case. To estimate mappings in these cases, variation in both the forms and sizes of kernels governing the varifold norm (see "Molecular scale varifold norm" section) can control the granularity to which matches between atlas and target should be evaluated, with coarser scales more appropriate to noisier tissue sections with higher numbers of artifacts. Additionally, the varifold normed difference in the cost function, as presented in (10) in "Variational problems" section can be appended with spatially varying weights, as used in settings of mapping digital pathology to MRI[22] to prioritize matching amongst certain intact regions over others. We utilize this strategy to align a partial BARseq section to a corresponding full CCFv3 section where initial geometric offset (Supplementary Fig. 3A) requires accurate estimation of scale,

rigid, and diffeomorphic transformations to position the partial capture within the scope of the correct CCFv3 hemisphere (Supplementary Fig. 3B).

## Quantitative and comparative evaluation of xIV-LDDMM

We specifically evaluated the efficacy of xIV-LDDMM in mapping CCFv3 sections to corresponding coronal sections of BARseq cell-segmented and subsequently cell-typed data. As described in ref. 37, cells were segmented in the BARseq data using Cellpose[45]. Gene reads were assigned to cells, and cells containing fewer than 5 unique genes and 20 total gene counts were excluded. Cells were clustered using an iterative clustering approach based on Louvain clustering to achieve a similar resolution as a subclass in recent single-cell RNAseq studies[46].

Given the difference in atlas and target modalities mapped with xIV-LDDMM, traditional measures of accuracy, such as Dice overlap score, as are often used to evaluate image registration tools cannot be directly applied. Furthermore, landmarks common to both atlas and target modalities are not typically readily available or necessarily identifiable given the diversity in measures taken by molecular modalities[3]. Consequently, to evaluate the accuracy, particularly in the geometric alignment achieved with xIV-LDDMM, we matched corresponding atlas feature values (regions) with target feature values (e.g. cell types) and quantified the resulting distance individual cells of a given type from the atlas region in which we expect to find them.

We quantified accuracy by computing the set distance between cells of types specific to hippocampal regions (e.g. CA1, CA2, and CA3 pyramidal cells and DG granule cells) and particles of these regions in the CCFv3. We specifically defined this distance for each cell, indexed by $c \in \mathcal{C}_R$, for region $R$, as:

$$d_c = \min_{a \in \mathcal{A}_R} \| x_a - y_c \|_2^2, \tag{4}$$

where location $y_c$ was the given 2D coordinate of the cell in the native BARseq coordinate system, and the set $\mathcal{A}_R$ refers to the set of indexed locations in the atlas section with feature value (region label) of $R$ (e.g. with $p_a(R) = 1$, and $p_a(f) = 0$ for $f \neq R$). We compared these distances to those after deformation, by replacing $y_c$ with the mapped position of each cell marker to the CCFv3 section: $\varphi^{-1}(y_c)$, via the inverse diffeomorphism estimated in our joint image-varifold-based method. Figure 4 depicts these measures for two separate CCFv3 sections mapped to two corresponding sections of BARseq cell data. Initial positions (Fig. 4a, c) of CCFv3 regions versus cells give distances (Fig. 4e, g) on the order of 1 mm, where notably, neighboring particles in the CCFv3 are at 10 $\mu m$, thus giving a lower bound to distance metrics we might expect for cells to be neighboring CCFv3 particles. In contrast, the positioning of CCFv3 regions versus cells following geometric transformation reflects the effects of rotations, translations, scale, and diffeomorphisms effectively estimated to bring them into correspondence (Fig. 4b, d). Kernel size governing diffeomorphism regularization and varifold norm matching were both on the order of 100 μm. Median distances for all three regions in the first section are on the order of 20−30 μm (Fig. 4f) whereas those in the second section are on the order of 5 μm for CA1 and DG and 300 μm for CA3 (Fig. 4h), which we assume is coming from the separate group of cells labeled as CA3 at the edge of CA1 (indicated by the white star) in contrast to the group overlaying the CCFv3 region, which falls within the first 10 μm bin. Note that this discrepancy likely stems in part from the approximation of each BARseq section as a strictly coronal section of the CCFv3, where the cutting plane is instead slightly offset from this coronal plane. In any case, we observe in both sections an accuracy on the order of 10−30 μm, analogous in the image setting to cells being mapped to within 1−3 pixels (at a resolution of 10 μm) to the appropriate atlas region.

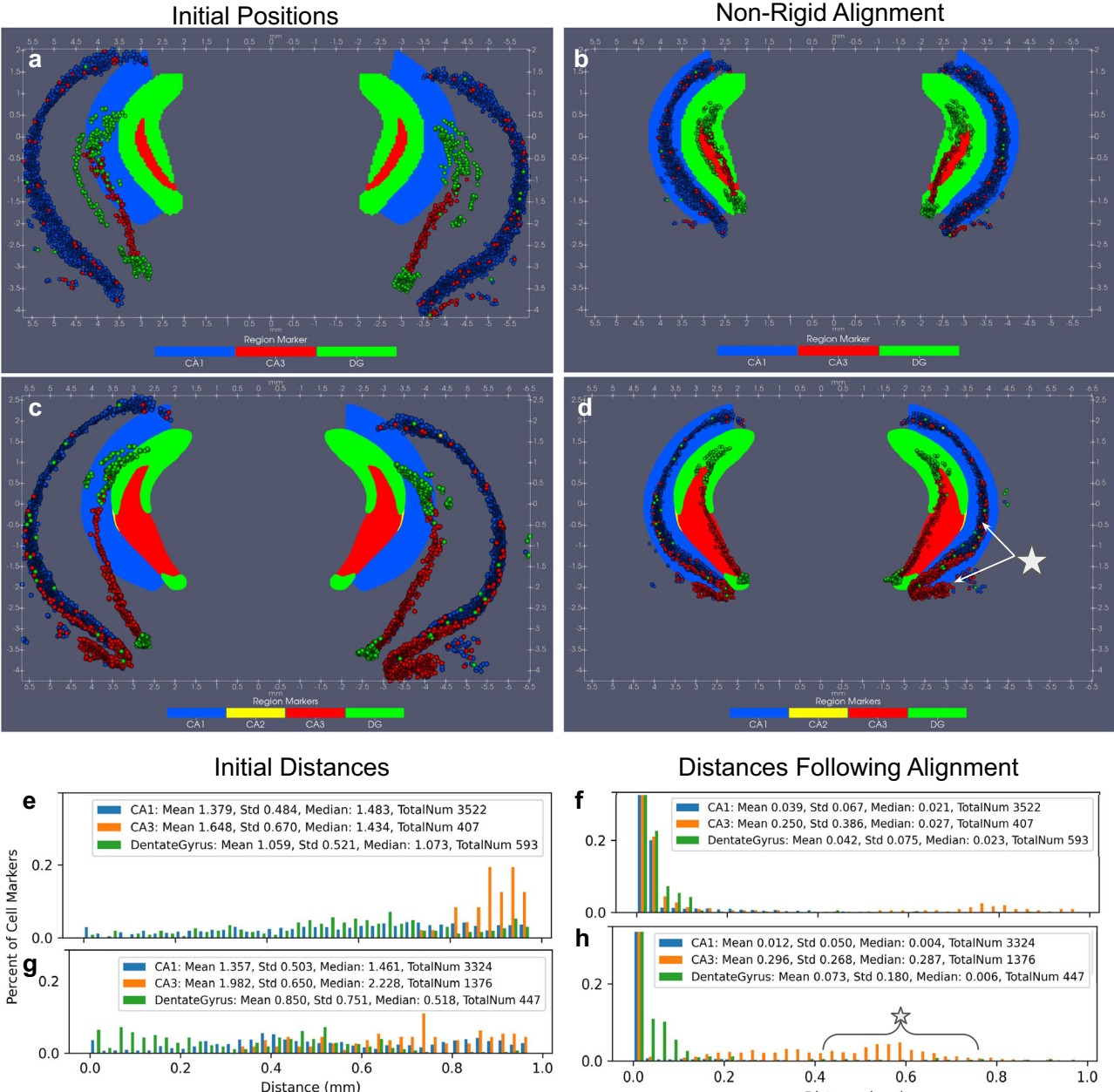

**Fig. 4 | Quantification of accuracy in two coronal slices of CCFv3 mapped to two corresponding slices of BARseq cell data classified into 52 cell types. a**, **b** Initial positions of cell markers (circles) from BARseq sections in three hippocampal regions (CA1, CA3, Dentate Gyrus (DG)) and initial positions of equivalent CCFv3 regions in corresponding coronal sections ($Z = 877$ and $Z = 837$ out of 1320). **e**, **g** Set distance of cell markers to CCFv3 region in initial positions. **c**, **d** Positions of cell markers (circles) in BARseq sections mapped to CCFv3 coordinates with xIV-

LDDMM and positions of equivalent CCFv3 regions in corresponding coronal sections. **f**, **h** Set distance of cell markers to CCFv3 region in aligned positions following estimation of diffeomorphism with xIV-LDDMM. Percents of cell markers clipped to 25% in all histograms, with 70%, 50%, and 60% cell markers within 10 μm in CA1, CA3, and DG, respectively in (**f**) and 95%, 35%, 60% cell markers within 10 μm in CA1, CA3, and DG, respectively in (**h**).

We also compared our joint image-varifold-based approach in xIV-LDDMM to a classical image-based approach matching, estimating only a geometric transformation and matching based on foreground-background (Fig. 5). We mapped one of the same CCFv3 sections ($Z = 837$) to an image rendering of the BARseq cell section, with grayscale intensity capturing density of cells over space. Both CCFv3 and BARseq images were discretized at 50 μm resolution to estimate the mapping, with the latter computed with Gaussian smoothing, and image-based LDDMM, as described in ref. 17 was used to estimate optimal diffeomorphic transformation of the atlas to the target. To compare these methods more directly, a 10 μm BARseq image was

transformed with the estimated diffeomorphism to the CCFv3 space (Fig. 5a) as closer to the particle-based representation at full resolution used for estimation of the diffeomorphism in the image-varifold based mapping (Fig. 5c). At this resolution, ~1.5 million pixels are needed to model the BARseq data as an image whereas only ~90k particles are needed in the image-varifold representation. In addition to the computational and memory expense of image handling, we compromise our ability to delineate one cell from another, as evidenced in the blur in the image rendering (Fig. 5a) versus image-varifold representation (Fig. 5c), with 75% of cells having a nearest neighbor less than 23 μm away and 25% having a nearest neighbor less than 3.5 μm away in the

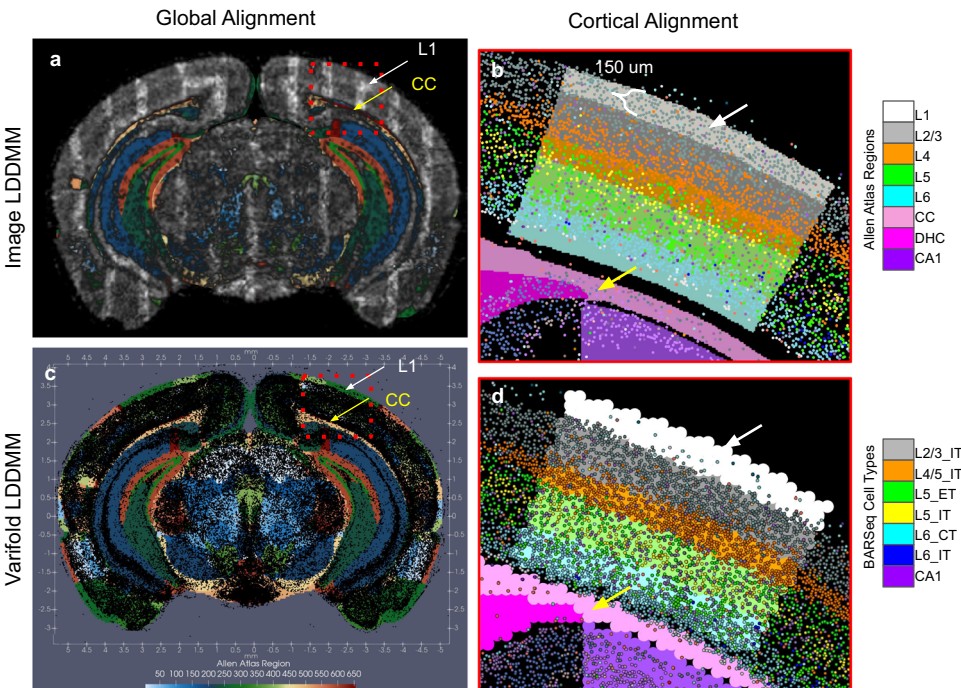

**Fig. 5 | Comparison of mapping CCFv3 section $Z = 837$ out of 1320 to corresponding BARseq cell data using image-based LDDMM[17] versus image-varifold-based mapping method. a** Estimated diffeomorphic transformation applied to 10 μm smoothed cell density image to bring into overall alignment with CCFv3 section in CCFv3 coordinates. Source and target images rendered at 50 μm for image-based LDDMM, with matching based on foreground/background in CCFv3 section to smoothed cell density in BARseq. **b** Estimated diffeomorphic transformation applied to 5 μm cell type image highlighting noticeable misalignment in (**a**) of high cell density areas to layer 1 and white matter structures (corpus

callosum and dorsal hippocampal commissure) in CCFv3 section with image-based LDDMM. Layer-by-layer mismatch (~150 μm) observed between characteristic BARseq prescribed layer-specific cell type and CCFv3 delineations. **c** Estimated diffeomorphic transformation with xIV-LDDMM applied to BARseq section to bring into overall alignment with CCFv3 section in CCFv3 coordinates without misalignment in white matter structures. **d** Layer-by-layer matching in CCFv3 partition of cortex to cell types in BARseq following image-varifold based mapping.

BARseq capture. Nevertheless, with a kernel of bandwidth 10 μm used to generate the 10 μm smoothed image and most regions at least on the order of 100 μm in width at their narrowest (e.g. corpus callosum), the majority of cell mass stayed primarily within the initial region it was found.

Regarding alignment, an image-based approach can achieve global alignment of the template to the target (Fig. 5a), with initial differences in scale and shape (Fig. 4b) resolved with the estimated transformation. However, we see clear differences in alignment on the order of 50−150 μm by examination of the overlap in the different layers of the cortex and adjacent corpus callosum (Fig. 5b, d). Importantly, as described in "Data model and optimization problem: image varifolds and transformations for molecular scales based on varifold norms" section, in the joint image-varifold-based approach to estimating geometric transformations and latent distributions, we model each atlas region as homogeneous and stationary with respect to space. This gives an optimal alignment between the atlas and target that maximizes similarity in distribution over features across each site in a single atlas region while minimizing the energy of the geometric deformation (diffeomorphism). We would consequently expect this to skew emphasis away from the foreground-background boundaries that typically govern image alignment and instead highlight the underlying assumptions in the architecture of the cartoon atlas, whose boundaries were initially constructed so as to maximize the homogeneity of the region. This becomes clear in comparing the alignment of BARseq to CCFv3 section in areas of low versus high cell density where these areas of low density correctly align with layer 1 and corpus callosum (CC) as a result of the image-varifold based approach (Fig. 5d) but not in the image-based approach (Fig. 5b). Notably, the corpus callosum and dorsal hippocampal commisure appear with equivalently

low cell density on both right and left hemispheres in the image-varifold-based approach (yellow arrow) whereas we see these areas partially covered by CA1 cells, particularly in the right hemisphere in the image-based approach. Layer 1 (white arrow) is covered by high cell density around the entire circumference of the coronal section in the image-based approach, whereas it is visible, with relatively few cells mapping to it, as expected, in the image-varifold-based approach. Supplementary Fig. 4 shows the relative cell density in each of the cortical layers and white matter structures within the neighborhood of the primary visual cortex, with xIV-LDDMM showing more accurate alignment with higher levels (4−6×) in layers 2−6 compared with layer 1 and white matter. Image-based LDDMM, in contrast, yields similar levels of cell density across all of these structures, with the band of high cell density in BARseq layers 2−6 covering the entirety of the cortex, thus generating overestimates (4−6×) of cell density in layer 1. Cell density within CA1 also covers the areas of the corpus callosum, leading to overestimates of density there. Finally, as the image-varifold-based approach jointly considers both variations in the total density of cells and relative distribution of cell type, we see within each layer of cortex delineated in the CCFv3, a majority of corresponding cell types indicating correct layer-by-layer alignment (Fig. 5d). Classification of cells by nearest atlas region yields 88% of layer 2/3 cells (total 443), 86% of layer 4/5 cells (total 436), 87% of layer 5 cells (total 278), and 70% of layer 6 cells (total 608) classified correctly according to cortical layer, within a 1.5 mm² section through the primary visual cortex shown in Fig. 5. In contrast, for a similar section in the result of image-based mapping (Fig. 5b), 38% of layer 2/3 pixels (total 308), 36% of layer 4 pixels (total 269), 64% of layer 5 pixels (total 225), and 78% of layer 6 pixels (total 326) are classified correctly according to nearest cortical layer.

## Cross-replicate comparison and atlas construction

As described in the Introduction, two key applications of xIV-LDDMM are in cross-sample comparison and atlas construction. We illustrate both of these applications in considering cell-typed MERFISH sections taken from three separate mice at approximately the same coronal level (Fig. 6a–c). CCFv3 section $Z = 890$ out of 1320 was mapped separately to each of these targets with both geometric transformations and latent feature distributions estimated in each case. As in the

classical imaging setting with LDDMM, comparison of the estimated diffeomorphisms taking the CCFv3 section to each respective target section offers one metric of similarity between both atlas and target and across targets[37,38] (Supplementary Fig. 5A–C). Here, the CCFv3 section contracts to a similar extent (-0.5) in all three mice in areas of the midbrain around the periacqueductal gray matter. In contrast, levels of contraction/expansion across the cortical layers in the primary visual area range from 0.7 to 1.4 across the three mice,

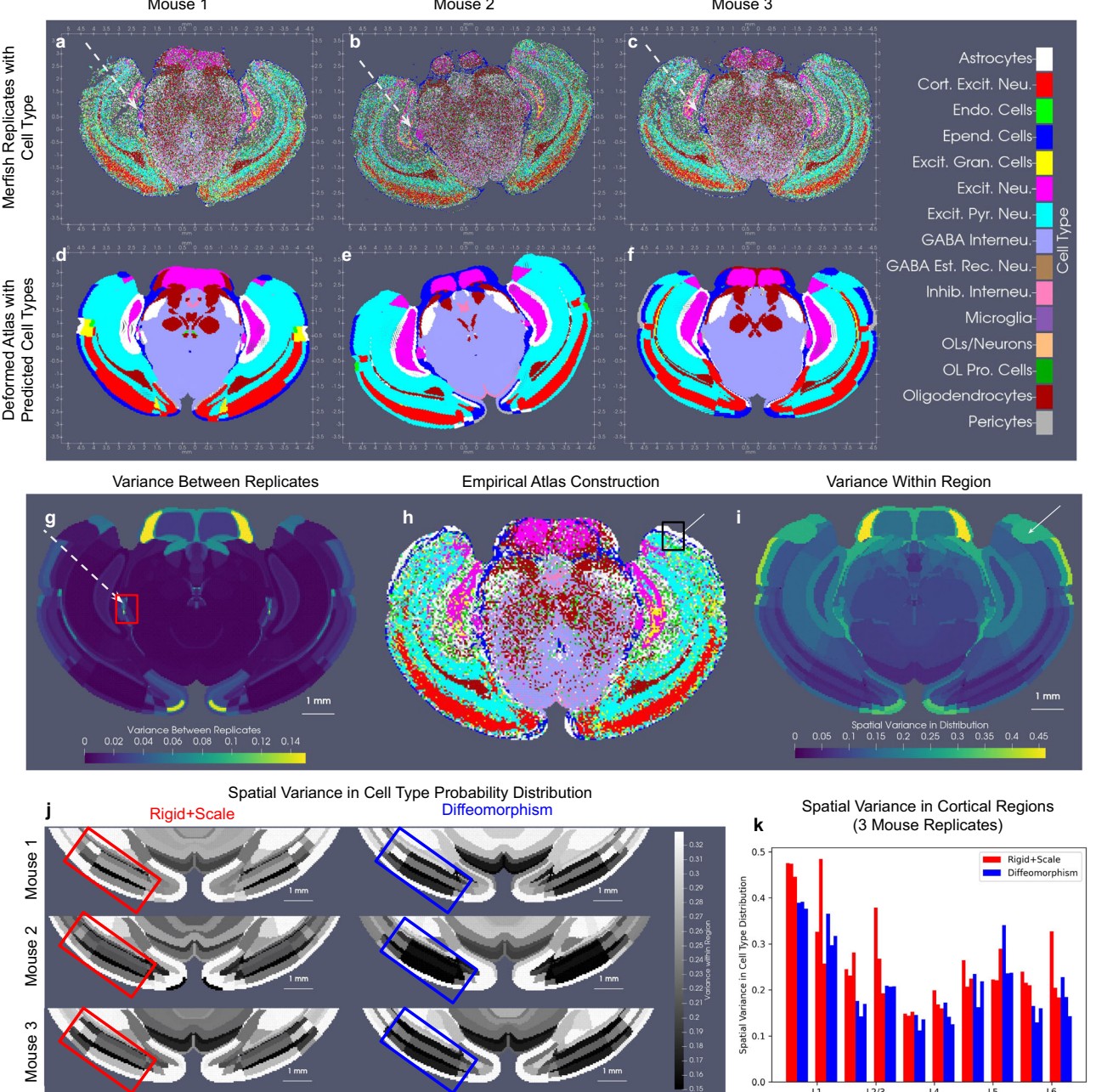

**Fig. 6 | Cross-replicate comparison and atlas construction from mapping CCFv3 section $Z = 890$ out of 1320 to MERFISH cell-typed coronal sections from three mice at approximately the same location. a–c** Position and cell type for all cells measured in each mouse section. **d-f** CCFv3 section geometrically transformed to each target space with the most probable cell type in estimated latent distribution shown for each atlas region. **g–i** Initial CCFv3 section with summative statistics of empirical probability distribution over cell types estimated from the inverse transformation of all three replicates to CCFv3 coordinates. **g** Variance in estimated empirical probability distributions across replicates. White dotted arrow

highlights area around medial geniculate nucleus with variance in cell types between replicates shown in (**a–c**). **h** Most likely cell type in empirical distribution estimated from all three replicates. **i** Spatial variance in empirical probability distribution per each atlas region with white arrow highlighting region of outer cortex varying in ependymal cell and excitatory pyramidal neuron distributions across the space of the region. **j** Spatial variance in empirical cell type distribution following rigid+scale (left) versus diffeomorphic (right) transformations in each of the three mice. **k** Reduction in spatial variance in two adjacent areas of cortex (red/blue box in **j**) following rigid+scale or diffeomorphic transformation in mice.

indicating differences in geometric shape. In xIV-LDDMM, the jointly estimated latent feature distributions offer a second metric of similarity (Supplementary Fig. 5D–G). Differences in the predicted cell type with highest probability occur, for instance, in the perirhinal and ectorhinal areas with excitatory granule cells, excitatory pyramidal neurons, or cortical excitatory neurons predominant in the different mice (Fig. 6d–f). The sample variance for each atlas region across the three estimated cell type probability distributions, modeled as vectors, $p \in \mathbb{R}^C$, for $C = 33$ cell types and $|p| = 1$, measures difference not just in the single cell type with highest probability, but amongst the entire estimated distributions over cell type for each mouse (see "Empirical distribution estimation and cross-replicate statistical comparison" section). Sample variance (Supplementary Fig. 5F, G) is computed as $\frac{1}{N-1} \sum_{i=1}^{N} \| p_i - \bar{p} \|_2^2$ in $\mathbb{R}^C$, with $N = 3$ and $\bar{p}$, the sample mean (Supplementary Fig. 5D, E). Notably, the area of the medial geniculate nucleus, while exhibiting excitatory neurons as the predominant cell type across mice, exhibits high variance in total distribution. This region is magnified in Supplementary Fig. 6A, illustrating this variance in relative distribution of excitatory neurons, astrocytes, endothelial cells, and ependymal cells.

Regarding atlas construction, the sample mean cell type distribution across those latent distributions estimated jointly with geometric transformations (Supplementary Fig. 5D, E) gives one potential construction of a cell type atlas over space. A finer-grained atlas can alternatively be achieved in this setting by pulling back each target MERFISH section onto the same CCFv3 section via the inverse estimated diffeomorphism (Fig. 1). This eliminates some of the variance in geometry between targets (Supplementary Fig. 5A–C), facilitating more direct comparison of cell type distributions. Particles (analogous to foreground pixels) in the CCFv3 section serve as a scaffold for resampling mapped cells to with either a nearest neighbor assignment or dispersion of cell mass according to kernel choice (see "Empirical distribution estimation and cross-replicate statistical comparison" section). Atlases can be constructed from the resampling of each target individually, with variance across the replicates' empirical cell type probability distributions (Fig. 6g) similar to those estimated jointly in xIV-LDDMM (Supplementary Fig. 5D–G), but also together for generation of a population average (Fig. 6h). Importantly, each particle, here, is treated independently without assuming homogeneity in the cell type distribution across particles belonging to the same atlas region. This is in contrast to the distributions estimated jointly with geometric transformations (Fig. 6d–f), and consequently results in more locally varying distributions. Overall, ventral areas exhibit higher spatial variance than dorsal areas (Fig. 6i), exhibiting that in such regions, an assumption of homogeneity within these regions may not always be appropriate. The white arrow, in particular, points to an area of cortex where this spatial variance is amongst the highest, with the outermost portion of cortex predominantly composed of ependymal cells versus the innermost portion predominantly excitatory pyramidal cells (see Supplementary Fig. 6B).

The spatial variance in cell type distribution within each atlas region and the variance in distribution between replicates of a given population both influence the statistical power needed to detect biologically relevant differences in these distributions between populations. Classically, they are accounted for in mixed effects models[47] aimed at detecting differences in group features. For instance, we have looked at differences in atrophy rate of medial temporal lobe structures between control and diseased cohorts in Alzheimer's disease[19,48]. Here, we observe greater spatial variance across ventral regions than dorsal regions (Fig. 6i), with particularly minimal variance in layers of cortex. Importantly, the use of non-rigid geometric transformations (diffeomorphisms) to pull back targets into CCFv3 coordinates reduces this spatial variance in distribution across cortical layers (Fig. 6j, k) compared with rigid and scaling transformations only in each of the three mice. Consequently, in

settings of comparing groups of replicates under different experimental conditions, we would expect this reduction in variance to facilitate detection of significant differences in cell type distribution per CCFv3 region between them.

Here, the small sample size ($n = 3$) of replicates, which are all produced under control conditions, precludes any definitive statistical statement about variations in cell type distributions observed between them and within CCFv3 regions. Nevertheless, summation across all CCFv3 regions of the variance per individual cell type probability between replicates and within each region (Fig. 7) elucidates the relative contribution of each cell (sub)type probability to the variances observed in Fig. 6g, i. We observe, for instance, with both types of variance, a large contribution from the variance in ependymal cell type probabilities (Fig. 7a, b). Specifically between replicates, excitatory neurons exhibit the second largest variance in cell type probabilities, as evidenced in the area of the dentate gyrus (Fig. 7c–e). With regard to spatial variance, we see specific astrocyte subtype probabilities with large variance, exhibited particularly in the areas of CA1 (yellow arrow) and the pons (orange arrow) both medially to laterally within the regions and when considering left versus right hemispheres (Fig. 7f, g). In contrast, other astrocyte subtype probabilities (Fig. 7h) are seen to be consistent over space, in line with our assumption of homogeneity within each CCFv3 region.

## Extension to additional tissue modalities for within and across species comparison

Constructed across institutions with varying combinations of molecular, chemical, genetic, and electrophysiological signals, multiple atlases per species now exist with different levels of granularity and intended applications[1,2,49–52]. Given this plethora of atlases, each of which might define a different partitioning scheme over the same area of tissue, questions of comparison and relevance of each atlas to emerging molecular and cellular signatures naturally arise[53]. While some atlases have been defined in the same coordinate framework—often achieved through existing methods of image registration or manual alignment[53]—many exist in different coordinate frameworks. Together with mismatches in the number, type, and positioning of partitions, this poses a challenge not only to the evaluation of each atlas ontology's fit to a molecular target but also to the ready comparison of atlas to atlas and the establishment of a clear metric of similarity between them. Hence, a second application of our method rests in its use to map not just atlas to the molecular dataset, but one atlas to another, with the geometric and functional correspondences ($\varphi$, $\pi$) yielded by our method serving as an anchor for cross-examination of existing ontologies both within and across species.

To map tissue-scale atlas to tissue-scale atlas, we model both objects as image varifolds, as described in "Data model and optimization problem: image varifolds and transformations for molecular scales based on varifold norms" section, but where both template and target are constrained to be of constant density ($w_i = 1$ for all $x_i$ in foreground tissue). Both geometric transformations and latent feature distributions, ($\varphi$, $\pi$) are estimated, with the latter giving distributions over the target atlas features (labels) for each template atlas label, following the assumption of spatial homogeneity with each template atlas label presumed to map onto a single distribution of target atlas labels.

The joint estimation of geometric transformation, $\varphi$ and conditional feature laws, $(\pi_\ell)_{\ell \in \mathcal{L}}$ in xIV-LDDMM offers two modes of quantitative comparison between atlas ontologies. First, as in the classical image setting of LDDMM, the determinant of the Jacobian, $|D\varphi|$, of the estimated diffeomorphism, can be used as a metric of how similar the atlas ontologies are, reflective of how much boundaries of partitions move to maximize the overlap between homogeneous regions. However, unlike in classical image settings, the estimation of the additional family of feature laws here, $\pi$, affords a second metric of similarity with

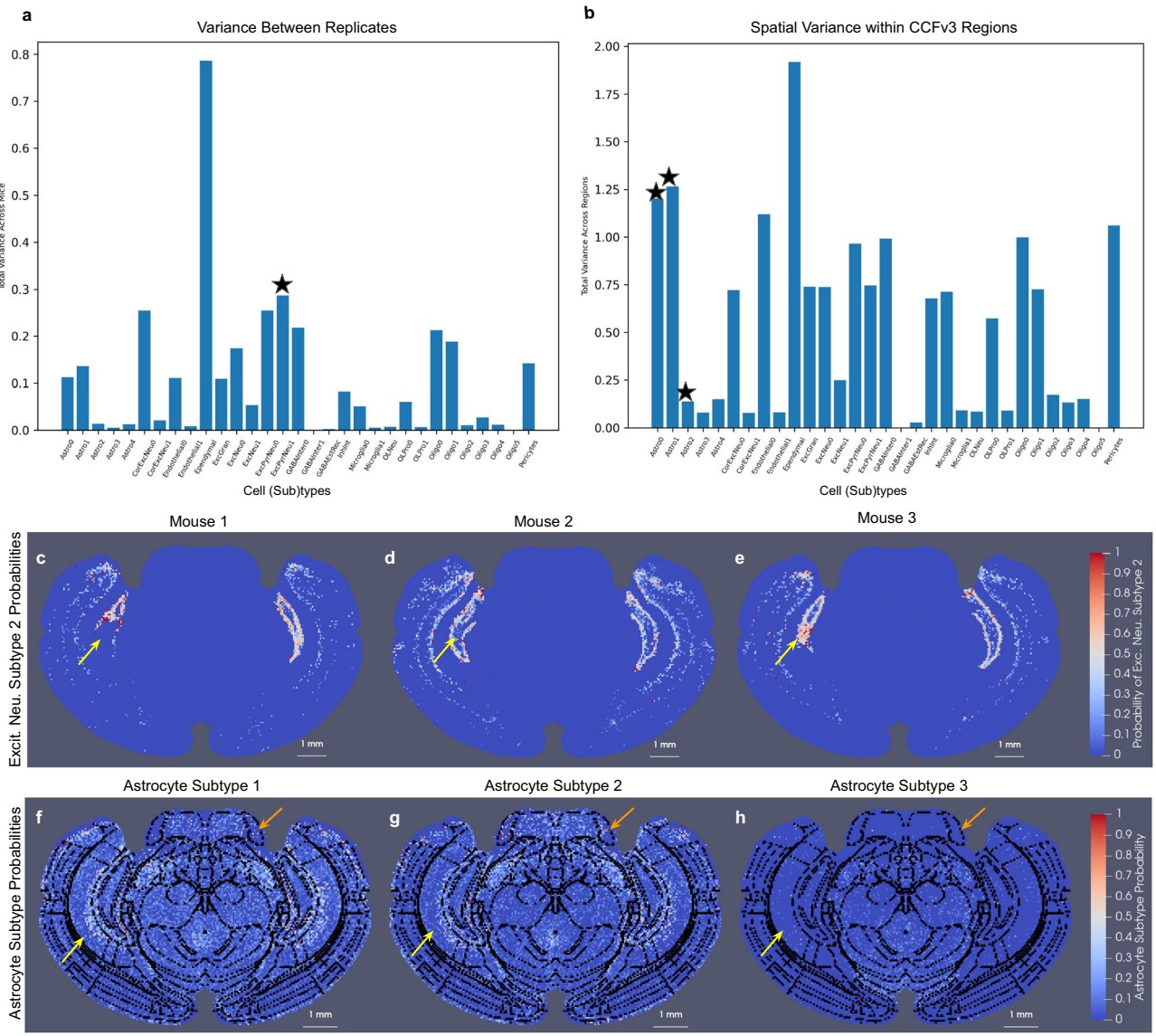

**Fig. 7 | Variance in individual cell subtype probabilities across replicates and across space within individual CCFv3 regions. a** Variance in estimated cell subtype probabilities per CCFv3 region across three replicates summed overall CCFv3 regions. **b** Spatial variance of cell subtype probabilities, estimated empirically from three pulled-back MERFISH sections, per CCFv3 region and summed across all regions. **c–e** Probability of excitatory neuron subtype 2 (star in **a**) for each of the three mice in CCFv3 coordinates. Yellow arrow highlights the area of the dentate gyrus with differences in excitatory neuron subtype 2 probabilities. **f–h** Probability for astrocyte subtypes 1,2, and 3 (stars in **b**) in empirical distribution computed from all three mice in CCFv3 coordinates (most likely cell type shown in Fig. 6h). Yellow arrow highlights area of CA1 with differences in astrocyte probability medially to laterally in subtypes 1 and 2 but not 3. Orange arrow highlights differences medially to laterally and left and right in areas of the pons in astrocyte probability for subtypes 1 and 2 but not 3. Black lines indicate boundaries between CCFv3 regions.

the computation of the entropy of the estimated conditional feature distributions.

The first panel in Fig. 8 shows the results of mapping one mouse atlas ontology to another with the Z section 680 in the CCFv3 mapped to the corresponding section in the Kim Lab Developmental atlas, termed Developmental Common Coordinate Framework (DevCCF)[51], (Fig. 8a–c and vice versa Fig. 8d–f). Allen CCFv3 and DevCCF anatomical delineations utilize two distinct ontologies based on cytoarchitecture and genoarchitecture, respectively[1,51,54]. Fig. 8a, d depict the geometry of the section under each ontology, with the CCFv3 section hosting ≈140 independent regions and the DevCCF section ≈80. In this setting, both atlases are published in the same coordinate framework, giving $\varphi = Id$ and thus, highlighting, instead, the estimated distributions over the other ontologies. Figure 8b, e depict the estimated conditional probability distributions, $p'_i, i \in I$, for each atlas section

over the other atlas section's ontology. The label with the highest probability in these distributions is plotted for each simplex in the mesh and which is consistent across each partition of each original atlas, given the homogeneity assumption in our model (i.e. a single $\pi_\ell$ for each $\ell \in \mathcal{L}$). The comparatively larger set of labels in the CCFv3 ontology results in labels being omitted from the corresponding estimated set of labels on the DevCCF ontology section Fig. 8e in the isocortex while multiple regions in the CCFv3 ontology carry the same most probable region in the DevCCF ontology. The differences in granularity with which each atlas segments the various areas of tissue are captured by the entropy of the estimated conditional feature distributions, for each simplex of the mesh (Fig. 8c, f). The entropy of the distributions estimated for the DevCCF ontology over the CCFv3 ontology Fig. 8f is on average, higher, than that of the distributions estimated for the CCFv3 ontology Fig. 8c, with probability mass

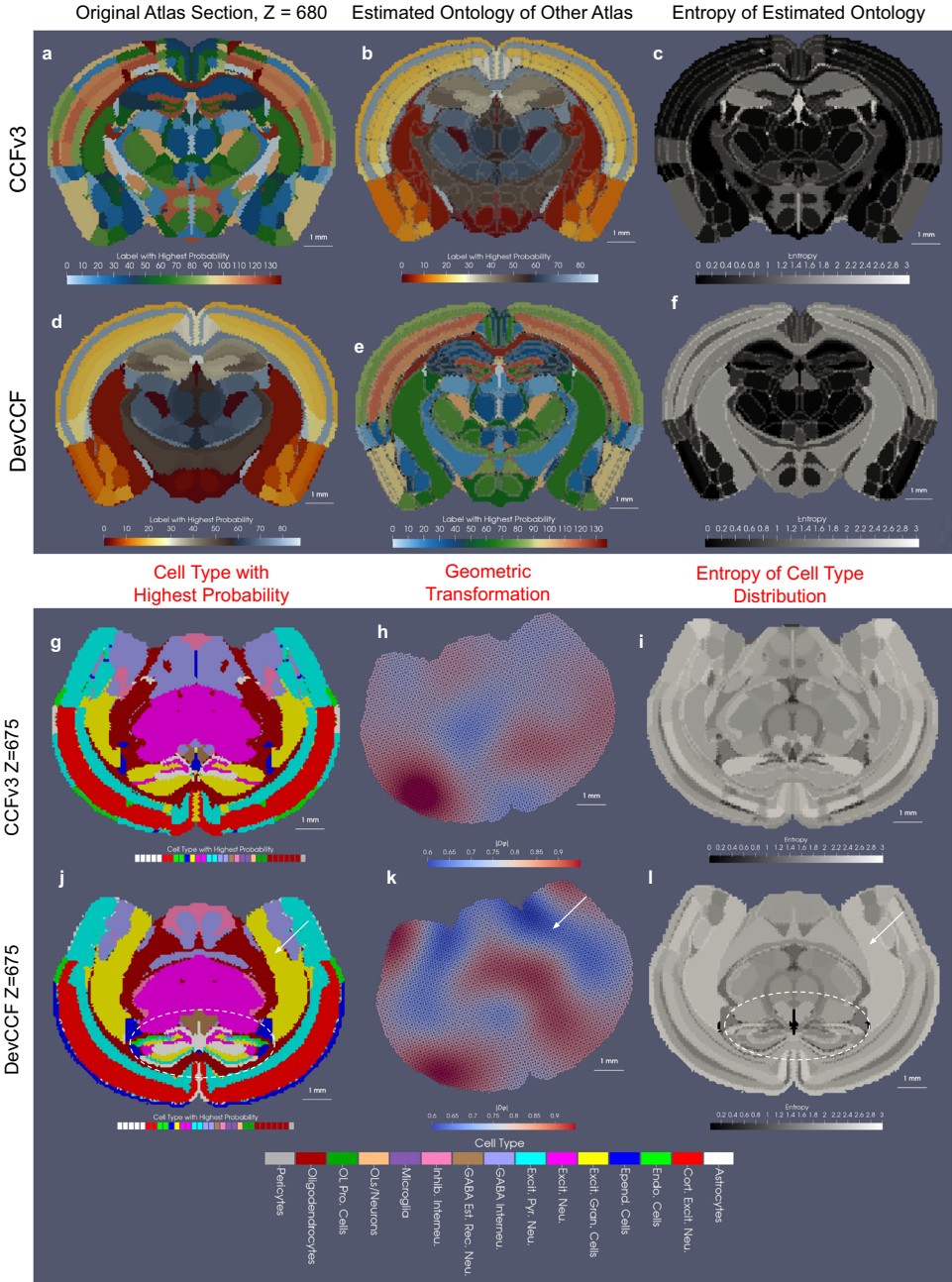

**Fig. 8 | Comparison of corresponding sections in CCFv3 and DevCCF atlases both independently and in the context of being mapped to a cell-based MER-FISH target. a, d** Original CCFv3 and DevCCF ontologies at location $Z = 680$ out of 1320. **b** CCFv3 geometry with predicted DevCCF atlas ontology. Delineations of original CCFv3 partitions are outlined in gray. **e** DevCCF atlas geometry with predicted CCFv3 ontology. Delineations of original DevCCF partitions are outlined in gray. **c, f** Entropy of predicted ontologies, with higher entropy values (light) indicating less 1:1 correspondence between ontologies. **g** Predicted cell type with highest probability per simplex in CCFv3 atlas following mapping to MERFISH target (shown in Fig. 3) with xIV-LDDMM. **j** Predicted cell type with a highest probability per simplex in DevCCF atlas following mapping to same MERFISH target with xIV-LDDMM. **h, k** Estimated geometric transformation, $\varphi_1$, in each setting applied to each atlas, with areas of expansion (red) and contraction (blue) as measured by the determinant of the Jacobian. White arrow highlights differences in ontologies in amygdala and striatum designation leading to different geometric transformations. **i, l** Entropy of estimated cell type distribution per simplex in atlas. Circled area of the hippocampus highlights differences in atlas ontologies leading to differences in the estimated entropy of cell type distributions.

distributed across ~5–7 different CCFv3 regions for each DevCCF region of the isocortex. Nevertheless, we see close to 1:1 correspondence between CCFv3 and DevCCF labels in the center section of the slice, where the entropy of the estimated distributions is near 0.

Atlas comparison can be conducted both independently and within the context of particular molecular targets. For example, here, we compare corresponding sections of the CCFv3 and DevCCF atlases[51] via mapping each section to the same cell-segmented

MERFISH section and comparing both estimated diffeomorphisms and cell type distributions (Fig. 8g–l). Cell type distributions are compared via visualization of the cell type with the highest probability within each region (Fig. 8g, j) and the overall entropy of the estimated distribution per region (Fig. 8i, l). The areas of the hippocampus (dashed circle) and striatum and amygdala (arrow) are partitioned with different levels of granularity. This leads to different optimal geometric transformations, as characterized by the determinant of the Jacobian

(Fig. 8b, e), and different predicted cell type distributions (Fig. 8c, f). As discussed above, though both atlases are published as geometrically aligned[51], the diffeomorphism solving the variational problem, here, transforms geometrically the homogeneous regions between the atlas and target. Hence, regions of the amygdala and striatum undergo significant contraction in the optimal mapping of DevCCF but not CCFv3 to MERFISH given the partitioning of this region into fewer and thus larger presumed homogeneous regions in the DevCCF atlas. Just as entropy in estimated distribution can be used to compare atlas to atlas, directly, we examine it here comparatively between atlases as an indication of which regions in which atlas achieve more or less homogeneous cell type distributions (Fig. 8i, l). Here, the hippocampus is more finely partitioned in the Kim atlas, which yields lower entropy distributions over cell types than in those estimated for the CCFv3.

Atlas ontologies can be mapped not just within species but also across them, where both geometric transformations and estimated ontology distributions, together reflect metrics of comparison between the two. As an example, we map a coronal section, $Z = 537$, in the CCFv3 to a coronal section, $Z = 628$ in the Waxholm Rat Brain Atlas[52], with both sections chosen to correspond as sections through the anterior commissure (Fig. 9). Here, the CCFv3 ontology is comprised of $\approx 120$ regions (Fig. 9a) with $\approx 30$ for the Waxholm atlas section (Fig. 9d). Initial differences in size and shape exist between the two tissue sections (Fig. 9b). After scaling the volume of the mouse brain by 1.5, additional deformation, with magnitude given by the determinant of the Jacobian, $|D\varphi|$, distorts both internal and external tissue boundaries to align homogeneous regions in each atlas, such as cingulate area to cingulate area (white arrow). Estimated distributions over the Waxholm ontology labels for each region in the CCFv3 are shown in Fig. 9c, f, summarized by the maximum probability label (Fig. 9f) and measures of entropy (Fig. 9c), which highlight in gray,

CCFv3 regions mapping to $\approx 3$–4 Waxholm regions versus those in black achieving 1:1 correspondence.

## Extension to additional cellular-scale modalities for image segmentation

With the computational constraints imposed by modeling spatial transcriptomics datasets as classical images, this work has emphasized the mapping of tissue-scale atlases to molecular datasets generated by emerging image-based rather than the often regularized spot-based spatial transcriptomics technologies. However, as exhibited through the modeling of atlas sections as image varifolds, themselves, both as template and target (see "Extension to additional tissue modalities for within and across species comparison" section), the xIV-LDDMM is equally capable of aligning tissue-scale atlases to alternative molecular and cellular-scale modalities. As an example, we take a DAPI-stained tissue section (Fig. 10), digitized at 2.5 μm resolution, corresponding to the gene-based MERFISH section shown in Fig. 1. The DAPI image (~13 million pixels) is converted to an image-varifold particle representation by discretization of its image values into ~35 discrete bins, and selection of foreground pixels with corresponding image values in the later 30 of these bins. Approximately 250k particles are used to represent these pixels, with each particle pertaining to a set of 25 neighboring pixels ($5 \times 5$ square). Particles each carry the distribution of bins into which the image values of these 25 pixels fall, retaining the individual values of each foreground pixel at the highest resolution in contrast to typical image downsampling, in which a pixel only captures the single mean image value of its neighbors.

Figure 10 depicts this particle representation of the DAPI image and its positioning before and after alignment with the corresponding CCFv3 section. Similar to the BARseq mapping illustrated in Fig. 5, we see areas of lower cell density (fewer foreground pixels) versus higher cell density (greater and higher intensity foreground pixels) aligning to

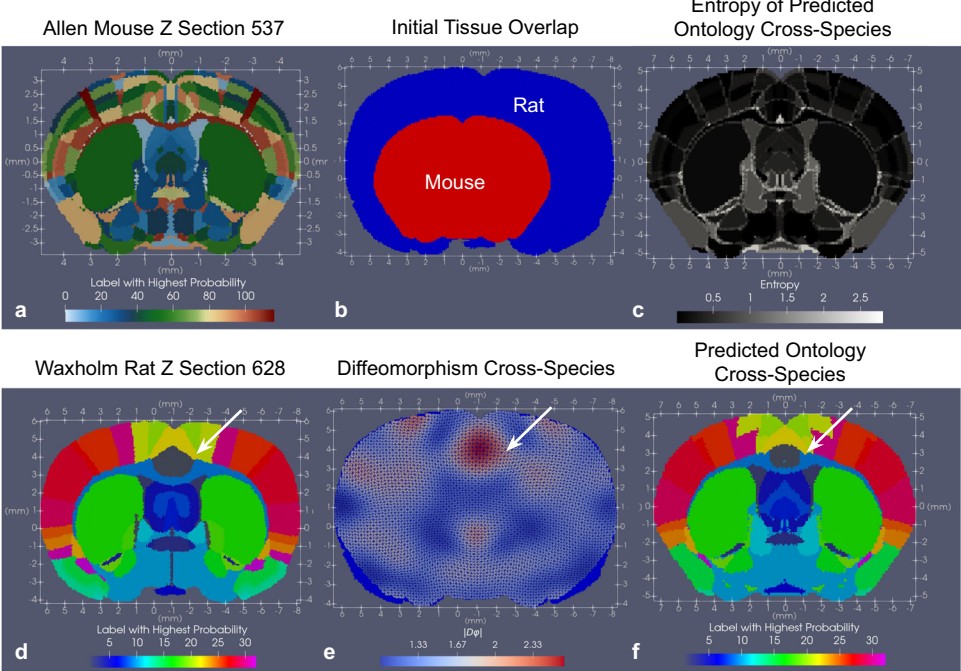

**Fig. 9 | Results of mapping coronal section $Z = 537$ of CCFv3 mouse atlas to corresponding coronal section of Waxholm Rat Brain Atlas at $Z = 628$, both chosen to be through the anterior commissure. a** CCFv3 section in CCFv3 coordinates. **d** Waxholm section in Waxholm atlas coordinates. **b** Initial tissue overlap between mouse and rat atlas sections shown in Waxholm atlas coordinates. **e** Resulting overlap between rat section (blue) and mouse section following action of estimated diffeomorphism on mouse section. The determinant of the Jacobian

highlights areas of expansion (red) and contraction (blue) in the mouse section deforming to match the rat section, with white arrow highlighting expansion in the cingulate area needed to match the region in the mouse to the corresponding region in the rat. **c** Entropy for each mouse region's predicted distribution of rat labels. **f** Predicted rat label with the highest probability for each region in CCFv3 mouse ontology.

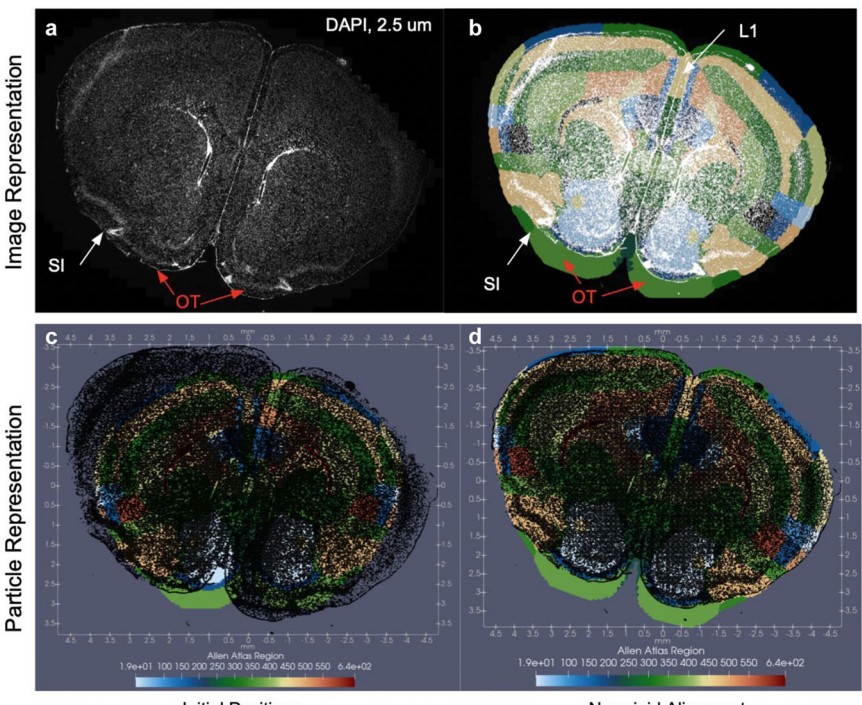

**Fig. 10 | Application of image-varifold-based method for mapping CCFv3 section to DAPI-stained image of tissue corresponding to MERFISH section. a** Original DAPI-stained image, digitized at 2.5 μm resolution for tissue section measured with MERFISH technology (Fig. 1). **c** Image-varifold particle representation (black points) of DAPI-stained image overlaying the corresponding CCFv3 section in their respective initial coordinate spaces. Thresholded fore-ground pixels from (**a**) converted to particle image-varifold representation over a feature space of ~30 binned grayscale values. **d** Alignment of CCFv3 section to DAPI particles following diffeomorphic transformation to the DAPI coordinate space. **b** Alignment of CCFv3 section and DAPI particles in image format, with the deformed CCFv3 section image generated by resampling the deformed CCFv3 particles onto a regular 2.5 μm grid. White arrows highlight areas of alignment in the area of the substantia inominata (SI) and layer 1 of the cortex whereas red arrows highlight areas of questionable alignment in the area of the olfactory tubercle (OT).

expected CCFv3 regions (e.g. layer 1 (L1) especially between the hemispheres, and substantia inominata (SI), respectively). Interestingly, the area of the olfactory tubercle (OT) in the CCFv3 section which is initially quite larger than that in the DAPI image remains large following geometric transformation to the target space, with optimal cost minimization favoring this alignment to one in which this area would drastically contract.

Finally, while the estimated geometric transforms here could equivalently pull back the DAPI image into the space of the CCFv3 section, as done in the setting of atlas construction and cross-replicate comparison (see "Cross-replicate comparison and atlas construction" section), we show, instead, the CCFv3 section transformed to the space of the DAPI target as an illustration of how our methodology can be used in settings of image segmentation. This facilitates the extraction of measures in a particular subset of regions, that can be directly compared to other types of measures (e.g. gene expression, cell type distributions) as detected by other types of technologies and equivalently localized to these same CCFv3 regions through mappings computed with xIV-LDDMM, as demonstrated here in "Generalizability of xIV-LDDMM to settings of complete and incomplete gene-based and cell-based data," "Quantitative and comparative evaluation of xIV-LDDMM," and "Cross-replicate comparison and atlas construction" sections. For instance, a comparison of the mean DAPI signal (Supplementary Fig. 7) to the maximally expressed gene per CCFv3 region, as shown in Fig. 1, illustrates a positive correlation between DAPI intensity and *Whrn* expression, particularly in areas bordering the corpus callosum versus negative correlation with *Mdga1* expression in outer cortical layers. Finally, as described in the setting of resampling image varifolds across scales[34], we demonstrate the feasibility of translating between particle and image representations in resampling

the deformed CCFv3 particles with Gaussian smoothing onto the same 2.5 μm grid of the original DAPI image (Fig. 10b). Note that this smoothing extends the borders of the atlas section slightly beyond their respective positioning as particles, as seen in the difference in overlay laterally within L1 between image and particle representations (Fig. 10b, d, respectively).

## Discussion

We have introduced, here, a universal method, xIV-LDDMM, for mapping tissue-scale, 'cartoon' atlases to molecular and cellular datasets arising in the context of emerging transcriptomics technologies. We root our method in the modeling of each object as an image varifold, as previously described[35], and map across scale and modality by simultaneously estimating a geometric transformation with classical deformation tools of LDDMM[17] and latent distribution over target features for each atlas region, under an assumption of homogeneity. Offering multiple means of implementation and optimization schemes (see "Alternating LDDMM and quadratic program algorithm for joint optimization" section), we have used both point cloud and mesh-based implementations of image-varifold objects, showcasing successful estimation of geometric mappings and latent feature distributions in both cases.

As presented, xIV-LDDMM fills a current need, as highlighted previously[55], for universal tools that can integrate the diverse types and large quantities of data emerging from the evolution of both transcriptomics and imaging technologies over the last decade. With each technology generating a slightly different perspective and a different set of animal or human samples to compare, a method that can stably handle the format of past, current, and future datasets will be paramount to integrate both new findings with the vast number of

datastores currently available across institutes. The image-varifold framework used here is general enough to model classical tissue-scale imaging data (as demonstrated in our atlas-to-atlas mappings), digital pathology (as demonstrated in our example with the DAPI-stained image) ("Extension to additional cellular-scale modalities for image segmentation" section), and emerging transcriptional data from both image-based and spot-resolution technologies that might generate data complete or partial tissue captures, irregularly sampled across space. Therefore, it provides a gateway for integrating data historically curated through MRI, immunohistochemistry, and other staining and imaging procedures in addition to the emerging transcriptomics methods. Importantly, this spatial integration will enable the correlation of disease signatures at different scales, as measured through different technologies. For instance, we aim to correlate tau tangle and amyloid-beta pathology in Alzheimer's disease to gene expression and imaging biomarkers, as vital for furthering our mechanistic understanding of the disease and developing early diagnostic strategies. Additionally, the specific estimation of diffeomorphic mappings compared with rigid+scale transformations decreases the spatial variance in feature distributions across regions, increasing the statistical power for detecting differences in feature distributions (e.g. gene expression, cell composition) not just across replicates within a single group, but between groups of replicates under different experimental conditions. For instance, we are currently looking at differences in cell type and gene distribution following neonatal binocular enucleation as measured with BARseq in four cases versus control hemispheres[56]. As presented, xIV-LDDMM fills a current need, as highlighted previously[55], for universal tools that can integrate the diverse types and large quantities of data emerging from the evolution of both transcriptomics and imaging technologies over the last decade. With each technology generating a slightly different perspective and a different set of animal or human samples to compare, a method that can stably handle the format of past, current, and future datasets will be paramount to integrate both new findings with the vast number of datastores currently available across institutes. The image-varifold framework used here is general enough to model classical tissue-scale imaging data (as demonstrated in our atlas-to-atlas mappings), digital pathology (as demonstrated in our example with the DAPI-stained image) ("Extension to additional cellular-scale modalities for image segmentation" section), and emerging transcriptional data from both image-based and spot-resolution technologies that might generate data complete or partial tissue captures, irregularly sampled across space. Therefore, it provides a gateway for integrating data historically curated through MRI, immunohistochemistry, and other staining and imaging procedures in addition to the emerging transcriptomics methods. Importantly, this spatial integration will enable the correlation of disease signatures at different scales, as measured through different technologies. For instance, we aim to correlate tau tangle and amyloid-beta pathology in Alzheimer's disease to gene expression and imaging biomarkers, as vital for furthering our mechanistic understanding of the disease and developing early diagnostic strategies. Additionally, the specific estimation of diffeomorphic mappings compared with rigid+scale transformations decreases the spatial variance in feature distributions across regions, increasing the statistical power for detecting differences in feature distributions (e.g. gene expression, cell composition) not just across replicates within a single group, but between groups of replicates under different experimental conditions. For instance, we are currently looking at differences in cell type and gene distribution following neonatal binocular enucleation as measured with BARseq in four cases versus control hemispheres[56]. As presented, xIV-LDDMM fills a current need, as highlighted previously[55], for universal tools that can integrate the diverse types and large quantities of data emerging from the evolution of both transcriptomics and imaging technologies over the last decade. With each technology generating a slightly different perspective and a different set of animal or human samples to compare, a method that can stably handle the format of past, current, and future datasets will be paramount to integrate both new findings with the vast number of datastores currently available across institutes. The image-varifold framework used here is general enough to model classical tissue-scale imaging data (as demonstrated in our atlas-to-atlas mappings), digital pathology (as demonstrated in our example with the DAPI-stained image) ("Extension to additional cellular-scale modalities for image segmentation" section), and emerging transcriptional data from both image-based and spot-resolution technologies that might generate data complete or partial tissue captures, irregularly sampled across space. Therefore, it provides a gateway for integrating data historically curated through MRI, immunohistochemistry, and other staining and imaging procedures in addition to the emerging transcriptomics methods. Importantly, this spatial integration will enable the correlation of disease signatures at different scales, as measured through different technologies. For instance, we aim to correlate tau tangle and amyloid-beta pathology in Alzheimer's disease to gene expression and imaging biomarkers, as vital for furthering our mechanistic understanding of the disease and developing early diagnostic strategies. Additionally, the specific estimation of diffeomorphic mappings compared with rigid+scale transformations decreases the spatial variance in feature distributions across regions, increasing the statistical power for detecting differences in feature distributions (e.g. gene expression, cell composition) not just across replicates within a single group, but between groups of replicates under different experimental conditions. For instance, we are currently looking at differences in cell type and gene distribution following neonatal binocular enucleation as measured with BARseq in four cases versus control hemispheres[56].

In parallel to the development and dispersion of diverse molecular datasets, there has been continued development on the side of

set of animal or human samples to compare, a method that can stably handle the format of past, current, and future datasets will be paramount to integrate both new findings with the vast number of datastores currently available across institutes. The image-varifold framework used here is general enough to model classical tissue-scale imaging data (as demonstrated in our atlas-to-atlas mappings), digital pathology (as demonstrated in our example with the DAPI-stained image) ("Extension to additional cellular-scale modalities for image segmentation" section), and emerging transcriptional data from both image-based and spot-resolution technologies that might generate data complete or partial tissue captures, irregularly sampled across space. Therefore, it provides a gateway for integrating data historically curated through MRI, immunohistochemistry, and other staining and imaging procedures in addition to the emerging transcriptomics methods. Importantly, this spatial integration will enable the correlation of disease signatures at different scales, as measured through different technologies. For instance, we aim to correlate tau tangle and amyloid-beta pathology in Alzheimer's disease to gene expression and imaging biomarkers, as vital for furthering our mechanistic understanding of the disease and developing early diagnostic strategies. Additionally, the specific estimation of diffeomorphic mappings compared with rigid+scale transformations decreases the spatial variance in feature distributions across regions, increasing the statistical power for detecting differences in feature distributions (e.g. gene expression, cell composition) not just across replicates within a single group, but between groups of replicates under different experimental conditions. For instance, we are currently looking at differences in cell type and gene distribution following neonatal binocular enucleation as measured with BARseq in four cases versus control hemispheres[56]. As presented, xIV-LDDMM fills a current need, as highlighted previously[55], for universal tools that can integrate the diverse types and large quantities of data emerging from the evolution of both transcriptomics and imaging technologies over the last decade. With each technology generating a slightly different perspective and a different set of animal or human samples to compare, a method that can stably handle the format of past, current, and future datasets will be paramount to integrate both new findings with the vast number of datastores currently available across institutes. The image-varifold framework used here is general enough to model classical tissue-scale imaging data (as demonstrated in our atlas-to-atlas mappings), digital pathology (as demonstrated in our example with the DAPI-stained image) ("Extension to additional cellular-scale modalities for image segmentation" section), and emerging transcriptional data from both image-based and spot-resolution technologies that might generate data complete or partial tissue captures, irregularly sampled across space. Therefore, it provides a gateway for integrating data historically curated through MRI, immunohistochemistry, and other staining and imaging procedures in addition to the emerging transcriptomics methods. Importantly, this spatial integration will enable the correlation of disease signatures at different scales, as measured through different technologies. For instance, we aim to correlate tau tangle and amyloid-beta pathology in Alzheimer's disease to gene expression and imaging biomarkers, as vital for furthering our mechanistic understanding of the disease and developing early diagnostic strategies. Additionally, the specific estimation of diffeomorphic mappings compared with rigid+scale transformations decreases the spatial variance in feature distributions across regions, increasing the statistical power for detecting differences in feature distributions (e.g. gene expression, cell composition) not just across replicates within a single group, but between groups of replicates under different experimental conditions. For instance, we are currently looking at differences in cell type and gene distribution following neonatal binocular enucleation as measured with BARseq in four cases versus control hemispheres[56].

reference atlases to reflect trends in these new measures and integrate these trends across even more samples of particular species. With regard to atlas refinement and creation, as shown in "Cross-replicate comparison and atlas construction" section and the second panel of Fig. 1, the invertibility of the estimated diffeomorphism in the setting of mapping atlas to molecular target, enables the carrying of each target into the same coordinate space of the atlas. Though we illustrate an example of atlas creation from samples of a single modality, each target to which the same atlas section is mapped can be pulled back into the same coordinate space. This effectively allows different molecular modalities to be *indirectly* mapped to one another, where the challenge of matching different molecular features as needed to map one modality directly to another is facilitated through the scaffold of the presumed homogeneous regions in the atlas that anchor the estimated feature distributions for alignment to each respective molecular modality. Furthermore, this enables the construction of a composite set of molecular and cellular-scale data across technologies and modalities (e.g. digital pathology and spatial transcriptomics, as seen in the comparative mappings of the same CCFv3 section to MERFISH (Fig. 1) and DAPI images (Fig. 10)), that can be integrated into a multi-factorial atlas with defined segmentation schemes according to homogeneous regions across these features. Our method also offers a tool for re-examining and comparing existing atlas ontologies in the context of new data[55], and serves as a means for developing new atlases in the future. As described in "Extension to additional tissue modalities for within and across species comparison" section, examination of the mappings achieved between different atlases and the same molecular target offers an indirect comparison between atlases in the context of a particular molecular setting. However, this comparison can also be made directly in a context-independent setting by harnessing our method to map atlas to atlas. In the field of evolutionary biology, for instance, our method could aid in the mapping and comparison of atlases across species[57,58] and in the field of developmental neurobiology, the available atlases of the brain at different stages of development[49–51,54,59]. Finally, as shown in "Extension to additional cellular-scale modalities for image segmentation" section, xIV-LDDMM can be applied across molecular and cellular modalities, not just for incorporating them into atlas creation, but also as a means of segmentation in the classical image sense.

While the results presented here survey a wide variety of potential applications of xIV-LDDMM to mapping atlas modalities to diverse targets, there remain uncertainties and potential modes of improvement that are the subject of current and future work. First, we have presented results mapping 2D sections of 3D atlases to corresponding 2D sections of MERFISH data. Both the Allen MERFISH and BARseq data showcased here are part of an entire set of serial sections that span the whole brain. While we have approximated these sections as strictly coronal in pairing them to respective coronal slices of the CCFv3 here, typically such datasets are generated by sectioning on an angled plane, which would be best approximated by the mapping of the entire set of sections as a 3D object to the 3D CCFv3. Consequently, we are optimizing our method to compute mappings of atlas to molecular targets in 3D, where both added dimensions and added magnitudes of data contribute to the theoretical and computational complexity of the problem. Indeed, with ≈6 billion individual transcripts measured across the span of the brain, treatment of this data as a regular lattice image would require on the order of 1000 billion voxels at 1 μm resolution, which is coarser than that needed even to resolve two molecules of mRNA. Hence, it becomes even more vital to treat such data in the particle setting, as presented here, where we capture the sparsity and irregularity of the data in modeling it effectively in its lowest dimension, as 6 billion individual particle measures. Second, though we have highlighted a range of gene-based and cell-based, whole and hemi-brain datasets achieved with image-based technologies, here, the future investigation includes the use of our method to map data from additional types of technologies, such as the spot-resolution SlideSeq[44], and in additional biological settings, where we might see further disruptions of tissue architecture (e.g. with tumors) and tissue types with varying levels of the organization and therefore varying levels to which homogeneous distributions over molecular features can be seen over physical space (e.g. in heart, breast, and lung).

Indeed, we finally emphasize that central to the model posed here is the underlying assumption that each compartment has a homogeneous distribution over molecular features that is stationary with respect to space. We make this assumption to govern the estimation of latent feature distributions for each atlas compartment to take the atlas into both the physical and feature space of the target. This is in contrast to learning-based approaches that estimate features independent of such assumptions to map atlas to target across feature (e.g. imaging) modalities but often require extensive training datasets and computational power to learn such features[18,26]. This assumption stems from the inherent construction of atlases often to delineate regions of particular cell types or with a particular function, and thus, where we see a set of predominant cell types or gene types consistently across the region in the molecular scale data, as in Fig. 3. The successful alignment of tissue-scale atlas to varying molecular scale data demonstrated through the examples shown here supports the benefit of such an assumption in jointly estimating geometric transformations and latent feature distributions. However, as manifest in Fig. 2, where the expression of *Trp53i11* appears to be distributed along a decreasing gradient medial to lateral within the striatum, or explicitly through the spatial variance in empirical distribution shown for each CCFv3 region in Fig. 6, this assumption of spatial homogeneity does not always hold. The subsequent construction of empirical distributions from the pulling back of different molecular targets, as shown in Fig. 6 enables the estimation of feature distributions at a finer scale without the assumption of homogeneity and simultaneously, an evaluation of where and to what extent it holds, with notably, the use of diffeomorphic mappings achieving a further reduction in this spatial variance than rigid+scale transformations alone. Furthermore, the results presented here reflect a particular balance between expected deformation and this homogeneity assumption, imposed by the relative weighting of the separate terms in the cost function. Current work at controlling this balance further includes the addition of a term controlling the divergence of the vector field to the energy defined in the variational problem (10), which leads to solutions more robust to deformation within the interior of the tissue. Future work will also include a more rigorous evaluation of how well this homogeneity assumption holds across different tissue contexts and the effect the given balance between the two terms might have in different settings.

## Methods

### Construction of image-varifold representation for different modalities

As introduced in "Data model and optimization problem: image varifolds and transformations for molecular scales based on varifold norms" section, we can represent each image-varifold object as a point cloud (particles) or a triangulated mesh. In the first case, data is modeled as a collection of particles, each with a center $x_i \in \mathbb{R}^2$, and a measure over its feature space, $w_i p_i \in \mathcal{M}(\mathcal{F})$. Particles may be individual mRNA detections, cell centers, or image pixels at the highest resolution, with thresholding or exclusion of particular image values done in the latter case to extract foreground pixels only. Sets of particles can be downsampled to a smaller set of discrete particles by capturing neighboring particle distributions together into one particle, as used in the implementation of the DAPI image ("Extension to additional cellular-scale modalities for image segmentation" section). We refer to the resolution of the rendering to indicate the span of the neighborhood each individual particle captures with the coarser the

resolution corresponding to larger and larger neighborhoods of measurements encapsulated in a single particle.

In the case of meshes, each mesh is built from a collection of vertices indexed by the set $I$, $\mathbf{x} = (x_i)_{i \in I}$ with each $x_i \in \mathbb{R}^2$. Each simplex in the mesh is defined from the vertices denoted as $\gamma(\mathbf{x})$ and is paired with a 3-tuple with components that index the vertices of the simplex, $(\gamma(\mathbf{x}), c = (c^1, c^2, c^3) \in I^3)$ and determine the center $m(\mathbf{x}) = \frac{1}{3}(x_{c^1} + x_{c^2} + x_{c^3})$. Each triangle simplex is defined by

$$\gamma(\boldsymbol{x}) = \left\{ y \in \mathbb{R}^2 : y = \sum_{k=1}^3 a_k x_{c^k}, a_k \geq 0, \sum_{k=1}^3 a_k = 1 \right\}, \quad (5)$$

with positive orientation and volume $|\gamma_c(\boldsymbol{x})| := \frac{1}{2} \| (x_{c^2} - x_{c^1}) \times (x_{c^3} - x_{c^1}) \| > 0$.

The total mesh $\tau$ is the collection of vertices $\mathbf{x}$, and simplices and centers $(\gamma_j(\mathbf{x}), c_j = (c_j^1, c_j^2, c_j^3), m_j(\mathbf{x}))_{j \in J}$, with the simplices indexed by the set $J$, and with the resolution determining the complexity as total numbers of vertices $|I|$ and the number of simplices $|J|$ in the mesh. To complete the image varifold we append to the mesh the spatial density defined over the area of each simplex: $\boldsymbol{\alpha} = (\alpha_j)_{j \in J}$ and the field of probability laws $\boldsymbol{p} = (p_j)_{j \in J}$ on $\mathcal{F}$. Hence, we denote an image varifold implemented as a mesh similar to the normalized definition of (1) as:

$$\mu_\tau = \sum_{j \in J} w_j (\delta_{m_j} \otimes p_j) \quad (6)$$

with $w_j = \alpha_j |\gamma_j|$ giving the assumed constant density over the area of the simplex. Notably, diffeomorphisms act directly on the vertices, with the center $m_j$ of each simplex shifted according to the movement of each of its vertices. The determinant of the Jacobian is introduced to retain the same spatial density of the object before and after transformation, as in (2) but with it evaluated at the center of each simplex, and again assumed constant over the area of each simplex, approximated as the ratio in the simplex area after and before transformation:

$$\varphi \cdot \mu_\tau = \sum_{j \in J} \alpha_j |D\varphi|_{m_j} |\gamma_j| (\delta_{\varphi(m_j)} \otimes p_j). \quad (7)$$

Additionally, in the setting of meshes, it is convenient to approximate the product $|D\varphi|_{m_j} |\gamma_j|$ by the new area of each simplex following transformation, $|\gamma_j(\varphi_1(\mathbf{x}))|$.

Meshes mapped in this work were constructed using Delauney triangulation[60] on a grid defined over the support of the starting dataset with the size of each square dictated by the input resolution. Spatial density and conditional feature measures, $\boldsymbol{\alpha}, \boldsymbol{p}$, were associated with the simplices of the mesh following the assignment of each individual data point (e.g. mRNA or cell read) into its single nearest simplex. Meshes were pruned of simplices that both contained fewer than 1 data point and existed outside the largest connected component of simplices containing at least one data point. In this manner, both for atlas images and transcriptomics datasets, resulting simplex meshes spanned the entire tissue foreground.

## Molecular scale varifold norm

We define the space of image varifolds $\mu \in W^*$ to have a norm $\| \cdot \|_{W^*}^2$, and transform the atlas coordinates onto the targets to minimize the norm. The space of varifold norms is associated to a reproducing kernel Hilbert space[34,61] (see (8) below) defined by the inner product of the space as $\langle \mu, \nu \rangle_{W^*}$, $\| \mu \|_{W^*}^2 = \langle \mu, \mu \rangle_{W^*}$. To specify the image-varifold norm for $\mu \in W^*$, $\| \cdot \|_{W^*}^2$, it suffices to provide the inner product between Diracs $\langle \delta_x \otimes \delta_f, \delta_{x'} \otimes \delta_{f'} \rangle_{W^*} = K((x,f), (x',f'))$. For any $\mu$ in (1)

then

$$\| \mu \|_{W^*}^2 = \sum_{i,j \in J} \sum_{f,f' \in \mathcal{F}} w_i p_i(f) w_j p_j(f') K((x_i, f), (x_j, f')). \quad (8)$$

Throughout we use the kernel product $K((x,f), (x',f')) = K_1(x, x') K_2(f, f')$ chosen as a Gaussian over physical space $K_1(x, y) = \exp(-\frac{\|x - x'\|^2}{2\sigma^2})$ with $K_2(f, f') = 1$ if $f = f'$, 0 otherwise giving:

$$\| \mu \|_{W^*}^2 = \sum_{j,k \in J} K_1(x_j, x_k) \sum_{f \in \mathcal{F}} w_j p_j(f) w_k p_k(f) \quad (9)$$

for the image-varifold object, as defined in (1) in "Data model and optimization problem: image varifolds and transformations for molecular scales based on varifold norms" section.

## Variational problems

As described in "Data model and optimization problem: image varifolds and transformations for molecular scales based on varifold norms" section, the mapping variational problem constructs a diffeomorphism $\varphi : \mathbb{R}^2 \to \mathbb{R}^2$ and feature laws $(\pi_\ell)_{\ell \in \mathcal{L}}$ on $\mathcal{F}$ to carry the atlas image varifold onto the target, minimizing the varifold normed difference between them, with norm defined as in "Molecular scale varifold norm" section. We follow LDDMM as described in the image case[17], parameterizing the diffeomorphism with the smooth time-varying velocity field, $v_t, t \in [0, 1]$, as $\dot{\varphi}_t = v_t \circ \varphi_t$. This gives the variational problem between the atlas image varifold, $\mu_A$, with indexed locations, $I$, and the target image varifold, $\mu_T$, with indexed locations, $J$:

$$\inf_{\substack{v \in L^2([0,1], V), \\ \pi_\ell, \ell \in \mathcal{L}}} \frac{1}{2} \int_0^1 \| v_t \|_V^2 dt + \| \varphi_1 \cdot \mu_A^\pi - \mu_T \|_{W^*}^2 \quad (10)$$

$$\text{with } \dot{\varphi}_t = v_t \circ \varphi_t, \ \varphi_0 = Id,$$

with $\mu_A^\pi$ depicting the feature-transformed atlas image varifold, as described in "Data model and optimization problem: image varifolds and transformations for molecular scales based on varifold norms" section, as $\sum_{i \in I} \delta_{x_i} \otimes w_i' p_i'$ with $w_i' p_i' = \sum_{\ell \in \mathcal{L}} w_i p_i(\ell) \pi_\ell$, the estimated distribution over target features for indexed location $i$ in the atlas image varifold. The space of smooth time-varying velocity fields giving the flow, $\varphi$, is defined as a reproducing kernel Hilbert space, equipped with norm, $\| \cdot \|_V^2$, ensuring smoothness and invertibility of $\varphi$ as a diffeomorphism[62].

We solve (10) for optimal $\varphi$, $\pi$ through either single or alternating algorithms as described in "Alternating LDDMM and quadratic program algorithm for joint optimization" section, with the opportunity to impose priors on the estimated distributions, $\pi$, appropriate to the specific setting. For instance, we typically impose positivity on all values in estimated distributions: $\pi_\ell(f) \geq 0$ for all $f \in \mathcal{F}$ and $\ell \in \mathcal{L}$. Given prior knowledge of the spatial density of cells or mRNA of a target, we may also specify a range of values for the resulting spatial density to take through constraints on the total measures of features estimated:

$$w^{min} \leq w_i' \leq w^{max}, \ i \in I. \quad (11)$$

This prior can be easily incorporated as a constraint in the setting of the quadratic program used in the alternating algorithmic approach for estimating optimal distributions $\pi$ (see "Alternating LDDMM and quadratic program algorithm for joint optimization" section). In the examples shown here, we typically take $w^{min}$ to be the 5th percentile of values $w_j, j \in J$ of the target image varifold. Additionally, in the setting of mapping tissue-scale atlases to each other, as discussed in "Extension to additional tissue modalities for within and across species comparison" section, we impose the greater constraint of ensuring

constant spatial density of 1, as we use for modeling both atlas and target image varifolds in this case. Finally, in other settings without prior knowledge or wish to impose any on the specific densities prescribed to each indexed location, we can add a general regularization term to (10), such as the Kullback–Liebler divergence between the normalized estimated $\pi$ probability distribution and a uniform distribution across target features:

$$d_l \simeq \sum_{f \in \mathcal{F}} \bar{\pi}_\ell(f) \log\left(\frac{\bar{\pi}_\ell(f)}{(1/|\mathcal{F}|)}\right), \ell \in \mathcal{L} \tag{12}$$

where $|\mathcal{F}|$ gives the number of discrete feature values in the target feature space and the probability distribution, $\bar{\pi}_\ell = \frac{1}{\sum_{f \in \mathcal{F}} \pi_\ell(f)} \pi_\ell$. We use this approach in the examples with BARseq data shown and in the context of a simultaneous rather than alternating optimization algorithm.

## Alternating LDDMM and quadratic program algorithm for joint optimization

For solving the variational problem of (10), optimal $\varphi, \pi$ are jointly estimated through either a single or alternating optimization scheme. In both cases, the template and target can be initially aligned through separate estimation of rigid transformations (translation and rotation) and a single isotropic scaling applied to the template to bring the total area of the template to equal that of the target. In this setting, rigid transformations are estimated by minimizing the varifold normed difference between the rotated and translated template atlas transformed to the target with L-BFGS.

Additionally, in both cases, a gradient-based optimization is performed until convergence or a specified number of iterations. We use L-BFGS optimization combined with a line search using the Wolf condition. In the single scheme, geometric parameters, $v_t$, and feature parameters, $\pi$ are optimized as a joint set of parameters in the gradient-based optimization. In contrast, an alternating scheme separates the estimation of geometric parameters and feature parameters with a gradient-based optimization for the former and the use of quadratic programming in the latter case.

The alternating scheme specifically follows[35], fixing the laws $(\pi_\ell)_{\ell \in \mathcal{L}}$ and optimizing over the control $v(t), t \in [0, 1]$ and integrating it, with the initial condition at the identity element, $\varphi_0 = Id$ (e.g. $\varphi_0(x) = x, \forall x \in \mathbb{R}^2$), to generate the diffeomorphism $\varphi_1$. The diffeomorphism is then fixed and quadratic programming, such as OSQP[63], used to estimate the feature laws. We outline this scheme explicitly below.

**Algorithm 1.**
**Initialize:** $\pi_\ell(f) = \frac{1}{|\mathcal{F}|} f \in \mathcal{F}$
**A: Solve for $v$:**
1. Update and fix $(\pi_\ell)_{\ell \in \mathcal{L}}$.
2. Solve LDDMM, optimizing (10) with respect to vector field $v_t, t \in [0, 1]$.
3. Solve for $\varphi_1$, integrating O.D.E $\varphi_1 = \int_0^1 v_t \circ \varphi_t dt + Id$.
4. Flow $\mu_A^\pi$ according to $\varphi_1$, giving $\varphi_1 \cdot \mu_A^\pi$.
**B: Solve for $(\pi_\ell)_{\ell \in \mathcal{L}}$:**
1. Fix spatial positions in the deformed template, $\varphi(\mathbf{x_i}), i \in I$.
2. Optimize quadratic program (13) with respect to $(\pi_\ell)_{\ell \in \mathcal{L}}$.
**Return to A**

For the atlas, we define the form of the image varifold: $\mu_A = \sum_{i \in I} \delta_{x_i} \otimes w_i p_i$, with, $w_i p_i \in \mathcal{M}(\mathcal{L})$, a measure over the atlas feature values (partitions). The estimated feature distributions are given via $\pi$ as the mixture distribution, $w_i' p_i' = \sum_{\ell \in \mathcal{L}} w_i p_i(\ell) \pi_\ell$. We specify the target over the index set, $J$, as: $\mu_T = \sum_{j \in J} \delta_{x_j} \otimes w_j p_j$ with each $w_j p_j$ and estimated $w_i' p_i' \in \mathcal{M}(\mathcal{F})$. We use this notation in defining the general

form of the quadratic program, with constraint given as described in "Variational problems" section:

$$\inf_{\pi_\ell, \ell \in \mathcal{L}} \| \varphi_1 \cdot \mu_A^\pi - \mu_T \|_{W^*}^2.$$
$$= \inf_{\pi_\ell, \ell \in \mathcal{L}} \sum_{i,i' \in I^2} |D\varphi_1|_{x_i} |D\varphi_1|_{x_{i'}} K_1(\varphi(x_i), \varphi(x_{i'})) \sum_{f \in \mathcal{F}} w_i' p_i'(f) w_{i'}' p_{i'}'(f)$$
$$- 2 \sum_{i \in I, j \in J} |D\varphi_1|_{x_i} K_1(\varphi(x_i), x_j) \sum_{f \in \mathcal{F}} w_j p_j(f) w_i p_i'(f) \tag{13}$$
$$\text{subject to } w^{min} \le w_i' \le w^{max}, i \in I.$$

**Remark 1.** In the algorithm, under the assumption of a constant density with $w_i = 1$ for all $i \in I$, we can approximate the estimated distributions, $w_i' p_i', i \in I$, by the single $\pi_\ell$ at each location (e.g. particle or simplex) for the $\ell$ with the largest mass. Defining the greedy maximizer map $\ell^*(i) = \arg\max_{\ell \in \mathcal{L}} p_i(\ell) \in \mathcal{L}$, we can simplify the computations by using the approximation $w_i' p_i' \simeq p_i(\ell^*(i)) \pi_{\ell^*(i)}$. This is particularly relevant for tissue-scale atlases where most particles or simplices in the image-varifold object are interior to a single atlas region and therefore carry a conditional feature distribution, $p_i = \delta_{\ell_i}$, with all mass attributed to a single feature value, $\ell_i$. For these indexed locations, the above approximation becomes an equality: $w_i' p_i' = \pi_{\ell^*(i)} = \pi_{\ell_i}$.

Both single and alternating optimization schemes were utilized in the examples shown in this work and are available with respective Python git repositories: https://github.com/kstouff4/xIV-LDDMM-Particle v1.0.0[64] and https://github.com/kstouff4/MeshLDDMMQP v1.0.0[65]. The single optimization scheme is provided in the context of point cloud image-varifold implementations, whereas the alternating scheme is provided in the context of mesh-based image-varifold implementations. The quadratic program solver used in the context of the alternating scheme is OSQP as implemented in the Python *qpsolvers* library. Both schemes were developed and tested on a 12 GB TITAN V GPU with CUDA 10.2 and NVIDIA-SMI and Driver 440.33.01, with expected runtime varying by the size of the dataset, but on the order of 2–4 h for most examples highlighted in this work.

## Empirical distribution estimation and cross-replicate statistical comparison

For empirical atlas construction from single or multiple MERFISH sections ("Cross-replicate comparison and atlas construction" section), we used a 50 μm particle rendering (~20k particles) of the CCFv3 section as a scaffold. We assigned each mapped MERFISH cell to its nearest neighbor in the CCFv3 section, giving a distribution over cell types per particle in the CCFv3 section. As described in "Cross-replicate comparison and atlas construction" section, we computed variance in probability distribution within a region by modeling the probability distributions per particle, $p_i$, as vectors in $\mathbb{R}^C$, with the number of cell (sub)types $C = 33$, and $|p_i| = 1$. We computed mean cell type probability distributions per CCFv3 region, denoted $\overline{p_\ell}$ for $\ell \in \mathcal{L}$ by selecting the subset of particles, $I_\ell \subset I$, for each region with at least half of their mass in that region. Under our model of spatial homogeneity per atlas region, we assume the conditional cell type probabilities per location (particle) within each atlas region are consistent. For a given cell (sub)type, $c$, and replicate, $k$, we computed spatial variance within a region, $\ell \in \mathcal{L}$ as:

$$\frac{1}{N_\ell - 1} \sum_{i \in I_\ell} |p_i(c) - \overline{p_\ell}(c)|^2 \tag{14}$$

with $N_\ell$, the number of particles in $I_\ell$. Total variance per region was computed by summing (14) over cell types (Fig. 6i) and per cell type was computed by summing (14) over regions (Fig. 7b).

To assess differences in cell type distribution between replicates (Figs. 6g and 7a), we similarly constructed empirical cell type distributions for each replicate independently, giving for the same indexed particle set, $I$, in the CCFv3 section, three separate image-varifold measures:

$$
\begin{aligned}
\mu^0 &= \sum_{i \in I} w_i^0 \delta_{x_i} \otimes p_i^0 \\
\mu^1 &= \sum_{i \in I} w_i^1 \delta_{x_i} \otimes p_i^1 \\
\mu^2 &= \sum_{i \in I} w_i^2 \delta_{x_i} \otimes p_i^2
\end{aligned}
\tag{15}
$$

with $w_i^k$ for $k = 0, 1, 2$ denoting the cell density (cells per 50 μm² area) for each of the three targets over the CCFv3 space and $p_i^k$ for $k = 0, 1, 2$. We computed spatial variance within each region for each replicate with (14) (Fig. 6j, k). We compared mean cell type probability distributions across replicates for a given cell type, $c$, and in a region, $\ell$, as:

$$
\frac{1}{2} \sum_{k=0}^{2} |\overline{p_\ell^k}(c) - \frac{1}{3}(\overline{p_\ell^0}(c) + \overline{p_\ell^1}(c) + \overline{p_\ell^2}(c))|^2
\tag{16}
$$

with total variance between replicates for a given cell type, $c$ computed by summing (16) across regions (Fig. 7a) and total variance between replicates for a given region computed by summing (16) over cell types (Fig. 6g).

### Reporting summary

Further information on research design is available in the Nature Portfolio Reporting Summary linked to this article.

## Data availability

Serial MERFISH sections from the Allen Institute were produced under the BRAIN Initiative Cell Census Network (BICCN, www.biccn.org, RRID:SCR_015820) and are available at the Brain Image Library (BIL, https://www.brainimagelibrary.org/index.html) under https://doi.org/10.35077/g.610. Cell-segmented MERFISH sections with cell type annotations are available at Zenodo (https://doi.org/10.5281/zenodo.8384018) from Clifton et al.[66]. Serial BARseq sections with cell-level data are available at Mendeley data (https://doi.org/10.17632/8bhhk7c5n9.1). The Waxholm rat brain atlas used in this study is available at https://www.nitrc.org/projects/whs-sd-atlas. The Allen CCFv3 used in this study is available at https://download.alleninstitute.org/informatics-archive/current-release/mouse_ccf/annotation/ccf_2022/. The DevCCF used in this study is available at https://kimlab.io/brain-map/DevCCF/.

## Code availability

Implementations of the algorithms described here can be found at: https://github.com/kstouff4/MeshLDDMMQP (triangulated mesh implementation) v1.0.0[65] and https://github.com/kstouff4/xIV-LDDMM-Particle (point cloud implementation) v1.0.0[64].

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

## Acknowledgements

This work was supported by the National Institutes of Health (1F30AG077736-01 and T32-GM13677 (K.S.); R01EB020062, R01NS102670, U19AG033655, P41-EB031771, and R01MH105660 (M.M.); NIH Brain Initiative Grant U19MH114830 (H.Z.); RF1MH124605 (Y.K.)); the National Science Foundation (NSF) (16-569 NeuroNex contract 1707298 (M.M.); the Computational Anatomy Science Gateway (M.M.) as part of the Extreme Science and Engineering Discovery Environment (XSEDE Towns et al., 2014), which is supported by the NSF grant ACI1548562; NSF CAREER 2047611 (J.F.); NSF 2124230 (L.Y.)); and the Kavli Neuroscience Discovery Institute supported by the Kavli Foundation (M.M.). We acknowledge Benjamin Charlier for his work on refactoring and improving the efficiency of the particle-based implementation of xIV-LDDMM. We acknowledge Fae Kronman for their contribution to the development of the DevCCF atlas used in this manuscript.

## Author contributions

M.M., A.T., and L.Y. developed the mathematical theory behind the manuscript. K.S. and M.M. drafted the manuscript. K.S. and L.Y. generated codes for algorithms described in the manuscript. K.S. created all the figures in the manuscript. M.K., L.N. and H.Z. generated serial MERFISH data. M.A. and J.F. annotated cell types for cell-segmented MERFISH data. Y.K. created the DevCCF atlas analyzed here with the CCFv3. X.C. and M.R. generated and cell typed the BARseq data analyzed in this manuscript, and specified particular cell type markers for

accuracy measurement. All authors contributed to the editing of the final manuscript.

## Competing interests

Under a license agreement between AnatomyWorks and the Johns Hopkins University, Dr. Miller and the University are entitled to royalty distributions related to technology described in the study discussed in this. Dr. Miller is a founder of and holds equity in AntomyWorks. This arrangement has been reviewed and approved by the Johns Hopkins University in accordance with its conflict of interest policies. The remaining authors declare no conflicts of interest.
