## [Peer Review File · Nature Communications]

Cross-Modality Mapping using Image Varifolds to Align Tissue-Scale Atlases to Molecular-Scale Measures with Application to 2D Brain SectionsREVIEWER COMMENTS

Reviewer #1 (Remarks to the Author):

The manuscript entitled "A Universal Method for Crossing Molecular and Atlas Modalities using Simplex-Based Image Varifolds and Quadratic Programming" by Stouffer et al. proposes an algorithm designed to tackle the challenge of aligning and integrating high dimensional spatial molecular data (showcased here only with MERFISH spatial transcriptomics) with tissue-scale atlases. This approach, combining simplex-based image varifolds and quadratic programming, promises to facilitate the integration of diverse molecular and cellular datasets into a unified coordinate system. This initiative addresses an important need in the field of spatial biology and, if implemented effectively and user-friendly, could be valuable to our understanding of complex biological systems. However, the current manuscript has numerous significant concerns that need to be addressed and I list them below:

1) I find the narrative style overly heavy and in-depth in a manner that might be more suited to a specialized mathematical or computational journal than a more generalist interdisciplinary one like Nature Communications. While the complexity of the subject requires technical language, the current approach could be an obstacle for a wider audience, including both biologists and other professionals who are not deeply trained in mathematics and computational science. I recommend a significant revision of the narrative style, with an emphasis on making it more intelligible for a broader readership. One possible approach would be to shorten the mathematical details in the main text and move these into a supplementary section or detailed methods section. The main text should focus more on intuitively explaining the method's core concepts, its application, and its implications in a manner intelligible to the broader audience.

2) A critical point of concern in the manuscript is the lack of comparison against other existing methods. Without this benchmarking, it is impossible gauge the performance, efficiency, and overall effectiveness of the proposed method. I would strongly recommend that the authors conduct a comparative analysis against other existing methods (e.g. landmark based methods with various matching algorithms, similarity based methods). Comparison against more recent methods e.g. GPSA (Jones, Engelhardt et al., 2022, bioRxiv) or SLAT (Xia, Gao et al., 2023, bioRxiv) would also be valuable. Such a comparison would provide a concrete basis for evaluating the method's performance and its relative advantages or shortcomings. Furthermore, it would give the readers a clearer understanding of where this new method stands in the broader context of the field.

3) As a continuation of the first point, the manuscript lacks clarity on the actual usage and implementation of the proposed method. For the audience to truly appreciate the power and potential of this algorithm, it is essential to explain its functioning in simpler, more tangible terms. This includes elucidating the inputs required, user parameters, expected outputs, and how these outputs can be utilized in downstream analyses.

4) Related to previous point, demonstrations of the method using real-world data and showcases of how it allows to answer real-world biological questions would be needed to provide insight into its performance and potential.

5) The quality and interpretability of the figures included in the manuscript currently fall short of expectations. While the authors present atlas images of the brain, MERFISH images, and mapped results, these illustrations do not offer a clear understanding of the algorithm's performance. The simple presentation of the 'before' and 'after' scenario without sufficient quantifiable metrics leaves room for interpretation and does not convincingly demonstrate the method's efficiency. I strongly encourage the authors to enrich their figures with quantitative metrics that can underscore the algorithm's effectiveness. Additionally, showcasing diffeomorphism mappings without reference to ideal and worst-case scenarios is meaningless without illustrating how these mappings would look under optimal and worst conditions. Authors need to provide a more comprehensive understanding of

their method's performance.

7) The current manuscript showcases the method primarily in the context of brain atlas, which is a highly structured and regionalized organ that allow manual and simple approaches to align to brain atlases with acceptable accuracy. While use of brain offers some insights, it limits our understanding of the method's versatility, power and adaptability. I recommend that the authors expand their research to cover a broader set of scenarios. Specifically, it would be enlightening to see how the method performs when applied to other organs with less stereotypical structure. Additionally, the method's performance when trying to align suboptimal matching data to an atlas would provide an important benchmark of its robustness and adaptability. Furthermore, the manuscript should address how the method deals with technical distortions in the spatial transcriptomic/molecular data, such as physical distortions, cracks, missing fields of view, locally distorted dynamic range or sensitivity of molecular data etc. It would also be pertinent to discuss how the method can handle situations where locally the data do not match, for example, when mapping a diseased or locally injured or otherwise impacted section onto an atlas.

8) The manuscript does not sufficiently discuss the computational requirements and demands associated with the implementation of the proposed method. Such information is crucial for readers and potential users to evaluate the feasibility of employing this method in their research, considering their available computational resources.

9) While the manuscript focuses on mapping spatial transcriptomic data to brain atlases, I would be curious to understand how adaptable this method is for aligning two different spatial or molecular modalities, especially when the feature sets are disjoint. This could represent a valuable new application of the method. Authors could discuss, and (if feasible), demonstrate the feasibility of applying their method for cross-modality mapping, such as aligning immunohistochemistry staining data with spatial omics data, or other spatial measurements. If feasible, they should explain any necessary modifications or considerations to the method for such applications.

Reviewer #2 (Remarks to the Author):

This paper presents a method to build spatial correspondence between the molecular-scale transcriptomics and tissue-scale atlases (references) based on an existing image-varifold large deformation diffeomorphic metric mapping (LDDMM) algorithm. In addition, the authors propose a joint framework to estimate the correspondences across different data modalities simultaneously with the associated density function of cell distributions. While the idea of studying molecular- and cellular-scale populations in a common space under the atlas-based deformation model is new and interesting, the current manuscript requires significant improvement in terms of structure organization, methodological development, and experimental evaluations. My major concerns and comments are listed as follows:

- The manuscript needs better organization. Sections 2 and 3 should be categorized as background information rather than results. Section 4 appears to present the proposed joint estimation method. It is essential for the authors to clearly distinguish the previous studies from the novelty and contributions of the proposed work. This will help readers understand the context and importance of the research in relation to existing literature.

- The authors have not quantitatively validated their experimental results. No thorough statistical analysis of the estimated results (e.g., estimated mRNA densities, the diffeomorphic transformations, and the predicted oncology using estimated transformations.) is provided. Additionally, no comparison is made between the proposed method and existing approaches (e.g., the original LDDMM for mesh data and other related works), making it difficult to assess the efficacy of the experimental results.

- The title "A Universal Method" may not be appropriate since the proposed method (according to section 4) is based on derived meshes from datasets with different modalities. The authors do not seem to have made significant methodological contributions to make the method universally applicable to different modalities.

- The paper lacks sufficient motivation for jointly estimating atlas-based transformations and the latent density functions of molecular features. Previous works in the literature (e.g., [1,2,3]) have demonstrated that combining atlas-based transformations as geometric features with image features within or across multiple modalities improves specific tasks such as image classification and segmentation. However, it is unclear why the jointly estimated transformations and density distributions of molecular features are important or necessary. The authors at least should compare with a simple disjoint approach.

[1] Wang and Zhang, Geo-SIC: Learning Deformable Geometric Shapes in Deep Image Classifiers, 2022.

[2] Ding et. al., Cross-Modality Multi-Atlas Segmentation via Deep Registration and Label Fusion, 2022.

[3] Ding and Niethammer, Aladdin: Joint Atlas Building and Diffeomorphic Registration Learning with Pairwise Alignment, 2022.

- Section 2.3 seems disconnected from the proposed method. It is unclear how it relates to the joint estimation method. Is dimensionality reduction particularly crucial for the core algorithm, or is it a strategy employed solely for the experimental results in this paper?

- In Figure 4, the estimated cell densities appear to be very noisy compared to the atlas. It would be helpful to clarify whether this noise is due to the discontinuity of the target subject. If so, the assumptions of diffeomorphic (smooth) transformations between the atlas and individual subjects need to be justified.

- The manuscript does not clearly state the objective functions for estimating the latent conditional feature laws. It would be beneficial to provide a clearer explanation in the current version.

- The mathematical notations in the manuscript are either poorly defined, or used before clear definitions are provided. For example,

L206: the definition of 'varifolds' needs to be defined earlier.

L221-223: please make sure to define all math notations before using them.

L265-266: what are I and J ? Please clarify.

L1111: the definition of id is missing.

L1126: the steps of using quadratic programming in Eq. (9) need to be clarified.

Reviewer #3 (Remarks to the Author):

The authors have developed a novel computational method aimed at mapping tissue atlases to gene-based and cell-based MERFISH datasets. Overall, the problem addressed in the manuscript is

interesting, and the mathematical foundation of the method appears solid. Yet, the current state of the manuscript is quite disorganized and conveys the feeling to have been written in a rushed way. Therefore, we feel that a major revision is necessary before the manuscript can be reconsidered for publication in Nature Communications. Our recommendations are as follows:

1. Absence of a method overview: After introducing the problem that the method aims to solve in the introduction, the authors delve directly into the mathematical equations of the computational tool in Sections 2.1 and 2.2. We recommend providing an overview of the computational tool at the beginning of Section 2, accompanied by a schematic figure, to improve the overall comprehension of the paper for readers.
2. Unclear value of the computational tool: While the motivation behind the work is compelling—aligning different functional modalities at various scales—the authors do not adequately demonstrate the significance of the alignment after applying their method. It would be beneficial to discuss any new biological insights gained and to explain how the method advances our understanding of the data.
3. Lack of quantitative validation: The computational tool has two distinct applications: gene prediction (Figure 3) and cell type distribution prediction (Figure 4). After making these predictions, the authors should compare their method with other existing computational tools and provide a more in-depth analysis of the biological insights derived from the predictions.
4. Inclusion of additional datasets: The paper focuses solely on mouse brain datasets, which, while understandable given that these datasets are the most readily accessible, limits the scope of the method's applicability. It would be advantageous for the authors to apply their tool to other datasets to evaluate its effectiveness and generalizability.
5. Unorganized figures: The manuscript includes ten main figures; however, some of them contain insufficient information, such as Figure 1. We recommend combining some of the figures to create a more cohesive presentation of the results.

In addition, we have some minor suggestions on the manuscript:

1. The math in Section 2.1 and 2.2 shall be moved to the Method.
2. It would be helpful to name the computational tool, so that it would be easier for researchers to refer to.

Dear Reviewers,

Thank you for your careful consideration of the manuscript and the thoughtful comments and suggestions provided. Please find below all of your comments reproduced in black boldface font. Our responses are colored in blue with copied edits from the revised manuscript shown in red.

1 Reviewer 1

The manuscript entitled “A Universal Method for Crossing Molecular and Atlas Modalities using Simplex-Based Image Varifolds and Quadratic Programming” by Stouffer et al. proposes an algorithm designed to tackle the challenge of aligning and integrating high dimensional spatial molecular data (showcased here only with MERFISH spatial transcriptomics) with tissue-scale atlases. This approach, combining simplex-based image varifolds and quadratic programming, promises to facilitate the integration of diverse molecular and cellular datasets into a unified coordinate system. This initiative addresses an important need in the field of spatial biology and, if implemented effectively and user-friendly, could be valuable to our understanding of complex biological systems. However, the current manuscript has numerous significant concerns that need to be addressed and I list them below:

1. I find the narrative style overly heavy and in-depth in a manner that might be more suited to a specialized mathematical or computational journal than a more generalist interdisciplinary one like Nature Communications. While the complexity of the subject requires technical language, the current approach could be an obstacle for a wider audience, including both biologists and other professionals who are not deeply trained in mathematics and computational science. I recommend a significant revision of the narrative style, with an emphasis on making it more intelligible for a broader readership. One possible approach would be to shorten the mathematical details in the main text and move these into a supplementary section or detailed methods section. The main text should focus more on intuitively explaining the method’s core concepts, its application, and its implications in a manner intelligible to the broader audience.

We thank the reviewer for their suggestion. As shown in the revised manuscript, we have made significant changes to the narrative style in the Introduction and Results sections. As suggested, we have moved much of the mathematical detail that was initially contained in the first two Results sections to the newly organized Methods sections. Regarding a more “intuitive explanation” of the method’s core concepts, we have also included a broader introduction to the method through the use of a schematic

diagram, as suggested by reviewer 3, and accompanying description of this diagram. Some of the relevant text in the first Results section is copied here:

Within this established framework, we align image varifolds capturing tissue-scale atlases to those capturing molecular-scale data via estimation of two types of correspondence: one between physical coordinates and the other between feature spaces. These correspondences act independently and in parallel on the physical and feature measure components of the image varifold object, as depicted in the schematic at the bottom of Figure 1, where the gray boxes denote these separate correspondences.

To define a similarity metric between deformed atlas and target, we also need to carry the feature component of tissue-scale atlases to the feature space of the molecular target. For this, we associate to each feature value in the atlas ontology, a distribution over target feature values, capturing the assumption of spatial homogeneity we make within each atlas region. We denote the set of distributions for each atlas feature, $\ell \in \mathcal{L}$ over target feature space as $(\pi_\ell)_{\ell \in \mathcal{L}}$ with each $\pi_\ell \in \mathcal{M}(\mathcal{F})$. Measures over the target feature space are generated for each discrete ‘‘Dirac’’ measure over physical space, indexed by $i \in I$, in the atlas as the mixture distribution:

$$\nu'_i = \sum_{\ell \in \mathcal{L}} \nu_i(\ell) \pi_\ell, \quad (1)$$

with $\nu'_i \in \mathcal{M}(\mathcal{F})$, $i \in I$, the corresponding set of conditional measures over the target feature space, as shown in column 3 of Figure 1. Importantly, we note that this set of distributions is not normalized, capturing both the spatial density of points (e.g. mRNA or cells) in the target as well as the conditional distribution over the feature space (e.g. gene type or cell type).

Finally, the estimation and application of both geometric transformations and latent feature distributions, φ, π , to the atlas image varifold take it to both physical and feature space of the target (see column 4, Figure 1). In the image setting, the similarity function used is often a norm on functions, and solving the problem of minimization of the norm in the space of diffeomorphisms gives the metric theory of LDDMM for generating geodesic matching between exemplar anatomies [1, 2]. Here, the similarity function is a norm on image varifolds (see Section 4.2), capturing proximity in both physical and feature spaces of the target...

Regarding applications of the method, we defer the reviewer to our responses to reviewer 3’s second comment where we explicate how we have restructured our Results sections to cover three particular applications of our method: cross-replicate comparison and atlas construction; within and across species atlas comparison; and image / molecular data segmentation. We showcase each of these applications using different modalities for our

target object to which we estimate mappings of a tissue scale atlas. These are respectively gene-based and cell-based spatial transcriptomics datasets, alternative mouse and rat brain atlases, and a DAPI-stained digital pathology image. The text within each of these sections has largely been added as new additions to this version of the manuscript and is documented heavily in our response to reviewer 3's second comment.

Finally, with regard to implications of our method, we have revised the introduction to highlight how our method differs from currently available methods in the spaces of image mapping and transcriptomics mapping methods. We expand on these changes in our response to the reviewer's second comment below, but specifically underscore here two of the key benefits (and differences) we see in our method over other existing methods. The first is in its ability to model equivalently in an image varifold based framework both regularly sampled gridlike objects (e.g. classical images and spot-based technologies) as well as irregularly sampled and very high dimensional objects (e.g. image-based spatial transcriptomics datasets). The second is in its use of a homogeneity assumption as a prior on estimated feature laws for defining a similarity metric between objects of fundamentally distinct modalities (e.g. tissue scale atlases with region label as feature versus molecular scale gene or cell-based data with gene type or cell type as feature). This stands in contrast to methods requiring an intersection of feature sets across technologies or extensive training datasets from which to learn informative features to transform one type of data to another. We highlight these benefits in our Introduction:

Here, we build on both of these lineages in presenting a model equally equipped at representing tissue-scale atlases and molecular-scale data, and an associated cross-modality registration mechanism that addresses both challenges of crossing scales and functional modalities by estimating simultaneously geometric and functional transformations to align each dataset to the other. Specifically, we harness the generalizability of image varifolds, which have emerged in molecular CA [3], for simultaneously modeling molecular and tissue-scale data with both irregular and gridlike sampling schemes in a common framework (addressed in SLAT, for instance, by a graph-based representation in the setting of molecular and cellular data [4]). A subproblem covered by the image-varifold theory outlined in [5] is the mapping of molecular scale data to atlas coordinate systems. As specified there, we estimate minimal energy diffeomorphic transformations through large deformation diffeomorphic metric mapping (LDDMM) [6] with the action of diffeomorphisms on image varifolds, defined in a manner consistent with how diffeomorphisms have transformed classical images from one coordinate space to another [3]. In tandem, we estimate a latent distribution over the molecular functional space for each atlas partition, without need for the large datasets often used in deep learning approaches,

but by instead relying on an assumption of spatial homogeneity in distribution across each atlas partition. We use this latent distribution to transform the functional space of each atlas (i.e., its ontology) to the molecular functional space.

We also underscore these implications again in the Discussion:

The image-varifold framework used here is general enough to model classical tissue-scale imaging data (as demonstrated in our atlas-to-atlas mappings), digital pathology (as demonstrated in our example with the DAPI-stained image (Section 2.6), and emerging transcriptional data from both image-based and spot-resolution technologies that might generate data complete or partial tissue captures, irregularly sampled across space...Indeed, we finally emphasize that central to the model posed here is the underlying assumption that each compartment has a homogeneous distribution over molecular features that is stationary with respect to space. We make this assumption to govern estimation of latent feature distributions for each atlas compartment to take the atlas into both physical and feature space of the target. This is in contrast to learning-based approaches that estimate features independent of such assumptions to map atlas to target across feature (e.g. imaging) modalities, but often require extensive training datasets and computational power to learn such features [7, 8].

2. **A critical point of concern in the manuscript is the lack of comparison against other existing methods. Without this benchmarking, it is impossible gauge the performance, efficiency, and overall effectiveness of the proposed method. I would strongly recommend that the authors conduct a comparative analysis against other existing methods (e.g. landmark based methods with various matching algorithms, similarity based methods). Comparison against more recent methods e.g. GPSA (Jones, Engelhardt et al., 2022, bioRxiv) or SLAT (Xia, Gao et al., 2023, bioRxiv) would also be valuable. Such a comparison would provide a concrete basis for evaluating the method's performance and its relative advantages or shortcomings. Furthermore, it would give the readers a clearer understanding of where this new method stands in the broader context of the field.**

We thank the reviewer for their suggestion about comparison to other mapping techniques and evaluative measures as relates to their point below. To address this point, we have first added a summary paragraph to the manuscript in the introduction comparing our method to other mapping techniques that have been proposed for handling molecular data such as GPSA and SLAT according to the reviewer's suggestion. We emphasize that the novelty introduced in our method is in its aim to find alignments

specifically between datasets defined at different scales and over different feature spaces, where data might be irregularly sampled over space, as in emerging image-based spatial transcriptomics technologies. Consequently, our method relates to GPSA in its estimation of geometric and feature based “transformations” as GPSA models both types of transformations through gaussian processes. However, GPSA appears to be aimed more at the application of atlas construction in bringing together replicates across a single technology, typically, into a single coordinate space, and using a template comprised of one of these replicates. Furthermore, GPSA is only showcased on spot-based technologies which exist in a gridlike fashion. In contrast, SLAT extends to finding maps using a graph-based approach for representing data and therefore exhibits success in aligning image-based spatial transcriptomics datasets, as we address in our paper, and that may be defined at slightly different scales (e.g. cellular and molecular scales). However, SLAT does not offer ways of crossing modalities without intersecting feature sets. Hence, we do not see either of these methods directly applying to the case of irregularly sampled data fundamentally of different modalities (e.g. tissue scale atlas and MERFISH gene-based or cell-based dataset) that we exhibit most frequently in this manuscript. The added introductory paragraph to address this comparison is as follows:

Consequently, an independent class of methods has been developing for aligning spatially resolved transcriptional datasets at these molecular scales. With influence from image-based methods, some of these including GPSA [9] and PASTE [10] focus exclusively on spot-based data in which gene expression is measured in a neighborhood of each “spot” for a regular array of spots, analogous to a voxel grid in an image. In contrast, image-based technologies such as STARmap, BARseq, and MERFISH, which take point measures of individual mRNA molecules or cells, cover measurements irregularly sampled over space according to tissue architecture and dynamics. Furthermore, natural fluctuation in gene expression over time and space coupled to the dynamics of each spatial transcriptomics technology leads each tissue section, at the molecular (1-100 micron) scale, to have a varying number of such particles with no natural ordering of particles consistently apparent between sections. Consequently, landmark-based methods [11] that assume direct permutation correspondence between particles are not applicable. Methods aimed at generalizing to allow alignment within and across these additional technologies are typically rooted in different data representations, such as graphs, in the case of SLAT [4] that aims to find correspondence between cells or groups of cells in atlas and target. Additionally, while many methods have showcased success at aligning different replicates within a single technology for both human and mouse samples, cross-technology alignment has typically relied on a nonempty intersection of feature sets across these technologies [4, 12], limiting extension of these methods to the type of cross-modality and cross-scale mapping required

for integrating tissue-scale atlases and molecular scale data...Specifically, we harness the generalizability of image varifolds, which have emerged in molecular CA [3], for simultaneously modeling molecular and tissue-scale data with both irregular and gridlike sampling schemes in a common framework (addressed in SLAT, for instance, by a graph-based representation in the setting of molecular and cellular data [4]).

3. **As a continuation of the first point, the manuscript lacks clarity on the actual usage and implementation of the proposed method. For the audience to truly appreciate the power and potential of this algorithm, it is essential to explain its functioning in simpler, more tangible terms. This includes elucidating the inputs required, user parameters, expected outputs, and how these outputs can be utilized in downstream analyses.**

We thank the reviewer for their comment. As indicated in our response to the reviewer's first comment, we have revised our introduction and results sections to highlight applications of our method that we believe will also clarify its usage in broad terms. We specifically highlight the application of atlas construction and cross-replicate comparison, as described in Section 2.4, where we further indicate how both the inverse of the diffeomorphic mappings, φ , as well as the estimated feature laws, π , can be used/further analyzed to determine the variance across replicates as well as the spatial variance in empirical distribution across atlas regions. The relevant text added in this section includes:

We illustrate both of these applications in considering cell-typed MERFISH sections taken from three separate mice at approximately the same coronal level, as shown in Figure 7. ARA section $Z = 890$ out of 1320 was mapped separately to each of these targets with both geometric transformations and latent feature distributions estimated in each case. As in the classical imaging setting with LDDMM, comparison of the estimated diffeomorphisms taking the ARA section to each respective target section offers one metric of similarity between both atlas and target and across targets [1, 2]. Here, comparison of the jointly estimated latent feature distributions offers a second metric of similarity...Regarding atlas construction, the sample mean cell type distribution across those latent distributions estimated jointly with geometric transformations, as shown in Supplementary Figure S4 gives one potential construction of a cell type atlas over space. A finer-grained atlas can alternatively be achieved in this setting by pulling back each target MERFISH section onto the same ARA section via the inverse estimated diffeomorphism. Particles (analogous to foreground pixels) in the ARA section serve as a scaffold for resampling mapped cells to with either a nearest neighbor assignment or dispersion of cell mass according to kernel choice.

Regarding the details of implementation, we have first explicated where choices in implementation can and were made such as in the form image varifold objects were implemented in (e.g. meshes or point clouds) and in the specific algorithmic scheme that can be used for optimization (e.g. single or alternating scheme). These explications are included in the Results section 2.1, with corresponding supplementary figure indicating the equivalence in representation and in mappings estimated via the use of meshes or point clouds, and Methods sections 4.1 and 4.4 as follows:

Results:

xIV-LDDMM jointly estimates optimal φ and π to minimize this normed difference in the space of diffeomorphisms through either simultaneous or alternating optimization algorithms using LDDMM (see Section 4.4), with additional regularization imposed in both settings on the estimated π to ensure, for instance, positive values (see Section 4.3 for explicit variational problems). Note that throughout we highlight mapping examples using both mesh-based [5] and particle-based (point cloud) [3] implementations of image varifolds, with detailed construction, similarities, and differences of each covered in Section 4.1. Supplementary Figure S1 shows equivalence in these two implementations and in the corresponding mappings estimated in each case for the CCFv3 and MERFISH sections shown in Figure 1.

The corresponding supplementary figure caption reads:

Comparable renderings at 50 μm resolution of MERFISH cell-independent transcriptomics section and corresponding geometric mapping of Allen CCFv3 section $Z = 385$ out of 1320 to transcriptomics section. Top shows rendering of initial section as point cloud with 63k particles, with estimated determinant jacobian of diffeomorphic mapping of Allen CCFv3 section to transcriptomics. Bottom shows rendering as mesh with 15k simplices and estimated determinant jacobian of diffeomorphic mapping of Allen CCFV3 section to transcriptomics with similar regions of expansion and contraction.

Methods:

As introduced in Section 2.1, we can represent each image-varifold object as a point cloud (particles) or a triangulated mesh. In the first case, data is modeled as a collection of particles, each with a center $x_i \in \mathbb{R}^2$, and a measure over its feature space, $\nu_i \in \mathcal{M}(\mathcal{F})$. Particles may be individual mRNA detections, cell centers, or image pixels at the highest resolution, with thresholding or exclusion of particular image values done in the latter case to extract foreground pixels only. Sets of particles can be “down-sampled” to a smaller set of discrete particles by capturing neighboring particle distributions together into one particle, as used in the implementation of the DAPI image (Section 2.6). We refer to the “resolution” of the rendering to indicate the span of neighborhood each individual particle

captures with the coarser the resolution corresponding to larger and larger neighborhoods of measurements encapsulated in a single particle...

For solving the variational problem of (10), optimal φ, π are jointly estimated through either a single or alternating optimization scheme....Additionally, in both cases, a gradient based optimization is performed until convergence or a specified number of iterations. We use L-BFGS optimization combined with a line search using the Wolf condition. In the single scheme, geometric parameters, v_t , and feature parameters, π are optimized as a joint set of parameters in the gradient-based optimization. In contrast, an alternating scheme separates estimation of geometric parameters and feature parameters with a gradient-based optimization for the former and the use of quadratic programming in the latter case.

Finally, we emphasize that the computational runtime and memory requirements will vary based on size of dataset including number of data points as well as feature space that the user wishes to model as an image varifold. For a benchmark, we have included details about how the examples showcased here were run in the Methods section 4.4 and on what type of machine, as also highlighted in our response to the reviewer's related comment (number 7):

Both single and alternating optimization schemes were utilized in the examples shown in this work and are available with respective python git repositories: <https://github.com/kstouff4/xIV-LDDMM-Particle> and <https://github.com/kstouff4/MeshLDDMMQP>. The single optimization scheme is provided in the context of point cloud image varifold implementations, whereas the alternating scheme is provided in the context of mesh-based image varifold implementations. The quadratic program solver used in the context of the alternating scheme is OSQP as implemented in the python *qpsolvers* library. Both schemes were developed and tested on a 12 GB TITAN V GPU with CUDA 10.2 and NVIDIA-SMI and Driver 440.33.01, with expected runtime varying by size of dataset, but on the order of 2-4 hours for most examples highlighted in this work.

4. **Related to previous point, demonstrations of the method using real-world data and showcases of how it allows to answer real-world biological questions would be needed to provide insight into its performance and potential.**

We thank the reviewer for their continued comment related to the previous point. Following on our above response, we additionally underscore that we have included additional examples from varying datasets in the revised manuscript as further demonstrations of our method across settings where these datasets might arise. In particular, to the reviewer's sixth

comment, we have included a “suboptimal” dataset of a partial coronal brain section of BARseq as a demonstration of its performance on what might be considered “real-world” data. We have also further discussed how our method can handle “real-world” data that might contain distortions or missing tissue with the alteration of the image varifold normed difference in the cost function being multiplied by spatially varying weights indicative of regions in which to prioritize achieving alignment versus not. These additions are detailed in our response to the reviewer’s 6th comment below with the particular relevant part copied here:

The image varifold representation, particularly in its semi-discrete form as used here, is amenable to representing data with regions of missing or disrupted capture as an extension of the irregularity in sampling that is assumed in the general case. To estimate mappings in these cases, variation in both the forms and sizes of kernels governing the varifold norm (see Section 4.2) can control the granularity to which matches between atlas and target should be evaluated, with coarser scales more appropriate to noisier tissue sections with higher numbers of artifacts. Additionally, the varifold normed difference in the cost function, as presented in (10) in Section 4.3 can be appended with spatially varying weights, as used in settings of mapping digital pathology to MRI [13] to prioritize matching amongst certain “intact” regions over others. We utilize this strategy in Figure 4 to align a partial BARseq section to a corresponding full CCFv3 section through depiction of the initial geometric offset (left) versus the transformed partial BARseq section to the CCFv3 coordinates (right), with appropriate scale, rotation, and diffeomorphic transformations estimated to position the partial capture within the scope of the correct CCFv3 hemisphere.

Additionally, regarding real-world biological questions, we have focused this revised manuscript, as stated in our other responses, around three potential applications of our method: cross-replicate comparison and atlas construction; within and across species atlas comparison; and image segmentation. For the purpose of surveying each of these applications, we provide a single example in most cases as a template for other groups to work from and to tailor to their particular questions of interest that might rest in a more narrow biological field or center around a more explicit question. We provide examples of these questions throughout our results sections, such as the question of where replicates are most varying (across them) in cell type distribution, what regions of different atlases (e.g. Allen reference atlas versus Kim atlas) are most heterogenous with respect to cell type distribution, and where different species, such as a mouse versus a rat, differ most in terms of brain structure, as classified by traditional atlases (e.g. Allen atlas versus Waxholm rat atlas). The relevant text covering these questions is copied below:

Section 2.4: Here, comparison of the jointly estimated latent feature distributions offers a second metric of similarity. The bottom left of Figure 7 highlights the sample variance for each atlas region across the three estimated cell type probability distributions, modeled as unit vectors, $x \in R^C$, for $C = 33$ cell types. Sample variance is computed as $\frac{1}{N-1} \sum_{n=1}^N \|x_n - \bar{x}\|_2^2$ in R^C , with $N = 3$ and \bar{x} , the sample mean, as illustrated in tabular form for each atlas region in Supplementary Figure S4. Black dotted arrows highlight the region of the medial geniculate nucleus in each MERFISH sample, which exhibits amongst the highest variance across the replicates. This region is magnified in the right column, illustrating this variance in relative distribution of excitatory neurons, astrocytes, endothelial cells, and ependymal cells.

Section 2.5: Atlas comparison can be conducted both independently and within the context of particular molecular targets as well....Just as entropy in estimated distribution can be used to compare atlas to atlas, directly, the right column exhibits the entropy of the distributions over cell types estimated for each region in each atlas, giving an indication to which regions hold more heterogenous cell type distributions. Here, the hippocampus is more finely partitioned in the Kim atlas, which yields lower entropy distributions over cell types than in those estimated for the Allen atlas

Section 2.5: Atlas ontologies can be mapped not just within species but also across them, where both geometric transformations and estimated ontology distributions, together reflect metrics of comparison between the two.

5. **The quality and interpretability of the figures included in the manuscript currently fall short of expectations. While the authors present atlas images of the brain, MERFISH images, and mapped results, these illustrations do not offer a clear understanding of the algorithm's performance. The simple presentation of the 'before' and 'after' scenario without sufficient quantifiable metrics leaves room for interpretation and does not convincingly demonstrate the method's efficiency. I strongly encourage the authors to enrich their figures with quantitative metrics that can underscore the algorithm's effectiveness. Additionally, showcasing diffeomorphism mappings without reference to ideal and worst-case scenarios is meaningless without illustrating how these mappings would look under optimal and worst conditions. Authors need to provide a more comprehensive understanding of their method's performance.**

We thank the reviewer for their comment. As discussed in our responses to many of the comments by this reviewer and the others, we have heavily

revised the manuscript so as to provide more comprehensive, comparative, and quantitative examples of our method for enabling clearer use by others and clearer understanding of the differences in our method versus some of the existing ones. These differences include the representation (image varifold versus regularized grid) we use for handling the diversely sampled data of spatial transcriptomics technologies as well as the assumption we make of homogeneity to estimate latent feature laws for effectively mapping data across modalities (and across scale). The specific additions made relevant to this comment are three-fold. First, we have exhibited successful alignment of atlas to target in the setting of additional datasets (e.g. BARseq, DAPI-stained images) with the added complexity of being potentially partial or incomplete captures of the area covered by an atlas (e.g. hemisphere versus whole brain section). As discussed in our response to the reviewer’s fourth comment on “real-world” data, we believe this example of mapping whole brain atlas to half brain spatial transcriptomics gives a benchmark as to the type of worst-case scenario data that might be addressed by our method. We provide additional suggestions on how to deal with further distortions and missing tissue in Results Section 2.2:

Additionally, the varifold normed difference in the cost function, as presented in (10) in Section 4.3 can be appended with spatially varying weights, as used in settings of mapping digital pathology to MRI [13] to prioritize matching amongst certain “intact” regions over others. We utilize this strategy in Figure 4 to align a partial BARseq section to a corresponding full CCFv3 section through depiction of the initial geometric offset (left) versus the transformed partial BARseq section to the CCFv3 coordinates (right), with appropriate scale, rotation, and diffeomorphic transformations estimated to position the partial capture within the scope of the correct CCFv3 hemisphere.

The second addition we include in this revision is a comparison to a simple disjoint approach, as suggested by Reviewer in their comments 2 and 4, which we chose to be an image-based version of LDDMM. We specifically highlight the difference an image-based representation has to an image-varifold in the inherent loss in resolution associated with discretizing irregularly spaced cells/genes into a grid. We also highlight the difference in alignment in this disjoint approach focusing on achieving a foreground-background match compared to the joint approach we present here that aims to align regions in atlas and target that appear homogeneous with respect to space in distribution over their respective features. We refer the reviewer to our responses to the second reviewer’s second and fourth comments for the particular areas of text added in Section 2.3 related to this comparison.

Finally, in addition to the changes discussed above, we include quantitative metrics of accuracy, as highlighted in our response to the second comment of Reviewer 2 that are based on mapping cell types (markers) extracted from sections of BARseq data to corresponding sections of the Allen Reference Atlas. We measure the distance of each of these cell markers to their corresponding atlas region (e.g. CA1 pyramidal cells to CA1 in the atlas section) by associating to each cell marker, its distance from the nearest indexed location in the corresponding region in the allen atlas. We compute this distance before and after alignment (pulling back the BARseq data onto the atlas via the inverse of the diffeomorphism). We observe mean and median distances in the regions analyzed typically on the order of $10 - 30\mu m$, giving a guideline of the expected performance of the algorithm in aligning these different datasets. We refer the reviewer to our response to the second reviewer's second comment below where we have copied the relevant additions to Section 2.3 along with the figure caption for the added Figure 5.

- 6. The current manuscript showcases the method primarily in the context of brain atlas, which is a highly structured and regionalized organ that allow manual and simple approaches to align to brain atlases with acceptable accuracy. While use of brain offers some insights, it limits our understanding of the method's versatility, power and adaptability. I recommend that the authors expand their research to cover a broader set of scenarios. Specifically, it would be enlightening to see how the method performs when applied to other organs with less stereotypical structure. Additionally, the method's performance when trying to align suboptimal matching data to an atlas would provide an important benchmark of its robustness and adaptability. Furthermore, the manuscript should address how the method deals with technical distortions in the spatial transcriptomic/molecular data, such as physical distortions, cracks, missing fields of view, locally distorted dynamic range or sensitivity of molecular data etc. It would also be pertinent to discuss how the method can handle situations where locally the data do not match, for example, when mapping a diseased or locally injured or otherwise impacted section onto an atlas.**

We thank the reviewer for their comment. We appreciate the suggestion to showcase the method on data drawn from tissue in other organs with different levels of organization. For consistency internal to the manuscript as well as in the context of the domains of the authors, we have expanded the range of examples showcased in the revised manuscript to highlight different technologies, but all from tissue drawn from the brain. We have noted the extension and testing of our methods on other tissue regions,

instead, as an area for future work in the Discussion:

...we are continuing to investigate the use of our method to map data from additional types of technologies, such as the spot-resolution SlideSeq [14], and in additional biological settings, where we might see further disruptions of tissue architecture (e.g. with tumors) and in tissue types with potentially varying levels of organization and therefore varying levels to which homogeneous distributions over molecular features can be seen over physical space (e.g. in heart, breast, and lung)).

Regarding the suggestion to illustrate performance in suboptimal matching situations, we have included an example of mapping a full tissue scale atlas section to a partial coronal in Section 2.2 as reflective of the “missing field of view” suggestion specifically mentioned by the reviewer. As described in the first Results section, the image varifold representation enables modeling of irregularly sampled data, which includes not just datasets in “control” settings with cells and genes dispersed as expected, but also includes settings of disease, missing tissue capture, and tissue distortion. We note that the cost function as described in our Methods Section 4.3, is also amenable to choice of different varifold norm and addition of spatial weights that might appropriately emphasize certain regions for matching over others as have been done in other settings of aligning digital pathology to MRI [13]. Notably, our restriction to estimating geometric transformations that are invertible diffeomorphisms prevents extreme distortion and changing of topology of atlas sections to match target sections, and different models and assumptions can be used if a focus on such settings is desired. We have added a discussion of these “suboptimal” cases in Section 2.2, copied below in addition to the figure caption for the added Figure 4:

Finally, we emphasize the generalizability of our mapping methodology across particularly image-based spatial transcriptomics technologies by showcasing the alignment of a partial cell-segmented BARseq [15, 16] coronal section to a corresponding full coronal CCFv3 section (Figure 4). Given the costs involved with many of these emerging technologies, the partial measurement of tissue areas is incredibly prevalent across both image-based and spot-based technologies [14]. Consequently, this generates even greater variety in the scope and shape of tissue sections measured, only further emphasizing the need for aligning such partial captures to a common scaffold, such as an atlas coordinate system, where information can be merged across the intersection of these captures. The image varifold representation, particularly in its semi-discrete form as used here, is amenable to representing data with regions of missing or disrupted capture as an extension of the irregularity in sampling that is assumed in the general case. To estimate mappings in these cases, variation in both the

forms and sizes of kernels governing the varifold norm (see Section 4.2) can control the granularity to which matches between atlas and target should be evaluated, with coarser scales more appropriate to noisier tissue sections with higher numbers of artifacts. Additionally, the varifold normed difference in the cost function, as presented in (10) in Section 4.3 can be appended with spatially varying weights, as used in settings of mapping digital pathology to MRI [13] to prioritize matching amongst certain “intact” regions over others. We utilize this strategy in Figure 4 to align a partial BARseq section to a corresponding full CCFv3 section through depiction of the initial geometric offset (left) versus the transformed partial BARseq section to the CCFv3 coordinates (right), with appropriate scale, rotation, and diffeomorphic transformations estimated to position the partial capture within the scope of the correct CCFv3 hemisphere.

Mapping CCFv3 section $Z = 437$ out of 1320 to BARseq cell-typed partial coronal section. Left shows initial positioning of CCFv3 and BARseq sections, with CCFv3 section depicted as black mask. Right shows alignment after pulling back BARseq onto CCFv3 section via inverse estimated diffeomorphism.

7. **The manuscript does not sufficiently discuss the computational requirements and demands associated with the implementation of the proposed method. Such information is crucial for readers and potential users to evaluate the feasibility of employing this method in their research, considering their available computational resources.**

We thank the reviewer for their comment. As noted, we have focused on presenting a model and structure of an overall method, with many potential ways of being implemented. In the revised submission, we indicate this flexibility in implementation in using two different implementations of the basic mapping method. We describe two implementations of the image varifold objects (meshes and point clouds) in the first methods section (4.1). We also describe two optimization schemes (simultaneous versus alternating) in the fourth methods section (4.4). As part of this latter section, we have responded specifically to this comment in providing some computational details on runtime and system details of the machine each implementation was developed and tested on coupled with the locations of each of these code repositories on github where further documentation is provided within the README files:

Both single and alternating optimization schemes were utilized in the examples shown in this work and are available with respective git repositories: <https://github.com/kstouff4/xIV-LDDMM-Particle> and <https://github.com/kstouff4/MeshLDDMMQP>. The single optimization

scheme is provided in the context of point cloud image varifold implementations, whereas the alternating scheme is provided in the context of mesh-based image varifold implementations. Both schemes were developed and tested on a 12 GB TITAN V GPU with CUDA 10.2 and NVIDIA-SMI and Driver 440.33.01, with expected runtime varying by size of dataset, but on the order of 2-4 hours for most examples highlighted in this work.

8. While the manuscript focuses on mapping spatial transcriptomic data to brain atlases, I would be curious to understand how adaptable this method is for aligning two different spatial or molecular modalities, especially when the feature sets are disjoint. This could represent a valuable new application of the method. Authors could discuss, and (if feasible), demonstrate the feasibility of applying their method for cross-modality mapping, such as aligning immunohistochemistry staining data with spatial omics data, or other spatial measurements. If feasible, they should explain any necessary modifications or considerations to the method for such applications.

We thank the reviewer for their inquiry. Regarding different molecular modalities, we have not only expanded our array of spatial transcriptomics technologies mapped in this manuscript to include BARseq as well as MERFISH, but we also have tested our method in aligning atlas sections to DAPI-stained images, as an example of digital pathology. This is exhibited in the context of image segmentation as an application of our method. Furthermore, we highlight the alignment in areas such as the outer layer of cortex as analogous to the performance we see in aligning atlas sections to the spatial transcriptomics datasets. This is covered in Section 2.6 in our paper which constitutes a newly added section compared with our initial submission.

With the computational constraints imposed by modeling spatial transcriptomics datasets as classical images, this work has emphasized the mapping of tissue-scale atlases to molecular datasets generated by emerging image-based rather than the often regularized spot-based spatial transcriptomics technologies. However, as exhibited through the modeling of atlas sections as image varifolds, themselves, both as template and target (see Section 2.5), the xIV-LDDMM is equally capable of aligning tissue-scale atlases to alternative molecular and cellular-scale modalities. As an example, we take a DAPI-stained tissue section (Figure 10), digitized at $2.5\mu\text{m}$ resolution, corresponding to the gene-based MERFISH section shown in Figure 1. The DAPI image (~ 13 million pixels) is converted to an image-varifold particle representation by discretization of its image values into ~ 35 discrete bins, and selection of foreground pixels with corresponding image values in the later 30 of these bins. Approximately 250k particles are

used to represent these pixels, with each particle pertaining to a set of 25 neighboring pixels (5×5 square). Particles each carry the distribution of bins into which the image values of these 25 pixels fall, retaining the individual values of each foreground pixel at the highest resolution in contrast to typical image downsampling, in which a pixel only captures the single mean image value of its neighbors.

Figure 10 depicts this particle representation of the DAPI image and its positioning before and after alignment with the corresponding CCFv3 section. Similar to the BARseq mapping illustrated in Figure 6, we see areas of lower cell density (fewer foreground pixels) versus higher cell density (greater and higher intensity foreground pixels) aligning to expected CCFv3 regions (e.g. layer 1 (L1) especially between the hemispheres, and substantia inominata (SI), respectively). Interestingly, the area of the olfactory tubercle (OT) in the CCFv3 section which is initially quite larger than that in the DAPI image remains large following geometric transformation to the target space, with optimal cost minimization favoring this alignment to one in which this area would drastically contract.

Finally, while the estimated geometric transforms here could equivalently pull back the DAPI image into the space of the CCFv3 section, as done in the setting of atlas construction (see Section 7), we show, instead, the CCFv3 section transformed to the space of the DAPI target as an illustration of how our methodology can be used in settings of atlas segmentation. In particular, as described in the setting of resampling image varifolds across scales [3], we demonstrate the feasibility of translating between particle and image representations in resampling the deformed CCFv3 particles with Gaussian smoothing onto the same $2.5\mu\text{m}$ grid of the original DAPI image (top right image in Figure 10). Note that this smoothing extends the borders of the atlas section slightly beyond their respective positioning as particles, as seen in the difference in overlay laterally within L1 between image and particle representations (top and bottom, respectively).

The added figure accompanying this section has the following caption:

Application of image-varifold based method for mapping CCFv3 section to DAPI-stained image of tissue corresponding to MERFISH section seen in top row of Figure 1. Top left depicts original DAPI-stained image, digitized at $2.5\mu\text{m}$ resolution. Thresholded foreground pixels converted to particle image-varifold representation over a feature space of ~ 30 binned grayscale values. Bottom left depicts this particle representation (black points) overlaying the corresponding CCFv3 section in their respective initial coordinate spaces. Bottom right shows alignment of CCFv3 section to DAPI particles following diffeomorphic transformation to the DAPI coordinate space. Top right shows this alignment in image format, with the deformed CCFv3 section image generated by resampling the deformed

CCFv3 particles onto a regular $2.5\mu\text{m}$ grid. White arrows highlight areas of alignment in the area of the substantia inominata (SI) and layer 1 of cortex whereas red arrows highlight areas of questionable alignment in the area of the olfactory tubercle (OT).

Additionally, as the reviewer notes, we focus this manuscript specifically on a method for mapping across modalities and particularly across scales in taking tissue scale atlases to molecular modalities. The assumption of homogeneity we make as a prior on estimated feature distributions is relevant particularly in this context of coarse scale (tissue scale) atlases as templates, with cases of mapping molecular to molecular scale data likely following a different modeling scheme. Consequently, we foresee our method used as described to “map” one molecular modality to another by pulling each of them back into the same atlas space, as done in the case of our unifying multiple MERFISH targets into the single atlas space (Section 2.4). We briefly describe how this can be done with extended modalities towards comparing them and consolidating them into a single atlas in the following added part in our discussion, in response to the reviewer’s comment:

Though we illustrate an example of atlas creation from samples of a single modality, each target to which the same atlas section is mapped can be pulled back into the same coordinate space. This effectively allows different molecular modalities to be *indirectly* mapped to one another, where the challenge of matching different molecular features as needed to map one modality directly to another is facilitated through the scaffold of the presumed homogeneous regions in the atlas that anchor the estimated feature distributions for alignment to each respective molecular modality. Furthermore, this enables the construction of a composite set of molecular and cellular scale data across technologies and modalities (e.g. digital pathology and spatial transcriptomics, as seen in the comparative mappings of the same CCFv3 section to MERFISH (Figure 1) and DAPI images (Figure 10)), that can be integrated into a multi-factorial atlas with defined segmentation schemes according to homogeneous regions across these features.

2 Reviewer 2

This paper presents a method to build spatial correspondence between the molecular-scale transcriptomics and tissue-scale atlases (references) based on an existing image-varifold large deformation diffeomorphic metric mapping (LDDMM) algorithm. In addition, the authors propose a joint framework to estimate the correspondences across different data modalities simultaneously with the associated density function of cell distributions. While the idea of

studying molecular- and cellular-scale populations in a common space under the atlas-based deformation model is new and interesting, the current manuscript requires significant improvement in terms of structure organization, methodological development, and experimental evaluations. My major concerns and comments are listed as follows:

1. **The manuscript needs better organization. Sections 2 and 3 should be categorized as background information rather than results. Section 4 appears to present the proposed joint estimation method. It is essential for the authors to clearly distinguish the previous studies from the novelty and contributions of the proposed work. This will help readers understand the context and importance of the research in relation to existing literature.**

We thank the reviewer for their comment. We have substantially revised and reorganized the manuscript in its current form. Most of the mathematical details that were in Sections 1 and 2 of the Results (Sections 2 and 3 in the overall manuscript) have been moved to Methods sections. We have instead structured the Results around the applications of our method, with the first Results section reserved for presenting the method through the use of an added schematic diagram. In an effort to distinguish our method from other methods in both the areas of classical image registration and spatial transcriptomics registration, we have included in our introduction a survey of approaches in both areas as well as highlights of where we see our method differing from these. In particular, we highlight the use of the image varifold for representing data at tissue scales and molecular scales with possible irregular sampling in space and in the homogeneous assumption we make for each compartment in the atlas to govern the estimation of latent feature laws over the target feature space, which consequently allows for cross-modality matching. This relevant text is copied here from the Introduction:

Hence, atlas construction and segmentation of new data with defined atlases comprise two classes of widespread problems across domains of biology. These are particularly relevant in the context of emerging datasets in spatial transcriptomics, where atlases have still yet to be constructed [9] and where the different technologies used number quite high, thus emphasizing the need for mapping data across samples and technologies into a common coordinate framework for comparison. A central challenge in both atlas construction and segmentation, however, is in aligning data in a single coordinate space that often exists at different scales (e.g. coarse scale atlases to fine scale datasets) and is fundamentally of different functional modalities (e.g. tissue regions to molecular images). These differences preclude an obvious definition of similarity for consequently optimizing alignment.

In the classical image setting, a large family of diffeomorphism-based methods [17] have been developed within the field of Computational Anatomy (CA) [18, 19] for transforming coordinate systems at the tissue scales. These come particularly from multiple labs in the magnetic resonance imaging (MRI) community and have been used both in the contexts of atlas construction and segmentation at these tissue scales [20, 21, 22, 23, 24, 25, 26, 27, 28]. While some applications have constructed atlases of the same modality as target images [29], emphasizing the importance of both affine and nonparametric deformation, as is granted through the use of diffeomorphisms, the challenge of crossing functional modalities has also been addressed through different approaches including matching based on analytical methods using cross-correlation [30] or localized texture features [31], and methods for transforming one range space to another in crossing modalities and scales based on polynomial transformations [32], scattering transforms [13], graphical models [33], and neural networks [34, 35, 8].

In the molecular setting, however, both the diversity and magnitude of data measured by spatial transcriptomics technologies [36] often prohibits the representation of such data as classical continuous images discretized as regular grids and consequently, the direct use of these image alignment methods at the molecular scales. As seen in repositories generated in the BRAIN Initiative Cell Census Network (BICCN) and archived at the Brain Image Library (BIL), these datasets are already on the order of terabytes and will only continue to increase as technologies shift from mouse to human measurements. Hence, while some methods in deep learning have been applied to align single-cell datasets, modeled as regular grid images, both to atlases at the histological scale [12] as well as reference transcriptional atlases that are beginning to emerge [37], most image-based tactics are limited in their ability to represent this high dimensional and memory-intensive data. Furthermore, many learning-based schemes in the context of classical images have relied on the use of extensive numbers (typically on the order of hundreds) of images from different subjects for extracting features useful for cross-modalities and mapping [7, 8]. In the setting of spatial transcriptomics technologies that have been developed only over the last few years, the acquisition of such large training datasets is typically prohibitive for some of these learning-based approaches.

Consequently, an independent class of methods has been developing for aligning spatially resolved transcriptional datasets at these molecular scales. With influence from image-based methods, some of these including GPSA [9] and PASTE [10] focus exclusively on spot-based data in which gene expression is measured in a neighborhood of each “spot” for a regular array of spots, analogous to a voxel grid in an image. In contrast, image-based technologies such as STARmap, BARseq, and MERFISH, which take point measures of individual mRNA molecules or cells, cover measurements irregularly sampled over space according to tissue architecture and dynamics.

Furthermore, natural fluctuation in gene expression over time and space coupled to the dynamics of each spatial transcriptomics technology leads each tissue section, at the molecular (1-100 micron) scale, to have a varying number of such particles with no natural ordering of particles consistently apparent between sections. Consequently, landmark-based methods [11] that assume direct permutation correspondence between particles are not applicable. Methods aimed at generalizing to allow alignment within and across these additional technologies are typically rooted in different data representations, such as graphs, in the case of SLAT [4] that aims to find correspondence between cells or groups of cells in atlas and target. Additionally, while many methods have showcased success at aligning different replicates within a single technology for both human and mouse samples, cross-technology alignment has typically relied on a nonempty intersection of feature sets across these technologies [4, 12], limiting extension of these methods to the type of cross-modality and cross-scale mapping required for integrating tissue-scale atlases and molecular scale data.

Here, we build on both of these lineages in presenting a model equally equipped at representing tissue-scale atlases and molecular-scale data, and an associated cross-modality registration mechanism that addresses both challenges of crossing scales and functional modalities by estimating simultaneously geometric and functional transformations to align each dataset to the other. Specifically, we harness the generalizability of image varifolds, which have emerged in molecular CA [3], for simultaneously modeling molecular and tissue-scale data with both irregular and gridlike sampling schemes in a common framework (addressed in SLAT, for instance, by a graph-based representation in the setting of molecular and cellular data [4]). A subproblem covered by the image-varifold theory outlined in [5] is the mapping of molecular scale data to atlas coordinate systems. As specified there, we estimate minimal energy diffeomorphic transformations through large deformation diffeomorphic metric mapping (LDDMM) [6] with the action of diffeomorphisms on image varifolds, defined in a manner consistent with how diffeomorphisms have transformed classical images from one coordinate space to another [3]. In tandem, we estimate a latent distribution over the molecular functional space for each atlas partition, without need for the large datasets often used in deep learning approaches, but by instead relying on an assumption of spatial homogeneity in distribution across each atlas partition. We use this latent distribution to transform the functional space of each atlas (i.e., its ontology) to the molecular functional space. We refer to this method, throughout, as “cross image-varifold LDDMM” (xIV-LDDMM), where the cross emphasizes the additional estimation of latent feature laws in tandem with a diffeomorphism as in the case of IV-LDDMM [5].

2. **The authors have not quantitatively validated their experimental results. No thorough statistical analysis of the estimated**

results (e.g., estimated mRNA densities, the diffeomorphic transformations, and the predicted oncology using estimated transformations.) is provided. Additionally, no comparison is made between the proposed method and existing approaches (e.g., the original LDDMM for mesh data and other related works), making it difficult to assess the efficacy of the experimental results.

We thank the reviewer for their suggestion. We have worked to quantify the accuracy of our method by assessing the set distance between cell markers of a given type and the corresponding region in the tissue atlas section to which we align these cell-based datasets. We specifically outline this procedure in our added Results Section 2.3. We have added authors, Xiaoyin Chen and Mara Rue, to our revised manuscript for their work in helping us to identify particular cell types they prescribe to measured cells in their BARseq data that correspond to particular atlas regions. We focus specifically on those in hippocampal regions of CA1, CA3, and the Dentate Gyrus. As shown in Figure 5, the initial alignment between atlas sections and BARseq cell-based datasets with cell-type features boasts differences in rotation, translation, scale, and overall geometry warranting estimation of both affine and nonrigid deformation. We quantify accuracy by transforming target sections to the atlas section via the inverse diffeomorphism. We then find for each cell of the given type, the closest indexed location in the atlas of the region to which this cell type belongs. The distance to this nearest point is measured for each cell of the given type and distances are shown in the histograms of Figure 5, with mean and median distances over each cell type analyzed typically on the order of $10 - 30\mu m$. This gives an expectation of the accuracy within which one might expect to match corresponding regions in a molecular target with those of a given atlas. The relevant text from the added section 2.3 is copied below for convenience:

We specifically evaluated the efficacy of xIV-LDDMM in mapping CCFv3 sections to corresponding coronal sections of BARseq cell-segmented and subsequently cell-typed data. As described in [16], cells were segmented in the BARseq data using Cellpose [38]. Gene reads were assigned to cells, and cells containing fewer than 5 unique genes and 20 total gene counts were excluded. Cells were clustered using an iterative clustering approach based on Louvain clustering to achieve a similar resolution as “subclass” in recent single-cell RNAseq studies [39].

Given the difference in atlas and target modalities mapped with xIV-LDDMM, traditional measures of accuracy, such as Dice overlap score, as are often used to evaluate image registration tools cannot be directly applied. Furthermore, landmarks common to both atlas and target modalities are not typically readily available or necessarily identifiable given the diversity in measures taken by molecular modalities [9]. Consequently, to

evaluate accuracy, particularly in the geometric alignment achieved with xIV-LDDMM, we matched corresponding atlas feature values (regions) with target feature values (e.g. cell types) and quantified the resulting distance individual cells of a given type are from the atlas region in which we expect to find them.

We quantified accuracy by computing the set distance between cells of types specific to hippocampal regions (e.g. CA1, CA2, and CA3 pyramidal cells and DG granule cells) and particles of these regions in the CCFv3. We specifically defined this distance for each cell, indexed by $c \in \mathcal{C}_R$, for region R , as:

$$d_c = \min_{a \in \mathcal{A}_R} \|x_a - y_c\|_2^2,$$

where location y_c was the given 2D coordinate of the cell in the native BARseq coordinate system, and the set \mathcal{A}_R refers to the set of indexed locations in the atlas section with feature value (region label) of R (e.g. with $\nu_a(R) = 1$, and $\nu_a(f) = 0$ for $f \neq R$). We compared these distances to those after deformation, by replacing y_c with the mapped position of each cell marker to the CCFv3 section: $\varphi^{-1}(y_c)$, via the inverse diffeomorphism estimated in our joint image-varifold based method. Figure 5 depicts these measures for two separate CCFv3 sections mapped to two corresponding sections of BARseq cell data. The left column shows initial positioning of CCFv3 regions versus cells, with the histogram of distances for all cells, $c \in \mathcal{C}_R$, of types $R \in \{\text{CA1, CA3, DG}\}$, depicted below. Initial distances are on the order of 1 mm, where notably, neighboring particles in the CCFv3 are at 10 μm , thus giving a lower bound to distance metrics we might expect for cells to be neighboring CCFv3 particles. The right column, in contrast, shows the positioning of CCFv3 regions versus cells following geometric transformation. Kernel size governing diffeomorphism regularization and varifold norm matching were both on the order of 100 μm . Median distances for all three regions in the first section are on the order of 20 – 30 μm whereas those in the second section are on the order of 5 μm for CA1 and DG and 300 μm for CA3, which we assume is coming from the separate group of cells labeled as CA3 at the edge of CA1 (indicated by the white star) in contrast to the group overlaying the CCFv3 region, which falls within the first 10 μm bin. Note that this discrepancy likely stems in part from the approximation of each BARseq section as a strictly coronal section of the Allen CCFv3, where the cutting plane is instead slightly offset from this coronal plane. In any case, we observe in both sections an accuracy on the order of 10 – 30 μm , analogous in the image setting to cells being mapped to within 1-3 pixels (at a resolution of 10 μm) to the appropriate atlas region.

The figure caption corresponding to the added Figure 5 is as follows:

Quantification of accuracy in two coronal slices of Allen CCFv3 ($Z = 877$ (top) and $Z = 837$ (bottom) out of 1320) mapped to two corresponding slices of BARseq cell data classified into 52 cell types. Histograms depict set distance of cell markers to corresponding atlas region in three hippocampal regions: CA1, CA3, and Dentate Gyrus (DG). Images illustrate overlap of cell markers (circles) on corresponding atlas regions. Left columns depict initial overlap and right columns depict overlap after deformation to atlas space using inverse estimated diffeomorphism.

In response to the latter half of this comment, we defer the reviewer to our response to their fourth comment that suggests a comparison to a simple disjoint approach. We have included in the added Section 2.3 not only these quantitative metrics but also a comparison of our method applied to map an atlas section to a BARseq cell-based dataset versus an image-based LDDMM method that aims to achieve alignment through minimization of a norm based on foreground-background. In illustration of this comparison, we have also provided two quantification measures. We quantified the relative density of mapped cells to areas of white matter (corpus callosum and dorsal hippocampal commissure) versus cortical layers to show misalignment in the case of image-based LDDMM versus image-varifold based LDDMM. We also quantified the percent of cells with a type corresponding to a cortical layer falling within the correct cortical layer following mapping with xIV-LDDMM, to highlight the majority ($\sim 80\%$) of cells falling within their corresponding cortical layer in the atlas section. We copy specifically the discussion of these quantitative measures below with the associated supplementary figure caption:

This becomes clear in comparing the alignment of BARseq to CCFv3 section in areas of low versus high cell density where these areas of low density correctly align with layer 1 and corpus callosum (CC) as a result of the image-varifold based approach but not in the image-based approach (right column). Notably, the corpus callosum and dorsal hippocampal commissure appear with equivalently low cell density on both right and left hemispheres in the image-varifold based approach (yellow arrow) whereas we see these areas partially covered by CA1 cells, particularly in the right hemisphere in the image-based approach. Layer 1 (white arrow) is covered by high cell density around the entire circumference of the coronal section in the image-based approach, whereas it is visible, with relatively few cells mapping to it, as expected, in the image-varifold based approach. Supplementary Figure S3 shows the relative cell density in each of the cortical layers and white matter structures within the neighborhood of the primary visual cortex, with xIV-LDDMM showing more accurate alignment with higher levels (4-6x) in layers 2-6 compared with layer 1 and white matter. Image-based LDDMM, in contrast, yields similar levels of cell density across all of these structures, with the band of high cell density

in BARseq layers 2-6 covering the entirety of the cortex, thus generating overestimates (4-6x) of cell density in layer 1. Cell density within CA1 also covers the areas of the corpus callosum, leading to overestimates of density there. Finally, as the image-varifold based approach jointly considers both variations in total density of cells and relative distribution of cell type, we see within each layer of cortex delineated in the CCFv3, a majority of corresponding cell types indicating correct layer-by-layer alignment (bottom right). Classification of cells by nearest atlas region yields 88% of layer 2/3 cells (total 443), 86% of layer 4/5 cells (total 436), 87% of layer 5 cells (total 278), and 70% of layer 6 cells (total 608) classified correctly according to cortical layer, within a 1.5 mm² section through the primary visual cortex shown in Figure 6. In contrast, for a similar section in the result of image-based mapping (Figure 6, top right), 38% of layer 2/3 pixels (total 308), 36% of layer 4 pixels (total 269), 64% of layer 5 pixels (total 225), and 78% of layer 6 pixels (total 326) are classified correctly according to nearest cortical layer.

Figure Caption: Comparative cell density in layers of cortex and white matter (corpus callosum and dorsal hippocampal commissure (CC+DHC)) for BARseq cell-segmented coronal section mapped to Allen CCFv3 section $Z = 837$ via estimation of diffeomorphism using image-based LDDMM (top) versus image-varifold based LDDMM (bottom). Densities per region computed as cells per square millimeter of atlas tissue in each respective region. For comparison between methods, densities in each region are normalized against the absolute density in layer 1 (L1). xIV-LDDMM yields expected cell densities 4-6x higher in layers 2-6 compared with L1 and CC+DHC versus LDDMM, where similar cell densities across cortical layers and white matter result from misalignment in these areas.

3. **The title "A Universal Method" may not be appropriate since the proposed method (according to section 4) is based on derived meshes from datasets with different modalities. The authors do not seem to have made significant methodological contributions to make the method universally applicable to different modalities.**

We thank the reviewer for their suggestion. We have revised the title to capture more specifically the body of problems we believe can be solved by our method and the results shown. While mesh-based data was our primary focus in the previous submission, we have added additional modalities rendered as particles, not meshes, to elucidate the generality of our method for mapping data at different scales and of different modalities including histology and spatial transcriptomics at the molecular scales and MRI and segmented atlases at the tissue scales.

The modified title is:

A 2D Cross-Modality Mapping Method using Image Varifolds to Align

Tissue-Scale Atlases to Molecular-Scale Measures

4. **The paper lacks sufficient motivation for jointly estimating atlas-based transformations and the latent density functions of molecular features. Previous works in the literature (e.g., [1,2,3]) have demonstrated that combining atlas-based transformations as geometric features with image features within or across multiple modalities improves specific tasks such as image classification and segmentation. However, it is unclear why the jointly estimated transformations and density distributions of molecular features are important or necessary. The authors at least should compare with a simple disjoint approach.**

[1] Wang and Zhang, **Geo-SIC: Learning Deformable Geometric Shapes in Deep Image Classifiers, 2022.**

[2] Ding et. al., **Cross-Modality Multi-Atlas Segmentation via Deep Registration and Label Fusion, 2022.**

[3] Ding and Niethammer, **Aladdin: Joint Atlas Building and Diffeomorphic Registration Learning with Pairwise Alignment, 2022.**

We thank the reviewer for their comment and the provided references. To address the concern of comparing to a simple disjoint approach, we have included in our Results Section 2.3, a comparison of our image varifold based joint methodology to an image-based disjoint approach, where the different feature spaces of the atlas and target are collapsed into a single foreground-background modality so that only geometric transformations are estimated. To address the reviewer's second comment about a comparison to "original LDDMM", we specifically use image-based LDDMM in this comparison. We highlight that inevitably, the representation of irregularly sampled data as an image discretized on a regular grid loses resolution that can be kept in the setting of image varifolds, implemented, for instance, as point clouds. Furthermore, the mapping based on foreground-background does not achieve as close alignment between the boundaries of regions internal to the overall structure of the atlas and target, such as the matching of corpus callosum and the outermost layer of cortex to areas of lower cell density in the target. The alignment of such areas comes as a result of our jointly estimating feature laws and geometric transformations that are based on the alignment of presumed homogenous regions, such as the lower density corpus callosum and cortical layer 1. We highlight these differences in the added Figure 6 to our manuscript and the corresponding text that discusses this comparison:

We also compared our joint image-varifold based approach in xIV-LDDMM to a classical image-based approach matching, estimating only a geometric transformation and matching based on foreground-background.

We mapped one of the same CCFv3 sections ($Z = 837$) to an image rendering of the BARseq cell section, with grayscale intensity capturing density of cells over space. Both CCFv3 and BARseq images were discretized at $50 \mu\text{m}$ resolution to estimate the mapping, with the latter computed with Gaussian smoothing, and image-based LDDMM, as described in [6] was used to estimate optimal diffeomorphic transformation of atlas to target. Figure 6 demonstrates both similarities and differences in these methods. Here, a $10 \mu\text{m}$ BARseq image was transformed with the estimated diffeomorphism to the CCFv3 space (top) for comparison to the particle-based representation at full resolution used for estimation of the diffeomorphism in the image-varifold based mapping (bottom). At this resolution, ~ 1.5 million pixels are needed to model the BARseq data as an image whereas only $\sim 90\text{k}$ particles are needed in the image-varifold representation. In addition to the computational and memory expense of image handling, we compromise our ability to delineate one cell from another, as evidenced in the blur in the image rendering versus image-varifold representation (left column, Figure 6), with 75% of cells having a nearest neighbor less than $23 \mu\text{m}$ away and 25% having a nearest neighbor less than $3.5 \mu\text{m}$ away in the BARseq capture. Nevertheless, with a kernel of bandwidth $10 \mu\text{m}$ used to generate the $10 \mu\text{m}$ smoothed image and most regions at least on the order of $100 \mu\text{m}$ in width at their narrowest (e.g. corpus callosum), the majority of cell mass stayed primarily within the initial region it was found.

Regarding alignment, the left column illustrates that such an image-based approach can achieve global alignment of template to target, with initial differences in scale and shape shown in the bottom left of Figure 5. However, we see clear differences in alignment on the order of $50 - 150 \mu\text{m}$ by examination of the overlap in the different layers of cortex and adjacent corpus callosum. Importantly, as described in Section 2.1, in the joint image-varifold based approach of estimating geometric transformations and latent distributions, we model each atlas region as homogeneous and stationary with respect to space. This gives an optimal alignment between atlas and target that maximizes similarity in distribution over features across each site in a single atlas region while minimizing the energy of the geometric deformation (diffeomorphism). We would consequently expect this to skew emphasis away from the foreground-background boundaries that typically govern image alignment and instead highlight the underlying assumptions in the architecture of the cartoon atlas, whose boundaries were initially constructed so as to maximize the homogeneity of the region. This becomes clear in comparing the alignment of BARseq to CCFv3 section in areas of low versus high cell density where these areas of low density correctly align with layer 1 and corpus callosum (CC) as a result of the image-varifold based approach but not in the image-based approach (right column). Notably, the corpus callosum and dorsal hippocampal commissure appear with equivalently low cell density on both right and left hemispheres in the image-varifold based approach (yellow arrow) whereas

we see these areas partially covered by CA1 cells, particularly in the right hemisphere in the image-based approach. Layer 1 (white arrow) is covered by high cell density around the entire circumference of the coronal section in the image-based approach, whereas it is visible, with relatively few cells mapping to it, as expected, in the image-varifold based approach. Supplementary Figure S3 shows the relative cell density in each of the cortical layers and white matter structures within the neighborhood of the primary visual cortex, with xIV-LDDMM showing more accurate alignment with higher levels (4-6x) in layers 2-6 compared with layer 1 and white matter. Image-based LDDMM, in contrast, yields similar levels of cell density across all of these structures, with the band of high cell density in BARseq layers 2-6 covering the entirety of the cortex, thus generating overestimates (4-6x) of cell density in layer 1. Cell density within CA1 also covers the areas of the corpus callosum, leading to overestimates of density there. Finally, as the image-varifold based approach jointly considers both variations in total density of cells and relative distribution of cell type, we see within each layer of cortex delineated in the CCFv3, a majority of corresponding cell types indicating correct layer-by-layer alignment (bottom right). Classification of cells by nearest atlas region yields 88% of layer 2/3 cells (total 443), 86% of layer 4/5 cells (total 436), 87% of layer 5 cells (total 278), and 70% of layer 6 cells (total 608) classified correctly according to cortical layer, within a 1.5 mm² section through the primary visual cortex shown in Figure 6. In contrast, for a similar section in the result of image-based mapping (Figure 6, top right), 38% of layer 2/3 pixels (total 308), 36% of layer 4 pixels (total 269), 64% of layer 5 pixels (total 225), and 78% of layer 6 pixels (total 326) are classified correctly according to nearest cortical layer.

The captions for the added figure and supplementary figure are also copied here for convenience:

Comparison of mapping CCFv3 section $Z = 837$ out of 1320 to corresponding BARseq cell data using image-based LDDMM [6] (top) versus image varifold-based mapping method (bottom). Source and target images rendered at 50 μm for image-based LDDMM, with matching based on foreground/background in Allen atlas section to smoothed cell density in BARseq. Estimated transformations applied to 10 μm smoothed cell density image (left) and 5 μm cell type image (right) for comparison between mapping methods. Overall alignment shown in left column, with noticeable misalignment of high cell density areas to layer 1 and white matter structures (corpus callosum and dorsal hippocampal commissure) in CCFv3 section with image-based LDDMM. Right column shows these regions magnified, highlighting the layer-by-layer mismatch ($\sim 150\mu\text{m}$) in the setting of image-based mapping versus the matching layers in CCFv3 partition of cortex to cell types in BARseq following image-varifold based

mapping (bottom right).

Comparative cell density in layers of cortex and white matter (corpus callosum and dorsal hippocampal commissure (CC+DHC)) for BARseq cell-segmented coronal section mapped to Allen CCFv3 section $Z = 837$ via estimation of diffeomorphism using image-based LDDMM (top) versus image-varifold based LDDMM (bottom). Densities per region computed as cells per square millimeter of atlas tissue in each respective region. For comparison between methods, densities in each region are normalized against the absolute density in layer 1 (L1). xIV-LDDMM yields expected cell densities 4-6x higher in layers 2-6 compared with L1 and CC+DHC versus LDDMM, where similar cell densities across cortical layers and white matter result from misalignment in these areas.

We have also reviewed the cited references and included them as part of our introduction as a comparison to the method we pose in this manuscript. For instance, compared with a deep learning approach such as [8, 7], that might require on the order of at least 50-100 images to train a model to extract relevant features for use in mapping across modalities, we instead impose a prior on our estimated feature laws, π , based on each atlas region's presumed homogeneity over space. By doing so, we eliminate the need for large training datasets as might not be available in the context of these emerging spatial transcriptomics technologies as of yet. We highlight the emphasis placed on using affine and nonparametric deformation in [29] as impetus for our estimation of diffeomorphisms over simply affine transformations, though the work in [29] focuses primarily on atlas construction from targets of a single modality rather than the distinct mapping of an atlas of one modality to targets of another. The paragraph summarizing the comparisons in the introduction is copied below:

While some applications have constructed atlases of the same modality as target images [29], emphasizing the importance of both affine and nonparametric deformation, as is granted through the use of diffeomorphisms, the challenge of crossing functional modalities has also been addressed through different approaches including matching based on analytical methods using cross-correlation [30] or localized texture features [31], and methods for transforming one range space to another in crossing modalities and scales based on polynomial transformations [32], scattering transforms [13] and machine learning [33, 34, 35, 8]. In the molecular setting, however, both the diversity and magnitude of data measured by spatial transcriptomics technologies [36] often prohibits the representation of such data as classical continuous images discretized as regular grids and consequently, the direct use of these image alignment methods at the molecular scales...Furthermore, many of learning-based schemes in the context of classical images have relied on the use of extensive numbers

(typically on the order of hundreds) of images from different subjects for extracting features useful for cross-modalities and mapping [7, 8]. In the setting of spatial transcriptomics technologies that have been developed only over the last few years, the acquisition of such large training datasets is typically prohibitive for some of these learning-based approaches.

5. **Section 2.3 seems disconnected from the proposed method. It is unclear how it relates to the joint estimation method. Is dimensionality reduction particularly crucial for the core algorithm, or is it a strategy employed solely for the experimental results in this paper?**

We thank the reviewer for their suggestion and inquiry. As presented, the method does not *require* dimensionality reduction. While the computational complexity (time and memory) increases with more features, this dimensionality reduction step is not integral to our method, and was used in this context for simplicity in interpretation of results (i.e. evaluating distributions over a smaller set of genes than an entire set on the order of hundreds) as well as reducing runtime. Upon review, we have elected to remove it from the manuscript, as we believe it might distract from the key points of the central method presented. We have adjusted the Figure caption related to the dataset for which we used this method to select a subset of genes as follows:

Figure Caption: Relative expression for three genes (*Gfap*, *Trp53i11*, *Wipf3*) out of a set of twenty (shown at the right), with demonstrated spatial variability according to a computed mutual information score on a section of MERFISH transcriptomics data (top row). Bottom shows estimated expression for same three genes in each region of the CCFv3 section $Z = 485$ out of 1320, as part of the latent distribution over genes estimated in tandem with a geometric transformation to align the CCFv3 section to MERFISH section. Right column depicts gene with highest probability in MERFISH target (top) and as predicted in the latent distribution (bottom) for the CCFv3 section.

6. **In Figure 4, the estimated cell densities appear to be very noisy compared to the atlas. It would be helpful to clarify whether this noise is due to the discontinuity of the target subject. If so, the assumptions of diffeomorphic (smooth) transformations between the atlas and individual subjects need to be justified.**

We thank the reviewer for their comment, but we are uncertain as to what the reviewer is particularly referring to in Figure 4 in comparing the “atlas” to “estimated cell densities”, as the cell densities displayed in the first row are the densities on the target (not estimated) versus those

displayed in the second row are those on the atlas (estimated). To address this comment, we first note that the mentioned Figure 4 is now part of Figure 3 in the second Results section, with the following description:

The left two columns indicate the target cell density and distribution over cell types (depicted per simplex as the cell type with highest probability) and corresponding latent cell type distribution per each CCFv3 region, estimated in tandem with the geometric transformation taking CCFv3 section to MERFISH spatial coordinates. The right two columns, however, highlight the result of mapping the same CCFv3 section to the MERFISH section with an alternatively defined feature space over gene types. We combine both cell density and gene expression by aggregating the individual mRNA transcripts into an average gene expression feature per cell and normalizing the total mRNA per cell to 1. The target and estimated probabilities for two gene types (*Baiap2*, *Slc17a6*) out of a chosen set of six are shown following estimation of an alternative geometric transformation and latent distribution, now, over gene types. Correspondence is seen in both the absolute magnitude as well as relative probability of each gene type between those in the target MERFISH section and those resulting from the estimated feature laws, π . Two examples include the area of high *Baiap2* gene type adjacent to the hippocampus (probability ≈ 0.95) and the area of high *Slc17a6* in the rhomboid nucleus (probability ≈ 0.6). This correspondence serves to reinforce the validity in the estimated feature laws in xIV-LDDMM.

Second, we mention that the images shown in the second column in this Figure reflect not the entire distribution over cell types at each indexed location, but the most likely cell type for each indexed location (simplex) in the target (top row) and each region in the atlas. Consequently, we might see more fluctuation (noise) in the most likely cell type from location to location in the target than in the probability of that cell type in the context of the whole distribution. The images in the third and fourth columns, for instance, show the probability of particular gene types in a set of 6 gene types rather than summarizing distributions via the most likely gene type only.

Third, we clarify the two “smoothness” assumptions we make in both the geometric transformation and feature distributions estimated throughout. Regarding the geometric transformation, we constrain ourselves to estimating smooth, invertible diffeomorphisms as we highlight examples throughout the manuscript where we wish to push atlases onto a target (e.g. in segmentation) versus where we wish to pull back targets into a single atlas space (e.g. cross-replicate comparison or atlas construction), therefore requiring this invertibility. We also build on the vast literature in

computational anatomy giving us the metric theory around such diffeomorphisms for measuring “distances” between shapes. Second, we highlight the assumption we make of homogeneity within each atlas compartment which speaks to the “smoothness” we assume of the distribution over target feature values over the space of each region. While this distribution might change abruptly from region to region, the estimation of a single distribution for the entirety of each region manifests in the same single most likely cell type shown over the entirety of each single region in the atlas, as in Figure 3. We have added some of this clarification to the description of Figure 2:

The fourth column of Figure 2 showcases another comparative summary of the estimated feature distributions for the CCFv3 section in depicting at each location the most probable gene type. The assumption of spatial homogeneity in distribution is evidenced here, particularly in large areas of the CCFv3, such as the striatum, where each gene carries a single probability of expression across the entire region and *Kirrel3* is uniformly the most probable gene type.

- 7. The manuscript does not clearly state the objective functions for estimating the latent conditional feature laws. It would be beneficial to provide a clearer explanation in the current version.**

We thank the reviewer for their suggestion. In accordance with suggestions from the other two reviewers, we have moved much of the mathematical detail of the manuscript to the Methods section. Here, for simplicity, we specify the objective function we seek to optimize in the general case. As further described in the Methods section on optimization algorithms used, we present both single and alternating schemes for optimizing jointly the geometric transformation and latent feature distributions. In the latter case, constraints on the latent feature distributions are imposed through the quadratic program defined whereas in the former case, regularization terms are added to the general objective function. We have added this clarification particularly the third methods section as follows:

As described in Section 2.1, the mapping variational problem constructs a diffeomorphism $\varphi : \mathbb{R}^2 \rightarrow \mathbb{R}^2$ and feature laws $(\pi_\ell)_{\ell \in \mathcal{L}}$ on \mathcal{F} to carry the atlas image varifold onto the target, minimizing the varifold normed difference between them, with norm defined as in Section 4.2. We follow LDDMM as described in the image case, [6], parameterizing the diffeomorphism with the smooth time varying velocity field, $v_t, t \in [0, 1]$, as $\dot{\varphi}_t = v_t \circ \varphi_t$. This gives the variational problem between atlas image varifold, μ_A , with indexed locations, I , and target image varifold, μ_T , with indexed locations, J :

Variational Problem 1

$$\inf_{\substack{v \in L^2([0,1],V), \\ \pi_\ell, \ell \in \mathcal{L}}} \frac{1}{2} \int_0^1 \|v_t\|_V^2 dt + \|\varphi_1 \cdot \mu_A^\pi - \mu_T\|_{W^*}^2 \quad (2)$$

$$\text{with } \dot{\varphi}_t = v_t \circ \varphi_t, \quad \varphi_0 = Id,$$

with μ_A^π depicting the feature transformed atlas image varifold, as described in Section 2.1, as $\sum_{i \in I} \delta_{x_i} \otimes \nu'_i$ with $\nu'_i = \sum_{\ell \in \mathcal{L}} \nu_i \pi_\ell$, the estimated distribution over target features for indexed location i in the atlas image varifold. The space of smooth time-varying velocity fields giving the flow, φ , is defined as a reproducing kernel Hilbert space, equipped with norm, $\|\cdot\|_V^2$, ensuring smoothness and invertibility of φ as a diffeomorphism [40].

We solve (10) for optimal φ, π through either single or alternating algorithms as described in Section 4.4, with the opportunity to impose priors on the estimated distributions, π , appropriate to the specific setting. For instance, we typically impose positivity on all values in estimated distributions: $\pi_\ell(f) \geq 0$ for all $f \in \mathcal{F}$ and $\ell \in \mathcal{L}$. Given prior knowledge on the spatial density of cells or mRNA of a target, we may also specify a range of values for the resulting spatial density to take through constraints on the total measures of features estimated:

$$w^{\min} \leq w'_i \leq w^{\max}, \quad i \in I \quad (3)$$

with $w'_i = \sum_{f \in \mathcal{F}} \nu'_i$ and ν'_i as in (8). This prior can be easily incorporated as a constraint in the setting of the quadratic program used in the alternating algorithmic approach for estimating optimal distributions π (see Section 4.4). In the examples shown here, we typically take w^{\min} to be the 5th percentile of values $w_j, j \in J$ of the target image varifold. Additionally, in the setting of mapping tissue- scale atlases to each other, as discussed in Section 2.5, we impose the greater constraint of ensuring constant spatial density of 1, as we use for modeling both atlas and target image varifolds in this case. Finally, in other settings without prior knowledge or wish to impose any on the specific densities prescribed to each indexed location, we can add a general regularization term to (10), such as the Kullback-Liebler divergence between the normalized estimated π probability distribution and a uniform distribution across target features:

$$d_l \simeq \sum_{f \in \mathcal{F}} \bar{\pi}_\ell(f) \log\left(\frac{\bar{\pi}_\ell(f)}{(1/|\mathcal{F}|)}\right), \quad \ell \in \mathcal{L} \quad (4)$$

where $|\mathcal{F}|$ gives the number of discrete feature values in the target feature space and the probability distribution, $\bar{\pi}_\ell = \frac{1}{\sum_{f \in \mathcal{F}} \pi_\ell(f)} \pi_\ell$. We use this approach in the examples with BARseq data shown and in the context of a simultaneous rather than alternating optimization algorithm.

8. **The mathematical notations in the manuscript are either poorly defined, or used before clear definitions are provided. For example,**

L206: the definition of ‘varifolds’ needs to be defined earlier.

We thank the reviewer for their suggestion. In response to many of the other comments by reviewers 1 and 3, we have moved much of the mathematical details in definitions and optimization schemes to the methods section and aimed to make the introduction and results more appreciable by a wider audience. Our first Results section still aims to specify our basic model and mapping approach and therefore, we have included a more explicit definition of an image varifold, as those objects specifically relevant to our model, at the very beginning of this section:

To accommodate the high feature dimensionality and spatial irregularity of molecular datasets, as described in the Introduction, we harness the recent work that extends the theory of diffeomorphisms in the setting of images to the setting of image varifolds and estimation of correspondences between them [3, 5]. We first describe this framework of image varifolds to emphasize their capacity for modeling diverse types of tissue-, cellular-, and molecular-scale data. Image varifolds are geometric measures over the product space, $\mathbb{R}^2 \times \mathcal{F}$, therefore encompassing measures over both physical and feature spaces, \mathbb{R}^2 and \mathcal{F} , respectively.

L221-223: please make sure to define all math notations before using them.

We thank the reviewer for their comment. While we have removed the specific lines of mathematical notation referenced in this comment from the current version of the manuscript, we have added clarification to the definition of the Dirac measure given as was referenced in these previous lines. This specification comes in the first Results section:

The “Dirac” measures over physical space are denoted throughout by δ_x with x a singleton set (e.g. single point) in \mathbb{R}^2 and therefore, $\delta_x(\{y\})$ evaluating to 1 on another singleton set, $\{y\} \in \mathbb{R}^2$ if $x = y$ and 0 otherwise.

L265-266: what are I and J ? Please clarify.

We thank the reviewer for their inquiry. As noted in the previous response, we have removed these particular lines in the current manuscript. The construction of a mesh implementation of an image varifold is instead only covered in the first methods section. We have accordingly added in

explicit clarifications of what the sets I and J correspond to in this context:

In the case of meshes, each mesh is built from a collection of vertices indexed by the set I , $\mathbf{x} = (x_i)_{i \in I}$ with each $x_i \in \mathbb{R}^2$. Each simplex in the mesh is defined from the vertices denoted as $\gamma(\mathbf{x})$ and is paired with a 3-tuple with components that index the vertices of the simplex, $(\gamma(\mathbf{x}), c = (c^1, c^2, c^3) \in I^3)$ and determine the center $m(\mathbf{x}) = \frac{1}{3}(x_{c^1} + x_{c^2} + x_{c^3})$. Each triangle simplex is defined by

$$\gamma(\mathbf{x}) = \left\{ y \in \mathbb{R}^2 : y = \sum_{k=1}^3 a_k x_{c^k}, a_k \geq 0, \sum_{k=1}^3 a_k = 1 \right\}, \quad (5)$$

with positive orientation and volume $|\gamma_c(\mathbf{x})| := \frac{1}{2} \|(x_{c^2} - x_{c^1}) \times (x_{c^3} - x_{c^1})\| > 0$.

The total mesh τ is the collection of vertices \mathbf{x} , and simplices and centers $(\gamma_j(\mathbf{x}), c_j = (c_j^1, c_j^2, c_j^3), m_j(\mathbf{x}))_{j \in J}$, with the simplices indexed by the set J , and with the resolution determining the complexity as total numbers of vertices $|I|$ and the number of simplices $|J|$ in the mesh. To complete the image varifold we append to the mesh the spatial density defined over the area of each simplex: $\alpha = (\alpha_j)_{j \in J}$ and the field of probability laws $\zeta = (\zeta_j)_{j \in J}$ on \mathcal{F} .

L1111: the definition of id is missing.

We thank the reviewer for their comment. We have added a clarification of id being the identity element in the group of diffeomorphisms, with $\varphi(x) = x$ to the last methods section covering the different optimization schemes:

The alternating scheme specifically follows [5], fixing the laws $(\pi_\ell)_{\ell \in \mathcal{L}}$ and optimizing over the control $v(t), t \in [0, 1]$ and integrating it, with initial condition at the identity element, $\varphi_0 = Id$ (e.g. $\varphi_0(x) = x, \forall x \in \mathbb{R}^2$), to generate the diffeomorphism φ_1 . The diffeomorphism is then fixed and quadratic programming, such as OSQP [41], used to estimate the feature laws.

L1126: the steps of using quadratic programming in Eq. (9) need to be clarified.

We thank the reviewer for their comment. In our revision of the methods section covering the quadratic program, we have first simplified the notation so as to cover the most general case (point clouds and not necessarily meshes) of implemented image varifolds. We hope this contributes to clarifying the steps involved in the quadratic program (as indicated in Algorithm 1) and the form of the quadratic program given in Equation (6).

Second, we clarify that we have not implemented a quadratic programming solver to solve the form of the equation given in (6); rather, we use the implementation of OSQP provided by the python package `qpsolvers`. We indicate this as well in the last Methods section in the revised manuscript with the specific alternations shown below:

Algorithm 1

Initialize: $\pi_\ell(f) = \frac{1}{|\mathcal{F}|}, f \in \mathcal{F}$

A: Solve for v :

- (a) Update and fix $(\pi_\ell)_{\ell \in \mathcal{L}}$.
- (b) Solve LDDMM, optimizing (10) with respect to vector field $v_t, t \in [0, 1]$.
- (c) Solve for φ_1 , integrating O.D.E $\varphi_1 = \int_0^1 v_t \circ \varphi_t dt + Id$.
- (d) Flow μ_A^π according to φ_1 , giving $\varphi_1 \cdot \mu_A^\pi$.

B: Solve for $(\pi_\ell)_{\ell \in \mathcal{L}}$:

- (a) Fix spatial positions in deformed template, $\varphi(\mathbf{x}_i), i \in I$.
- (b) Optimize quadratic program (6) with respect to $(\pi_\ell)_{\ell \in \mathcal{L}}$.

Return to A

For the atlas, we define the form of the image varifold: $\mu_A = \sum_{i \in I} \delta_{x_i} \otimes \nu_i$, with, $\nu_i \in \mathcal{M}(\mathcal{L})$, a measure over the atlas feature values (partitions). The estimated feature distributions are given via π as the mixture distribution, $\nu'_i = \sum_{\ell \in \mathcal{L}} \nu_i(\ell) \pi_\ell$. We specify the target over the index set, J , as: $\mu_T = \sum_{j \in J} \delta_{x_j} \otimes \nu_j$ with each ν_j and estimated $\nu'_i \in \mathcal{M}(\mathcal{F})$. We use this notation in defining the general form of the quadratic program, with constraint given as described in Section 4.3:

$$\begin{aligned}
 & \inf_{\pi_\ell, \ell \in \mathcal{L}} \|\varphi_1 \cdot \mu_A^\pi - \mu_T\|_{W^*}^2 \\
 & = \inf_{\pi_\ell, \ell \in \mathcal{L}} \sum_{i, i' \in I^2} |D\varphi_1|_{x_i} |D\varphi_1|_{x_{i'}} K_1(\varphi(x_i), \varphi(x_{i'})) \sum_{f \in \mathcal{F}} \nu'_i(f) \nu'_{i'}(f) \\
 & - 2 \sum_{i \in I, j \in J} |D\varphi_1|_{x_i} K_1(\varphi(x_i), x_j) \sum_{f \in \mathcal{F}} \nu_j(f) \nu'_i(f) \tag{6} \\
 & \text{subject to } w^{\min} \leq w'_i \leq w^{\max}, i \in I.
 \end{aligned}$$

...The quadratic program solver used in the context of the alternating scheme is OSQP as implemented in the python `qpsolvers` library.

3 Reviewer 3

The authors have developed a novel computational method aimed at mapping tissue atlases to gene-based and cell-based MERFISH

datasets. Overall, the problem addressed in the manuscript is interesting, and the mathematical foundation of the method appears solid. Yet, the current state of the manuscript is quite disorganized and conveys the feeling to have been written in a rushed way. Therefore, we feel that a major revision is necessary before the manuscript can be reconsidered for publication in Nature Communications. Our recommendations are as follows:

1. **Absence of a method overview:** After introducing the problem that the method aims to solve in the introduction, the authors delve directly into the mathematical equations of the computational tool in Sections 2.1 and 2.2. We recommend providing an overview of the computational tool at the beginning of Section 2, accompanied by a schematic figure, to improve the overall comprehension of the paper for readers.

We thank the reviewer for their suggestion. In addition to combining what were previously Figures 1 and 2, as suggested below, we have included a schematic figure as part of this combined figure summarizing the transformations that are estimated of atlas to target with our method. In particular, we exhibit the independence of each of the transformations in being applied exclusively to physical space (the geometric transformation including affine and non-rigid diffeomorphism) versus exclusively to the feature space of the atlas to bring to transform its regions into distributions over target features. The figure caption for this new figure reads:

Figure Caption: Geometric and functional mappings of two coronal sections of Allen CCFv3 to MERFISH spatial transcriptomic counts of 20 selected genes. 10 μm atlas sections at $Z = 385$ and $Z = 485$ out of 1320 (column 1) visually chosen to match MERFISH architecture (column 5). Both datasets are shown rendered as meshes at $100\mu m$ and $50 \mu m$, respectively, with total transcriptional density depicted as the functional feature for MERFISH slices (column 5). Geometric mappings of atlas sections to target physical space depicted with approximate determinant jacobian showing areas of contraction (blue) and expansion (red) in column 2. Column 3 depicts functional mapping as estimated total transcriptional density per atlas region in native atlas geometry. Column 4 depicts fully transformed atlas following application of geometric and functional feature maps to target physical and feature space. Schematic at bottom depicts source and target objects, defined over different feature spaces (\mathcal{L} , \mathcal{F}). Geometric and functional mappings (gray boxes) applied independently and in parallel to source object, with both estimated in xIV-LDDMM to achieve match between source and target.

Additionally, we have significantly revised the introduction and first Results section (2.1) of the manuscript so that we provide a broader (i.e.

less technical) overview of the method. In particular, we emphasize the representation of an image varifold we use to model both tissue scale atlases and molecular data as crucial to handling the irregularly sampled and high dimensional data found emerging in the setting of most transcriptomics technologies as follows:

To accommodate the high feature dimensionality and spatial irregularity of molecular datasets, as described in the Introduction, we harness the recent work that extends the theory of diffeomorphisms in the setting of images to the setting of image varifolds and estimation of correspondences between them [3, 5]. We first describe this framework of image varifolds to emphasize their capacity for modeling diverse types of tissue-, cellular-, and molecular-scale data. Image varifolds are geometric measures over the product space, $\mathbb{R}^2 \times \mathcal{F}$, therefore encompassing measures over both physical and feature spaces, \mathbb{R}^2 and \mathcal{F} , respectively. At the finest scale of capture in both classical imaging and image-based spatial transcriptomics technologies, we might model a set of pixels or detections (e.g. mRNA reads) as a discrete set of point measures (particles) with an elementary “Dirac” measure, centered over physical location and feature value for each detection. At coarser (e.g. cellular) scales or in spot-based technologies, however, point measurements taken at single physical locations may capture a range of individual detections. To generalize to the range of technologies, we model both tissue-scale atlases and molecular data with semi-discrete image varifolds, defining a collection, indexed by $i \in I$ of point measures over physical space with a sum of “Dirac” measures, $\delta_{x_i}, x_i \in \mathbb{R}^2$, but with corresponding conditional distributions, ν_i , over the associated feature space:

$$\mu \doteq \sum_{i \in I} \delta_{x_i} \otimes \nu_i. \quad (7)$$

The “Dirac” measures over physical space are denoted throughout by δ_x with x a singleton set (e.g. single point) in \mathbb{R}^2 and therefore, $\delta_x(\{y\})$ evaluating to 1 on another singleton set, $\{y\} \in \mathbb{R}^2$ if $x = y$ and 0 otherwise. Feature spaces considered here are finite, comprised of cell types, gene types, or regions in a given atlas ontology. Notably, the conditional feature distribution, ν_i , as defined is not necessarily a probability measure, but can be factored as $\nu_i = w_i \zeta_i$, into a given weight $w_i = \sum_{f \in \mathcal{F}} \nu_i(f)$ and probability distribution $\zeta_i = \frac{1}{w_i} \nu_i$. As such, (7) can be rewritten as $\mu = \sum_{i \in I} w_i \delta_{x_i} \otimes \zeta_i$, giving the marginal distribution $\rho(A) = \mu(A \times \mathcal{F}), A \subset \mathbb{R}^2$ on physical space and consequently a measure of spatial density as mass per unit area : $\alpha = \rho(A)/|A|$. This is governed by the values of the weights, w_i for the semi-discrete form of the image varifold given in (7) with $\rho(A) = \sum_{i \in I : x_i \in A} w_i$. In the setting of detecting individual mRNA independent of cell boundaries, as in the two MERFISH sections shown in Figure 1, this density reflects number of mRNA per

square mm of tissue, while in other settings, it might reflect number of cells per square mm of tissue or be a constant, $\alpha = 1$, across the tissue area, as we assume in the case of atlas images.

We second emphasize both physical transformation and latent feature distribution estimated not just through equations, but also through the reorganization and accompanying schematic shown in Figure 1. We emphasize the rationale for estimating both as integral to defining a similarity metric between deformed atlas and target. The relevant edits to the first Results section are copied below:

Within this established framework, we align image varifolds capturing tissue-scale atlases to those capturing molecular-scale data via estimation of two types of correspondence: one between physical coordinates and the other between feature spaces. These correspondences act independently and in parallel on the physical and feature measure components of the image varifold object, as depicted in the schematic at the bottom of Figure 1, where the gray boxes denote these separate correspondences.

To define a similarity metric between deformed atlas and target, we also need to carry the feature component of tissue-scale atlases to the feature space of the molecular target. For this, we associate to each feature value in the atlas ontology, a distribution over target feature values, capturing the assumption of spatial homogeneity we make within each atlas region. We denote the set of distributions for each atlas feature, $\ell \in \mathcal{L}$ over target feature space as $(\pi_\ell)_{\ell \in \mathcal{L}}$ with each $\pi_\ell \in \mathcal{M}(\mathcal{F})$. Measures over the target feature space are generated for each discrete ‘‘Dirac’’ measure over physical space, indexed by $i \in I$, in the atlas as the mixture distribution:

$$\nu'_i = \sum_{\ell \in \mathcal{L}} \nu_i(\ell) \pi_\ell, \quad (8)$$

with $\nu'_i \in \mathcal{M}(\mathcal{F})$, $i \in I$, the corresponding set of conditional measures over the target feature space, as shown in column 3 of Figure 1. Importantly, we note that this set of distributions is not normalized, capturing both the spatial density of points (e.g. mRNA or cells) in the target as well as the conditional distribution over the feature space (e.g. gene type or cell type).

Finally, the estimation and application of both geometric transformations and latent feature distributions, φ, π , to the atlas image varifold take it to both physical and feature space of the target (see column 4, Figure 1). In the image setting, the similarity function used is often a norm on functions, and solving the problem of minimization of the norm in the space of diffeomorphisms gives the metric theory of LDDMM for generating geodesic matching between exemplar anatomies [1, 2]. Here, the similarity function is a norm on image varifolds (see Section 4.2), capturing proximity in both physical and feature spaces of the target. xIV-LDDMM jointly estimates optimal φ and π to minimize this normed difference in

the space of diffeomorphisms through either simultaneous or alternating optimization algorithms using LDDMM (see Section 4.4), with additional regularization imposed in both settings on the estimated π to ensure, for instance, positive values (see Section 4.3 for explicit variational problems). Note that throughout we highlight mapping examples using both mesh-based [5] and particle-based (point cloud) [3] implementations of image varifolds, with detailed construction, similarities, and differences of each covered in Section 4.1. Supplementary Figure S1 shows equivalence in these two implementations and in the corresponding mappings estimated in each case for the CCFv3 and MERFISH sections shown in Figure 1.

2. **Unclear value of the computational tool: While the motivation behind the work is compelling—aligning different functional modalities at various scales—the authors do not adequately demonstrate the significance of the alignment after applying their method. It would be beneficial to discuss any new biological insights gained and to explain how the method advances our understanding of the data.**

We thank the reviewer for their comment. In response to this comment as well as those of the other reviewers, we have reorganized our results sections to highlight 3 different applications of our method, with explanation of what achieved alignment can be used for. In the third Results section, we showcase the use of our method for novel atlas construction and cross-replicate / cross-sample comparison. We show how a chosen established atlas (e.g. Allen reference atlas) can be used as a coordinate scaffold for integrating molecular measures across samples and technologies and subsequently building an empirical distribution from the collective data to give a finer scale estimate of feature distributions across the tissue sections than that estimated in our optimization scheme where atlas regions are assumed homogeneous over space. Furthermore, we indicate two measures of variance that can be useful in biological settings: the first being a measure of variance across the estimated distributions in an atlas region deformed to each of the targets, indicating how variable the targets are in each region. The second is a measure of spatial variance in the empirical distribution estimated from the composite set of targets across the span of each atlas region, giving an evaluation of how homogeneous the region is across space. The relevant text here is encompassed in the entirety of the newly added Results section:

As described in the Introduction, two key applications of xIV-LDDMM are in cross-sample comparison and atlas construction. We illustrate both of these applications in considering cell-typed MERFISH sections taken from three separate mice at approximately the same coronal level, as shown in Figure 7. CCFv3 section $Z = 890$ out of 1320 was mapped separately

to each of these targets with both geometric transformations and latent feature distributions estimated in each case. As in the classical imaging setting with LDDMM, comparison of the estimated diffeomorphisms taking the CCFv3 section to each respective target section offers one metric of similarity between both atlas and target and across targets [1, 2]. Here, comparison of the jointly estimated latent feature distributions offers a second metric of similarity. The bottom left of Figure 7 highlights the sample variance for each atlas region across the three estimated cell type probability distributions, modeled as unit vectors, $x \in \mathbb{R}^C$, for $C = 33$ cell types. Sample variance is computed as $\frac{1}{N-1} \sum_{n=1}^N \|x_n - \bar{x}\|_2^2$ in \mathbb{R}^C , with $N = 3$ and \bar{x} , the sample mean, as illustrated in tabular form for each atlas region in Supplementary Figure S4. Black dotted arrows highlight the region of the medial geniculate nucleus in each MERFISH sample, which exhibits amongst the highest variance across the replicates. This region is magnified in the right column, illustrating this variance in relative distribution of excitatory neurons, astrocytes, endothelial cells, and ependymal cells.

Regarding atlas construction, the sample mean cell type distribution across those latent distributions estimated jointly with geometric transformations, as shown in Supplementary Figure S4 gives one potential construction of a cell type atlas over space. A finer-grained atlas can alternatively be achieved in this setting by pulling back each target MERFISH section onto the same CCFv3 section via the inverse estimated diffeomorphism. Particles (analogous to foreground pixels) in the CCFv3 section serve as a scaffold for resampling mapped cells to with either a nearest neighbor assignment or dispersion of cell mass according to kernel choice. The center figure in the third row in Figure 7 showcases per particle the cell type with highest probability amongst those assigned to it as their nearest neighbor. Importantly, each particle, here, is treated independently without assuming homogeneity in the cell type distribution across particles belonging to the same atlas region. This is in contrast to the distributions estimated jointly with geometric transformations, and consequently results in more locally varying distributions. Indeed, the third figure in the third row illustrates the variance in cell type distribution across the particles belonging to each specific atlas region, exhibiting where an assumption of homogeneity within these regions may not always be appropriate. The white arrow, in particular, points to an area of cortex where this spatial variance is amongst the highest, with the empirical distribution magnified in the image at the bottom right. Differences in cell types occurring within the outermost portion of the cortex (with predominantly ependymal cells) versus the innermost portion of the cortex (with predominantly excitatory pyramidal cells) are exhibited. Note that the mean cell type distribution across particles is shown in tabular form in Supplementary Figure S4 for comparison to the sample mean estimated across the latent feature laws, π , with very similar structure to them both, supporting the feasibility in jointly estimating geometric transformations and feature laws per atlas

region in xIV-LDDMM.

The accompanying figure also has the following caption:

Cross-replicate comparison of mapping CCFv3 section $Z = 890$ out of 1320 to MERFISH cell-typed coronal sections from three mice at approximately the same location. Top row shows position and cell type for all cells measured in each mouse section. Second row shows CCFv3 section geometrically transformed to each target space with most probable cell type in estimated latent distribution shown for each atlas region. Third row shows initial CCFv3 section with (left) variance in estimated latent probability distributions across replicates, (middle) most likely cell type in empirical distribution estimated from all three replicates pulled back onto the CCFv3 section, (right) spatial variance in empirical probability distribution per each atlas region. Right column shows magnified area of the medial geniculate nucleus in three target MERFISH sections with high variation in cell type distributions between them (top) and magnified area of cortex in CCFv3 section with high spatial variance in cell type distribution from outer ependymal cell dominance to interior excitatory pyramidal neuron dominance.

The second application we highlight is of atlas comparison, both within and across species. Though we showcased the mappings in the 4th Results section in our initial submission, we clarify the use of our method for this purpose in highlighting both direct and indirect ways of comparing atlases (i.e. within the context and independent of molecular datasets) with the edits scattered throughout this section:

Atlas comparison can be conducted both independently and within the context of particular molecular targets as well. The second set of figures in Figure 8 shows the results of mapping nearby corresponding sections of both the ARA and Kim Lab Developmental atlas [42] to the cell-segmented MERFISH section of Figure 3. The images of predicted cell types with the highest probability (left column) for each compartment are shown for each ontology in the left column. The areas of the hippocampus (dashed circle) and striatum and amygdala (arrow) are partitioned with different levels of granularity. This leads to different optimal geometric transformations, as characterized by the determinant Jacobian (middle column), and different predicted cell type distributions (right column). As discussed above, though both atlases are published as geometrically aligned [42], the diffeomorphism solving the variational problem, here, transforms geometrically the homogenous regions between the atlas and target. Hence, regions of the amygdala and striatum undergo significant contraction in the optimal mapping of Kim but not Allen atlas to MERFISH given the partitioning of this region into fewer and thus larger presumed homogenous regions in the Kim atlas. Just as entropy in estimated distribution can be used to

compare atlas to atlas, directly, the right column exhibits the entropy of the distributions over cell types estimated for each region in each atlas, giving an indication to which regions hold more heterogeneous cell type distributions. Here, the hippocampus is more finely partitioned in the Kim atlas, which yields lower entropy distributions over cell types than in those estimated for the Allen atlas.

Finally, we introduce a third application of our method as image (or molecular dataset) segmentation. We show this in tandem with manifesting the success of our method in mapping tissue scale atlas to an additional molecular modality, as suggested by Reviewer 1 of a DAPI-stained image. We underscore, specifically, the use here of pushing an atlas onto a target in contrast to the first application of atlas construction involving the reverse: pulling targets onto a single atlas coordinate space. This is possible through the invertability we impose on estimating strictly a diffeomorphism as our geometric transformations. The relevant text added is contained in the final Results section:

With the computational constraints imposed by modeling spatial transcriptomics datasets as classical images, this work has emphasized the mapping of tissue-scale atlases to molecular datasets generated by emerging image-based rather than the often regularized spot-based spatial transcriptomics technologies. However, as exhibited through the modeling of atlas sections as image varifolds, themselves, both as template and target (see Section 2.5), the xIV-LDDMM is equally capable of aligning tissue-scale atlases to alternative molecular and cellular-scale modalities. As an example, we take a DAPI-stained tissue section (Figure 10), digitized at $2.5\mu\text{m}$ resolution, corresponding to the gene-based MERFISH section shown in Figure 1. The DAPI image (~ 13 million pixels) is converted to an image-varifold particle representation by discretization of its image values into ~ 35 discrete bins, and selection of foreground pixels with corresponding image values in the later 30 of these bins. Approximately 250k particles are used to represent these pixels, with each particle pertaining to a set of 25 neighboring pixels (5×5 square). Particles each carry the distribution of bins into which the image values of these 25 pixels fall, retaining the individual values of each foreground pixel at the highest resolution in contrast to typical image downsampling, in which a pixel only captures the single mean image value of its neighbors.

Figure 10 depicts this particle representation of the DAPI image and its positioning before and after alignment with the corresponding CCFv3 section. Similar to the BARseq mapping illustrated in Figure 6, we see areas of lower cell density (fewer foreground pixels) versus higher cell density (greater and higher intensity foreground pixels) aligning to expected CCFv3 regions (e.g. layer 1 (L1) especially between the hemispheres, and

substantia inominata (SI), respectively). Interestingly, the area of the olfactory tubercle (OT) in the CCFv3 section which is initially quite larger than that in the DAPI image remains large following geometric transformation to the target space, with optimal cost minimization favoring this alignment to one in which this area would drastically contract.

Finally, while the estimated geometric transforms here could equivalently pull back the DAPI image into the space of the CCFv3 section, as done in the setting of atlas construction (see Section 7), we show, instead, the CCFv3 section transformed to the space of the DAPI target as an illustration of how our methodology can be used in settings of atlas segmentation. In particular, as described in the setting of resampling image varifolds across scales [3], we demonstrate the feasibility of translating between particle and image representations in resampling the deformed CCFv3 particles with Gaussian smoothing onto the same $2.5\mu\text{m}$ grid of the original DAPI image (top right image in Figure 10). Note that this smoothing extends the borders of the atlas section slightly beyond their respective positioning as particles, as seen in the difference in overlay laterally within L1 between image and particle representations (top and bottom, respectively).

The accompanying figure caption provided with the example showcasing image segmentation is copied below:

Application of image-varifold based method for mapping CCFv3 section to DAPI-stained image of tissue corresponding to MERFISH section seen in top row of Figure 1. Top left depicts original DAPI-stained image, digitized at $2.5\mu\text{m}$ resolution. Thresholded foreground pixels converted to particle image-varifold representation over a feature space of ~ 30 binned grayscale values. Bottom left depicts this particle representation (black points) overlaying the corresponding CCFv3 section in their respective initial coordinate spaces. Bottom right shows alignment of CCFv3 section to DAPI particles following diffeomorphic transformation to the DAPI coordinate space. Top right shows this same alignment in image format, with the deformed CCFv3 section image generated by resampling the deformed CCFv3 particles onto a regular $2.5\mu\text{m}$ grid. White arrows highlight areas of alignment in the area of the substantia inominata (SI) and layer 1 of cortex whereas red arrows highlight areas of questionable alignment in the area of the olfactory tubercle (OT).

3. **Lack of quantitative validation:** The computational tool has two distinct applications: gene prediction (Figure 3) and cell type distribution prediction (Figure 4). After making these predictions, the authors should compare their method with other existing computational tools and provide a more in-depth analysis of the biological insights derived from the predictions.

We thank the reviewer for their comment. We first aim to clarify that the “predictions” the reviewer references have been emphasized in the context of this work as part of a methodology for bringing atlas and target into geometric alignment rather than serving as standalone “predictors”. We discuss the use of them for alignment purposes in the context of atlas construction in Section 2.4, where we illustrate that a more fine-grained “prediction” can be made of a distribution over target cell types or gene types through the estimation of an empirical distribution based on pulling back multiple targets of the same data type into the atlas space. Each latent feature law estimated in tandem with geometric transformations is based on the data seen in one replicate of data and is constructed from the assumption of homogeneity within each atlas region. As a means of comparing the estimated latent distribution to that given by the construction of an empirical distribution from the multiple targets pulled back onto the single atlas section, we have added Supplementary Figure S4 that shows the mean estimated distribution (top) versus the composite empirically computed one (bottom). The similarity in the two distributions reinforces the validity of the values estimated independently in each case of the latent feature distributions and also indicates the validity in the homogeneity assumption we make to estimate these latent feature distributions. Indeed, we quantify the level to which each atlas compartment truly exhibits homogeneity in distribution across the physical space it covers by computing the variance in empirical distribution at each indexed location within the compartment, with lowest levels in deeper areas of the cortex and highest in the pons, cerebellum, and outer cortex. The following text has been added to the revised manuscript to cover this discussion:

Regarding atlas construction, the sample mean cell type distribution across those latent distributions estimated jointly with geometric transformations, as shown in Supplementary Figure S4 gives one potential construction of a cell type atlas over space. A finer-grained atlas can alternatively be achieved in this setting by pulling back each target MERFISH section onto the same CCFv3 section via the inverse estimated diffeomorphism. Particles (analogous to foreground pixels) in the CCFv3 section serve as a scaffold for resampling mapped cells to with either a nearest neighbor assignment or dispersion of cell mass according to kernel choice. The center figure in the third row in Figure 7 showcases per particle the cell type with highest probability amongst those assigned to it as their nearest neighbor. Importantly, each particle, here, is treated independently without assuming homogeneity in the cell type distribution across particles belonging to the same atlas region. This is in contrast to the distributions estimated jointly with geometric transformations, and consequently results in more locally varying distributions. Indeed, the third figure in the third row illustrates the variance in cell type distribution across the particles belonging to each specific atlas region, exhibiting where an assumption

of homogeneity within these regions may not always be appropriate. The white arrow, in particular, points to an area of cortex where this spatial variance is amongst the highest, with the empirical distribution magnified in the image at the bottom right. Differences in cell types occurring within the outermost portion of the cortex (with predominantly ependymal cells) versus the innermost portion of the cortex (with predominantly excitatory pyramidal cells) are exhibited. Note that the mean cell type distribution across particles is shown in tabular form in Supplementary Figure S4 for comparison to the sample mean estimated across the latent feature laws, π , with very similar structure to them both, supporting the feasibility in jointly estimating geometric transformations and feature laws per atlas region in xIV-LDDMM.

Supplementary Figure Caption:

Mean and variance for estimated (top row) and empirically computed (bottom row) latent cell type distributions for CCFv3 section $Z = 890$ of 1320 mapped to cell-typed MERFISH sections of three separate mice. Variance in estimated cell type distributions computed as sample variance across three replicates (top right). Mean empirical probability distribution computed from combined set of cells from all three mouse sections pulled back onto CCFv3 section with inverse diffeomorphism and assigned to nearest CCFv3 particle (bottom left). Variance in empirical probability distribution over space computed as sample variance in cell type probability distribution for each CCFv3 particle in given region following assignment of pulled back MERFISH cells to nearest neighbor. Arrows indicate corresponding rows for areas of high variance across replicates (medial geniculate nucleus) and high spatial variance (area of cortex) as depicted in Figure 7.

We additionally emphasize that though we aim to estimate feature laws corresponding to those values seen in each target section, the assumption of homogeneity per atlas region in these distributions leads us to estimate what might more appropriately be considered mean distributions without necessarily a meaningful 1:1 comparison with the targets per indexed location. To evaluate the validity in our estimated distributions, we have instead focused on global comparison both of the maximally likely feature value as well as the individual probabilities of each feature value for each atlas compartment and at each indexed location in the target. We have adjusted our Figures 2 and 3 to reflect these two modes of comparison with the associated explanations as well:

Figure 2, for instance, shows the relative expression of three genes, (*Gfap*, *Trp53i11*, *Wipf3*) over space out of a total set of 20 in the target MERFISH section (top) shown in the bottom row of Figure 1. The bottom row of Figure 2 shows the corresponding expression of each of these genes

in each region of the CCFv3 section, as estimated through π (bottom), with notable similarity in probability magnitude and variation over space. The fourth column of Figure 2 showcases another comparative summary of the estimated feature distributions for the CCFv3 section in depicting at each location the most probable gene type. The assumption of spatial homogeneity in distribution is evidenced here, particularly in large areas of the CCFv3, such as the striatum, where each gene carries a single probability of expression across the entire region and *Kirrel3* is uniformly the most probable gene type...

The target and estimated probabilities for two gene types (*Baiap2*, *Slc17a6*) out of a chosen set of six are shown following estimation of an alternative geometric transformation and latent distribution, now, over gene types. Correspondence is seen in both the absolute magnitude as well as relative probability of each gene type between those in the target MERFISH section and those resulting from the estimated feature laws, π . Two examples include the area of high *Baiap2* gene type adjacent to the hippocampus (probability ≈ 0.95) and the area of high *Slc17a6* in the rhomboid nucleus (probability ≈ 0.6). This correspondence serves to reinforce the validity in the estimated feature laws in xIV-LDDMM.

Associated Figure Captions:

Relative expression for three genes (*Gfap*, *Trp53i11*, *Wipf3*) out of a set of twenty (shown at the right), with demonstrated spatial variability according to a computed mutual information score on a section of MERFISH transcriptomics data (top row). Bottom shows estimated expression for same three genes in each region of the CCFv3 section $Z = 485$ out of 1320, as part of the latent distribution over genes estimated in tandem with a geometric transformation to align the CCFv3 section to MERFISH section. Right column depicts gene with highest probability in MERFISH target (top) and as predicted in the latent distribution (bottom) for the CCFv3 section.

Mapping of CCFv3 section $Z = 675$ out of 1320 to single MERFISH cell-based section with either cell type (left) or gene type (right) feature spaces. Top row depicts MERFISH target with varying feature spaces: 33 cell types (left two columns) versus 6 select gene types (right two columns). Bottom row depicts estimated latent feature distribution for each CCFv3 region on original CCFv3 section geometry.

- Inclusion of additional datasets:** The paper focuses solely on mouse brain datasets, which, while understandable given that these datasets are the most readily accessible, limits the scope of the method's applicability. It would be advantageous for the

authors to apply their tool to other datasets to evaluate its effectiveness and generalizability.

We thank the reviewer for their suggestion. We have continued to focus the manuscript on mouse brain datasets for consistency and in light, as the reviewer points out, of the availability of spatial transcriptomic data accessible. However, we have applied our method to datasets from additional technologies including BARseq (both half-brain and whole brain sections) and to additional modalities including DAPI-stained images to demonstrate generalizability. These have been included in the Results sections 2.2 and 2.4 in Figures 4, 6, 10, whose captions are provided for context below:

Figure Caption: Mapping CCFv3 section $Z = 437$ out of 1320 to BARseq cell-typed partial coronal section. Left shows initial positioning of CCFv3 and BARseq sections, with CCFv3 section depicted as black mask. Right shows alignment after pulling back BARseq onto CCFv3 section via inverse estimated diffeomorphism.

Figure Caption: Comparison of mapping CCFv3 section $Z = 837$ out of 1320 to corresponding BARseq cell data using image-based LDDMM [6] (top) versus image varifold-based mapping method (bottom). Source and target images rendered at $50 \mu\text{m}$ for image-based LDDMM, with matching based on foreground/background in Allen atlas section to smoothed cell density in BARseq. Estimated transformations applied to $10 \mu\text{m}$ smoothed cell density image (left) and $5 \mu\text{m}$ cell type image (right) for comparison between mapping methods. Overall alignment shown in left column, with noticeable misalignment of high cell density areas to layer 1 and white matter structures (corpus callosum and dorsal hippocampal commissure) in CCFv3 section with image-based LDDMM. Right column shows these regions magnified, highlighting the layer-by-layer mismatch ($\sim 150 \mu\text{m}$) in the setting of image-based mapping versus the matching layers in CCFv3 partition of cortex to cell types in BARseq following image-varifold based mapping (bottom right).

Figure Caption: Application of image-varifold based method for mapping CCFv3 section to DAPI-stained image of tissue corresponding to MERFISH section seen in top row of Figure 1. Top left depicts original DAPI-stained image, digitized at $2.5 \mu\text{m}$ resolution. Thresholded foreground pixels converted to particle image-varifold representation over a feature space of ~ 30 binned grayscale values. Bottom left depicts this particle representation (black points) overlaying the corresponding CCFv3 section in their respective initial coordinate spaces. Bottom right shows alignment of CCFv3 section to DAPI particles following diffeomorphic transformation to the DAPI coordinate space. Top right shows this same

alignment in image format, with the deformed CCFv3 section image generated by resampling the deformed CCFv3 particles onto a regular $2.5\mu\text{m}$ grid. White arrows highlight areas of alignment in the area of the substantia inominata (SI) and layer 1 of cortex whereas red arrows highlight areas of questionable alignment in the area of the olfactory tubercle (OT).

5. **Unorganized figures: The manuscript includes ten main figures; however, some of them contain insufficient information, such as Figure 1. We recommend combining some of the figures to create a more cohesive presentation of the results.**

We thank the reviewer for their suggestion. We have replaced many of the previous figures with new ones highlighting the quantification of accuracy as well as applications of our method in different settings, in accordance with many of the reviewers' comments. As suggested, we have specifically combined Figures 1 and 2 into a single figure highlighting the output of the method. The first figure caption now reads:

Figure Caption: Geometric and functional mappings of two coronal sections of Allen CCFv3 to MERFISH spatial transcriptomic counts of 20 selected genes. $10\ \mu\text{m}$ atlas sections at $Z = 385$ and $Z = 485$ out of 1320 (column 1) visually chosen to match MERFISH architecture (column 5). Both datasets are shown rendered as meshes at $100\mu\text{m}$ and $50\ \mu\text{m}$, respectively, with total transcriptional density depicted as the functional feature for MERFISH slices (column 5). Geometric mappings of atlas sections to target physical space depicted with approximate determinant of the Jacobian showing areas of contraction (blue) and expansion (red) in column 2. Column 3 depicts functional mapping as estimated total transcriptional density per atlas region in native atlas geometry. Column 4 depicts fully transformed atlas following application of geometric and functional feature maps to target physical and feature space. Schematic at bottom depicts source and target objects, defined over different feature spaces (\mathcal{L} , \mathcal{F}). Geometric and functional mappings (gray boxes) applied independently and in parallel to source object, with both estimated in xIV-LDDMM to achieve match between source and target.

In addition, we have some minor suggestions on the manuscript:

1. **The math in Section 2.1 and 2.2 shall be moved to the Method.**

We thank the reviewer for their suggestion. We have moved the math in what was originally section 2.2 to three different methods sections. The first parts of section 2.2. have been incorporated into the first methods section, which was found in our initial submission, covering the form of mesh representations (versus particle representations) of image varifolds. A particular addition to this section is:

Hence, we denote an image varifold implemented as a mesh similar to the normalized definition of (7) as:

$$\mu_\tau = \sum_{j \in J} w_j (\delta_{m_j} \otimes \zeta_j) = \sum_{j \in J} \delta_{m_j} \otimes \nu_j, \quad (9)$$

with $w_j = \alpha_j |\gamma_j|$ giving the assumed constant density w_j over the area of simplex and $\nu_j = w_j \zeta_j$.

The description of the quadratic programming scheme for estimating latent feature distributions has been moved to the fourth methods section. The bulk of this material is found in the following paragraph from the 4th methods section:

For the atlas, we define the form of the image varifold: $\mu_A = \sum_{i \in I} \delta_{x_i} \otimes \nu_i$, with, $\nu_i \in \mathcal{M}(\mathcal{L})$, a measure over the atlas feature values (partitions). The estimated feature distributions are given via π as the mixture distribution, $\nu'_i = \sum_{\ell \in \mathcal{L}} \nu_i(\ell) \pi_\ell$. We specify the target over the index set, J , as: $\mu_T = \sum_{j \in J} \delta_{x_j} \otimes \nu_j$ with each ν_j and estimated $\nu'_i \in \mathcal{M}(\mathcal{F})$. We use this notation in defining the general form of the quadratic program, with constraint given as described in Section 4.3:

Finally, the definition of the variational problem has been moved to its own methods section (Section 4.3) where we include the basic problem as well as the variations on this problem that can be introduced in different contexts (i.e. priors on the estimated distributions, π). The entirety of this section, as a new addition building off of some of the math in what was previously Section 2.2 is copied below:

As described in Section 2.1, the mapping variational problem constructs a diffeomorphism $\varphi : \mathbb{R}^2 \rightarrow \mathbb{R}^2$ and feature laws $(\pi_\ell)_{\ell \in \mathcal{L}}$ on \mathcal{F} to carry the atlas image varifold onto the target, minimizing the varifold normed difference between them, with norm defined as in Section 4.2. We follow LDDMM as described in the image case, [6], parameterizing the diffeomorphism with the smooth timing varying velocity field, $v_t, t \in [0, 1]$, as $\dot{\varphi}_t = v_t \circ \varphi_t$. This gives the variational problem between atlas image varifold, μ_I and target image varifold, μ_J :

Variational Problem 2

$$\inf_{\substack{v \in L^2([0,1], V), \\ \pi_\ell, \ell \in \mathcal{L}}} \frac{1}{2} \int_0^1 \|v_t\|_V^2 dt + \|\varphi_1 \cdot \mu_I^\pi - \mu_J\|_{W^*}^2 \quad (10)$$

$$\text{with } \dot{\varphi}_t = v_t \circ \varphi_t, \quad \varphi_0 = Id,$$

with μ_i^π depicting the feature transformed atlas image varifold, as described in Section 2.1, as $\sum_{i \in I} \delta_{x_i} \otimes \nu'_i$ with $\nu'_i = \sum_{\ell \in \mathcal{L}} \nu_i \pi_\ell$, the estimated distribution over target features for indexed location i in the atlas image varifold. The space of smooth time-varying velocity fields giving the flow, φ , is defined as a reproducing kernel Hilbert space, equipped with norm, $\|\cdot\|_{\mathcal{V}}^2$, ensuring smoothness and invertibility of φ as a diffeomorphism [40].

We solve (10) for optimal φ, π through either single or alternating algorithms as described in Section 4.4, with the opportunity to impose priors on the estimated distributions, π , appropriate to the specific setting. For instance, we typically impose positivity on all values in estimated distributions: $\pi_\ell(f) \geq 0$ for all $f \in \mathcal{F}$ and $\ell \in \mathcal{L}$. Given prior knowledge on the spatial density of cells or mRNA of a target, we may also specify a range of values for the resulting spatial density to take through constraints on the total measures of features estimated:

$$w^{min} \leq w'_i \leq w^{max}, \quad i \in I \quad (11)$$

with $w'_i = \sum_{f \in \mathcal{F}} \nu'_i$ and ν'_i as in (8). This prior can be easily incorporated as a constraint in the setting of the quadratic program used in the alternating algorithmic approach for estimating optimal distributions π (see Section 4.4. In the examples shown here, we typically take w^{min} to be the 5th percentile of values $w_j, j \in J$ of the target image varifold. Additionally, in the setting of mapping tissue scale atlases to each other, as discussed in Section 2.5, we impose the greater constraint of ensuring constant spatial density of 1, as we use for modeling both atlas and target image varifolds in this case. Finally, in other settings without prior knowledge or wish to impose any on the specific densities prescribed to each indexed location, we can add a general regularization term to (10), such as the KL divergence between the normalized estimated π probability distribution and a uniform distribution across target features:

$$d_i \simeq \bar{\pi}_\ell \log\left(\frac{\bar{\pi}_\ell}{(1/|\mathcal{F}|)}\right), \quad \ell \in \mathcal{L} \quad (12)$$

where $|\mathcal{F}|$ gives the number of discrete feature values in the target feature space and the probability distribution, $\bar{\pi}_\ell = \frac{1}{\sum_{f \in \mathcal{F}} \pi_\ell(f)} \pi_\ell$. We use this approach in the examples with BARseq data shown and in the context of a simultaneous rather than alternating optimization algorithm.

Note that we have retained a few equations in the first Results section (2.1) as a means of explaining our basic model and method. This is also in response to Reviewer 2's last comment requesting the definition of varifold to be done earlier.

2. **It would be helpful to name the computational tool, so that it would be easier for researchers to refer to.**

We thank the reviewer for their suggestion. As we provide varying implementations of our method, we do not give a name specific to each implementation at this point. However, for convenience, we give the name “xIV-LDDMM” which we use throughout the work for describing the method as cross-modality image-varifold LDDMM to emphasize the dual estimation of a diffeomorphism, as in classical IV-LDDMM and the latent feature laws as is necessary in crossing-modalities.

References

- [1] Miller, M.I., Trouvé, A., Younes, L.: Geodesic shooting for computational anatomy. *Journal of Mathematical Imaging and Vision* **24** (2006)
- [2] Miller, M.I., Younes, L., Trouvé, A.: Diffeomorphometry and geodesic positioning systems for human anatomy. *Technology* **02**(01), 36–43 (2014). <https://doi.org/10.1142/s2339547814500010>
- [3] Miller, M., Tward, D., Trouvé, A.: Molecular computational anatomy: Unifying the particle to tissue continuum via measure representations of the brain. *BME Frontiers* (2022)
- [4] Xia, C.-R., Cao, Z.-J., Tu, X.-M., Gao, G.: Spatial-linked alignment tool (slat) for aligning heterogenous slices properly. *bioRxiv*, 2023–04 (2023)
- [5] Miller, M.I., Trouvé, A., Younes, L.: Image Varifolds on Meshes for Mapping Spatial Transcriptomics. *arXiv* (2022). <https://doi.org/10.48550/ARXIV.2208.08376>. <https://arxiv.org/abs/2208.08376>
- [6] Beg, M.F., Miller, M.I., Trouvé, A., Younes, L.: Computing large deformation metric mappings via geodesic flows of diffeomorphisms. *International Journal of Computer Vision* **61**(2), 139–157 (2005). <https://doi.org/10.1023/B:VISI.0000043755.93987.aa>
- [7] Wang, J., Zhang, M.: Geo-sic: learning deformable geometric shapes in deep image classifiers. *Advances in Neural Information Processing Systems* **35**, 27994–28007 (2022)
- [8] Ding, W., Li, L., Zhuang, X., Huang, L.: Cross-modality multi-atlas segmentation using deep neural networks. In: *International Conference on Medical Image Computing and Computer-Assisted Intervention*, pp. 233–242 (2020). Springer
- [9] Jones, A., Townes, F.W., Li, D., Engelhardt, B.E.: Alignment of spatial genomics data using deep gaussian processes. *Nature Methods*, 1–9 (2023)
- [10] Zeira, R., Land, M., Strzalkowski, A., Raphael, B.J.: Alignment and integration of spatial transcriptomics data. *Nature Methods* **19**(5), 567–575 (2022)
- [11] Joshi, S.C., Miller, M.I.: Landmark matching via large deformation diffeomorphisms. *IEEE Transactions on Image Processing* **9**(8), 1357–1370 (2000). <https://doi.org/10.1109/83.855431>

- [12] Biancalani, T., Scalia, G., Buffoni, L., Avasthi, R., Lu, Z., Sanger, A., Tokcan, N., Vanderburg, C.R., Segerstolpe, Å., Zhang, M., *et al.*: Deep learning and alignment of spatially resolved single-cell transcriptomes with tangram. *Nature methods* **18**(11), 1352–1362 (2021)
- [13] Stouffer, K.M., Witter, M.P., Tward, D.J., Miller, M.I.: Projective diffeomorphic mapping of molecular digital pathology with tissue mri. *Communications Engineering* **1**(1), 44 (2022)
- [14] Rodrigues, S.G., Stickels, R.R., Goeva, A., Martin, C.A., Murray, E., Vanderburg, C.R., Welch, J., Chen, L.M., Chen, F., Macosko, E.Z.: Slide-seq: A scalable technology for measuring genome-wide expression at high spatial resolution. *Science* **363**(6434), 1463–1467 (2019)
- [15] Chen, X., Sun, Y.-C., Zhan, H., Kebschull, J.M., Fischer, S., Matho, K., Huang, Z.J., Gillis, J., Zador, A.M.: High-throughput mapping of long-range neuronal projection using in situ sequencing. *Cell* **179**(3), 772–786 (2019)
- [16] Chen, X., Fischer, S., Zhang, A., Gillis, J., Zador, A.: Modular cell type organization of cortical areas revealed by in situ sequencing. *BioRxiv*, 2022–11 (2022)
- [17] Christensen, G.E., Rabbitt, R.D., Miller, M.I.: Deformable templates using large deformation kinematics. *IEEE Transactions on Image Processing* **5**(10), 1435–1447 (1996). <https://doi.org/10.1109/83.536892>
- [18] Grenander, U., Miller, M.I.: Computational Anatomy: An Emerging Discipline. *Applied Mathematics* **56**(4), 617–694 (1998)
- [19] Thompson, P., Toga, A.: A framework for computational anatomy. *Comput Visual Sci* **5**, 13–34 (2002)
- [20] Miller, M.I., Trouné, A., Younes, L.: On the metrics and Euler-Lagrange equations of computational anatomy. *Annual Review of Biomedical Engineering* **4**, 375–405 (2002). <https://doi.org/10.1146/annurev.bioeng.4.092101.125733>
- [21] Avants, B., Gee, J.C.: Geodesic estimation for large deformation anatomical shape averaging and interpolation. *NeuroImage* **23**(SUPPL. 1), 139–150 (2004). <https://doi.org/10.1016/j.neuroimage.2004.07.010>
- [22] Joshi, S., Davis, B., Jomier, M., Gerig, G.: Unbiased diffeomorphic atlas construction for computational anatomy. *NeuroImage* **23**, 151–160 (2004). <https://doi.org/10.1016/j.neuroimage.2004.07.068>. *Mathematics in Brain Imaging*
- [23] Miller, M.I.: Computational anatomy: shape, growth, and atrophy comparison via diffeomorphisms. *NeuroImage* **23**, 19–33 (2004)
- [24] Ashburner, J.: A fast diffeomorphic image registration algorithm. *NeuroImage* **38**(1), 95–113 (2007). <https://doi.org/10.1016/j.neuroimage.2007.07.007>
- [25] Avants, B.B., Epstein, C.L., Grossman, M., Gee, J.C.: Symmetric diffeomorphic image registration with cross-correlation: Evaluating automated labeling of elderly and neurodegenerative brain. *Medical Image Analysis* **12**(1), 26–41 (2008). <https://doi.org/10.1016/j.media.2007.06.004>

- [26] Vercauteren, T., Pennec, X., Perchant, A., Ayache, N.: Diffeomorphic demons: Efficient non-parametric image registration. *NeuroImage* **45**(1, Supplement 1), 61–72 (2009). <https://doi.org/10.1016/j.neuroimage.2008.10.040>. Mathematics in Brain Imaging
- [27] Spherical demons: fast diffeomorphic landmark-free surface registration. *IEEE Trans Med Imaging* **29**(3), 650–68 (2010). <https://doi.org/10.1109/TMI.2009.2030797>
- [28] Ashburner J, R.G.: Symmetric diffeomorphic modeling of longitudinal structural mri. *Front Neurosci* **5**(6) (2013). <https://doi.org/10.3389/fnins.2012.00197>.
- [29] Ding, Z., Niethammer, M.: Aladdin: Joint atlas building and diffeomorphic registration learning with pairwise alignment. In: Proceedings of the IEEE/CVF Conference on Computer Vision and Pattern Recognition, pp. 20784–20793 (2022)
- [30] Avants, B.B., Epstein, C.L., Grossman, M., Gee, J.C.: Symmetric Diffeomorphic Image Registration with Cross-Correlation: Evaluating Automated Labeling of Elderly and Neurodegenerative Brain
- [31] Heinrich, M.P., et al: MIND: Modality independent neighbourhood descriptor for multi-modal deformable registration. *Medical Image Analysis* **16**(7), 1423–1435 (2012). <https://doi.org/10.1016/j.media.2012.05.008>
- [32] Tward, D., et al: Diffeomorphic Registration With Intensity Transformation and Missing Data: Application to 3D Digital Pathology of Alzheimer’s Disease. *Frontiers in Neuroscience* **14**(February), 1–18 (2020). <https://doi.org/10.3389/fnins.2020.00052>
- [33] Iglesias, J.E., et al: Joint registration and synthesis using a probabilistic model for alignment of mri and histological sections. *Medical Image Analysis* **50**, 127–144 (2018)
- [34] Yang, Q., et al: Mri cross-modality image-to-image translation. *Scientific Reports* **10**(1), 1–18 (2020)
- [35] Islam, K.T., Wijewickrema, S., O’Leary, S.: A deep learning based framework for the registration of three dimensional multi-modal medical images of the head. *Scientific Reports* **11**(1), 1–13 (2021)
- [36] Tian, L., Chen, F., Macosko, E.Z.: The expanding vistas of spatial transcriptomics. *Nature Biotechnology* (2022). <https://doi.org/10.1038/s41587-022-01448-2>
- [37] Vahid, M.R., Brown, E.L., Steen, C.B., Zhang, W., Jeon, H.S., Kang, M., Gentles, A.J., Newman, A.M.: High-resolution alignment of single-cell and spatial transcriptomes with cytospace. *Nature Biotechnology*, 1–6 (2023)
- [38] Stringer, C., Wang, T., Michaelos, M., Pachitariu, M.: Cellpose: a generalist algorithm for cellular segmentation. *Nature methods* **18**(1), 100–106 (2021)
- [39] Yao, Z., van Velthoven, C.T., Nguyen, T.N., Goldy, J., Sedeno-Cortes, A.E., Baftizadeh, F., Bertagnolli, D., Casper, T., Chiang, M., Crichton, K., et al.: A taxonomy of transcriptomic cell types across the isocortex

- and hippocampal formation. *Cell* **184**(12), 3222–3241 (2021)
- [40] Dupuis, P., Grenander, U., Miller, M.I.: Variational problems on flows of diffeomorphisms for image matching. *Quarterly of Applied Mathematics* **56**(3), 587–600 (1998)
- [41] Stellato, B., Banjac, G., Goulart, P., Bemporad, A., Boyd, S.: OSQP: an operator splitting solver for quadratic programs. *Mathematical Programming Computation* **12**(4), 637–672 (2020). <https://doi.org/10.1007/s12532-020-00179-2>
- [42] Kronman, F.A., Liwang, J.K., Betty, R., Vanselow, D.J., Wu, Y.-T., Tustison, N.J., Bhandiwad, A., Manjila, S.B., Minter, J.A., Shin, D., et al.: Developmental mouse brain common coordinate framework. *bioRxiv*, 2023–09 (2023)

REVIEWER COMMENTS

Reviewer #1 (Remarks to the Author):

I appreciate the efforts made in addressing the feedback and improving the manuscript from its initial submission. While the changes are extensive but primarily organizational, the critical aspects that require further attention to fully evaluate the method remained. Below are my thoughts on the revisions made in response to the eight key comments:

1. Narrative Style Revision:

o The Introduction and Results have been simplified for a broader audience, and technical details have been moved to the Methods section. However, the manuscript still reads somewhat densely and lacks a clear, easy-to-understand schematic overview of LDDMM. The figures, although improved, seem somewhat rushed and use terms like "gene types" that aren't immediately clear.

2. Comparison to Other Methods:

o Mention of GPSA and SLAT in the introduction and the highlighting of the novel approach for irregular data and different modalities are noted. Regrettably, a clear understanding of how LDDMM compares to methods such as (Geo-SIC, Ding et. al., , 2022., Ding and Niethammer, Aladdin: Joint Atlas Building, 2022 and CAST Tang 2023) is still missing. The paper seems to rely heavily on theoretical discussions rather than experimental benchmarking, making it more suited for a specialized mathematical or computational journal rather than an interdisciplinary one like Nature Communications.

3. Clarity on Usage and Implementation:

o The elaboration on method applications and clarification of implementation details are well-received. however more concrete examples demonstrating real biological insights from the combination of different modalities would had been beneficial, such as how LDDMM can integrate datasets to enhance cell type characterization or distinguish molecular characteristics between disease and normal conditions.

4. Real-world Data Demonstrations:

o The addition of examples using diverse datasets is acknowledged. However, the example of mapping a whole brain atlas to half a brain in spatial transcriptomics seems somewhat simplistic and not entirely realistic.

5. Improving Figures and Interpretability:

o The inclusion of more comprehensive examples and comparative analyses is a positive development.
o Yet, the manuscript still lacks an easy-to-understand schematic overview, and the figures come across more as screenshots than as part of a quantitative analysis.

6. Versatility and Adaptability:

o The expanded use of different brain tissue datasets and discussion on the generalizability of the method are appreciated.
o However, no examples outside of brain tissues are provided, which limits the demonstration of the method's adaptability.

7. Computational Requirements Discussion:

o This section has been well addressed, with detailed computational requirements and links to repositories provided.

8. Cross-Modality Mapping:

o The extension to include DAPI-stained images and digital pathology is commendable.
o However, the manuscript would benefit from providing novel insights based on this integration, rather than just demonstrating image matching.

Overall, the manuscript relies heavily on its mathematical foundations and lacks comprehensive biological comparisons or examples of how it would enrich analyses. Given that many readers of Nature Communications may not have the expertise to thoroughly evaluate these mathematical foundations, the manuscript might be more suited to a more specialized journal.

Reviewer #2 (Remarks to the Author):

Thanks for the authors' efforts responding to the questions and concerns. This reviewer has no further comments on the revised manuscript.

Reviewer #3 (Remarks to the Author):

In the modified version of the manuscript titled 'A 2D Cross-Modality Mapping Method using Image Varifolds to Align Tissue-Scale Atlases to Molecular-Scale Measures', the author has made substantial amendments by incorporating a broader spectrum of data analysis (including new datasets such as BARseq and a more comprehensive quantitative analysis), and provided a more lucid explanation of the computational tool xIV-LDDMM. I commend the author's dedication to enhancing the quality of the manuscript. I am of the opinion that the technical part of work is sufficiently innovative and rigorous to warrant publication in Nature Communication. However, I believe there are still a few queries, previously posed by the reviewers, that need to be resolved before it can be deemed ready for publication:

1. While the author has put efforts into restructuring the manuscript, the narrative style remains overly technical. This could potentially impede comprehension for the broad readership of Nature Communications. One example is Section 2.1, which is densely populated with mathematical details. It would be advisable to relocate some of these extensive technical details to the supplementary information. As a general guideline, the main body of the text should be kept succinct and easily comprehensible.

2. Presentation of figures: The figures need to be arranged in a cleaner way:

(a) It might be better to label the panels by (a) (b) (c) instead of trying to refer to 'ith row/jth column of Figure XXX'

(b) In certain figures, such as Figure 1, the labels on the scale bars and the spatial axis are quite difficult to read. I would recommend refining these figures to ensure that only the pertinent information is included, thereby enhancing their clarity and readability.

(c) The schematic diagram in Figure 1 does not give a very elaborative view of the tools. Rather than relying heavily on textual descriptions, incorporating illustrative drawings/diagrams could greatly aid readers in grasping the core elements of the computational tool more effectively.

(d) Figure 5 needs to be more concise. The quantitative result (2nd and 4th rows of the figures) can be summarized into one panel with the main message getting highlighted. The figures with detailed information could be in supplement information.

3. The reviewers' previous suggestions emphasized the need for added quantitative analysis within the manuscript. This revision has addressed some of these concerns, as evidenced by additions of Figure 5. However, the manuscript would benefit from more and further quantitative analysis. This does not necessarily mean incorporating entirely new analyses. For instance, in the third row of Figure 7, there is an opportunity for the author to delve deeper into the details of the figure, explaining the variance differences in a more quantitative way as well as their biological implications.

4. Unclear value of the computational tool in biology. In addressing the Comment 2 from Review 3 about the value, the author has highlighted three main applications of the method. However, it will be great if the author can elaborate why these questions can be important and what new biological insights have been obtained from the analysis.

5. Some minor questions about the right part of Figure 3:

(a) What is the difference between gene and gene type? Do they refer to the same thing?

(b) Why are six gene types shown in the legend while only two gene types are plotted? What is special about these six gene types?

1 Reviewer 1

Overall, the manuscript relies heavily on its mathematical foundations and lacks comprehensive biological comparisons or examples of how it would enrich analyses. Given that many readers of Nature Communications may not have the expertise to thoroughly evaluate these mathematical foundations, the manuscript might be more suited to a more specialized journal.

We thank the reviewer for their comment. In accordance with the reviewers' comments (reviewer 1 and 3), we have made the manuscript more accessible to a wider audience as fitting for Nature Communications:

1. Addition of an extensive schematic diagram highlighting the optimization scheme and application of estimated transformations (Figure 1).
2. Change of notation to describe weight and probability as w_i, p_i .
3. Removal of measure theory details from Section 2.1 and replacement with:
 - MERFISH gene sections: w_i is total mRNA at location x_i , and p_i is the probability distribution on gene (~ 700).
 - BARseq cell sections: w_i is total cells at location x_i and p_i is the probability distribution on cell type (~ 30).
 - CCFv3 tissue sections: $w_i = 1$ for location x_i in foreground tissue and p_i is the probability distribution on ontology label (~ 700).
4. Clarification between "LDDMM" used for simple tissue-scale contrasts and the image varifold LDDMM for incorporating gene and cell type feature distributions (Introduction).
5. Addition of a new result (Figure 6 (J,K)) showing the reduction of variance in mixed effects models achieved with our method, which facilitates statistical testing of biological hypotheses such as differences in cell type distribution between control and binocular enucleated mice [1].

Please see our respective responses to each of the other comments for details on these changes.

I appreciate the efforts made in addressing the feedback and improving the manuscript from its initial submission. While the changes are extensive but primarily organizational, the critical aspects that require further attention to fully evaluate the method remained. Below are my thoughts on the revisions made in response to the eight key comments:

1. **Narrative Style Revision:** The Introduction and Results have been simplified for a broader audience, and technical details have been moved to the Methods section. However, the manuscript still reads somewhat densely and lacks a clear, easy-to-understand schematic overview of LDDMM. The figures, although improved, seem somewhat rushed and use terms like "gene types" that aren't immediately clear.

We thank the reviewer for their comment. Regarding delineation of LDDMM from the method we propose here (xIV-LDDMM), please see our response to your comment below.

Regarding the schematic overview, we have provided a detailed schematic demonstrating the optimization scheme and application of forward and inverse transformations.

Regarding figures, we have reformatted the figures to increase readability of scale bars and axes. We have replaced “gene type” with gene throughout the manuscript to refer to the “type” of mRNA read captured (e.g. to which gene it corresponds).

The revisions in Figure 1 caption including the description of the schematic are copied below:

Methodology and results of xIV-LDDMM for transforming Allen CCFv3 sections...Top panel shows iterative optimization scheme to estimate geometric transformation (φ) and latent feature distribution (π) by minimizing the normed difference (error) between geometric and feature transformed CCFv3 section to target MERFISH. Middle panel illustrates left to right the application of estimated geometric transformation (φ) to deform CCFv3 atlas to MERFISH coordinates; application of latent feature distribution (π) to generate gene distributions on initial CCFv3 geometry; and application of inverse geometric transformation (φ^{-1}) to deform MERFISH genes to CCFv3 coordinates. Gene with highest probability of expression at each location is shown as target MERFISH feature....Geometric mappings (φ) of CCFv3 sections to MERFISH coordinates...Estimated mRNA density per atlas region (w'_i in (2)), as given by π is shown in CCFv3 coordinates (C,H) and coupled to deformation to MERFISH coordinates (D,I).

2. **Comparison to Other Methods: Mention of GPSA and SLAT in the introduction and the highlighting of the novel approach for irregular data and different modalities are noted. Regrettably, a clear understanding of how LDDMM compares to methods such as (Geo-SIC, Ding et. al., , 2022., Ding and Niethammer, Aladdin: Joint Atlas Building, 2022 and CAST Tang 2023) is still missing. The paper seems to rely heavily on theoretical discussions rather than experimental benchmarking, making it more suited for a specialized mathematical or computational journal rather than an interdisciplinary one like Nature Communications.**

We thank the reviewer for their comment comparing LDDMM to CAST, Geo-SIC, and Aladdin. It is true that all of these methods use diffeomorphisms, and we have added all of these to the manuscript as references. Our group introduced diffeomorphisms in 2000, and they have

become the standard in the field. The significant difference we emphasize, thanks to your comment, between classical LDDMM and the method described here, xIV-LDDMM, is the mapping between classical tissue-scale image types (e.g. atlases) to near infinite dimensional features (e.g. genes, cell types), in which the feature is a latent variable, unknown, and to be estimated simultaneously with the geometric transformation. To date, the multi-modality version of LDDMM as pioneered by us [2] has been limited to relatively small number of contrasts as encountered in various image types (e.g. DTI imaging). The method described here, xIV-LDDMM, demonstrates how to introduce distance norms on the space of gene and cell types while estimating them simultaneously with diffeomorphisms for mapping across scales and modalities.

We have specifically added/changed the following in the introduction to clarify classical LDDMM and its relation to our method as well as the methods cited above.

The smoothness, invertibility, and non-rigid nature of diffeomorphisms are particularly relevant in medical image alignment as they reflect the mechanics/dynamics of soft tissue [3]. Consequently, methods rooted in diffeomorphic registration come particularly ...Large Deformation Diffeomorphic Metric Mapping (LDDMM) [4], on which many such methods are built, specifically equips diffeomorphisms with a metric that allows them to be reduced to a low-dimensional representation that conveniently identifies shape. Consequently, it has been harnessed in the setting of image classification and the study of atrophy in diseased versus control populations [4, 5, 6]. While successful at aligning images of the same modality, most of these diffeomorphism-based methods focus exclusively on generating geometric transformations (e.g. diffeomorphisms). ...in a classical imaging setting through coupling such methods to variations in matching cost or additional transformations....

...CAST [7], a deep graph-based neural network (GNN) algorithm which learns a graph representation of single cell resolution omics data and subsequently aims to align datasets via these graph representations. ...

...as in image-based LDDMM [4], but where the action of diffeomorphisms on images has been adapted to the setting, here, of image-varifolds in a consistent manner [8]....We refer to this method of jointly estimating diffeomorphisms and latent feature distributions, as “cross image-varifold LDDMM” (xIV-LDDMM), where the cross emphasizes its ability to map across scales and modalities [9].

3. Clarity on Usage and Implementation: The elaboration on method applications and clarification of implementation details

are well-received. however more concrete examples demonstrating real biological insights from the combination of different modalities would had been beneficial, such as how LDDMM can integrate datasets to enhance cell type characterization or distinguish molecular characteristics between disease and normal conditions.

We thank the reviewer for their comment and appreciate the probes for clarification on how xIV-LDDMM can aid in distinguishing characteristics under different conditions.

We have added to our analysis of variance in MERFISH replicates (section 2.4) a new result showing the reduction in variance in feature distribution (e.g. gene distribution or cell type distribution) over space within CCFv3 regions that occurs with diffeomorphic transformation. We have described this in the context of the variances in the mixed effects models used for group testing. The additions read:

The spatial variance in cell type distribution within each atlas region and the variance in distribution between replicates of a given population both influence the statistical power needed to detect biologically-relevant differences in these distributions between populations. Classically, they are accounted for in mixed effects models [10] aimed at detecting differences in group features. For instance, we have looked at differences in atrophy rate of medial temporal lobe structures between control and diseased cohorts in Alzheimer’s disease [11, 6]. Here, we observe greater spatial variance across ventral regions than dorsal regions (Figure 6 (I)), with particularly minimal variance in layers of cortex. Importantly, the use of non-rigid geometric transformations (diffeomorphisms) to pull back targets into CCFv3 coordinates reduces this spatial variance in distribution across cortical layers (Figure 6 (J-K)) compared with rigid and scaling transformations only in each of the three mice. Consequently, in settings of comparing groups of replicates under different experimental conditions, we would expect this reduction in variance to facilitate detection of significant differences in cell type distribution per CCFv3 region between them.

We have expanded on the biological insights gleaned through our method with references in the discussion to current/future work looking specifically at groups of replicates under different experimental conditions:

Additionally, the specific estimation of diffeomorphic mappings compared with rigid+scale transformations decreases the spatial variance in feature distributions across regions, increasing the statistical power for detecting differences in feature distributions (e.g. gene expression, cell composition) not just across replicates within a single group, but between groups of replicates under different experimental conditions. For instance,

we are currently looking at differences in cell type and gene distribution following neonatal binocular enucleation as measured with BARseq in four case versus control hemispheres [1].

We have highlighted the different angles of analysis that result from our method with regard to comparison of both geometric and feature distributions across replicates / groups. This is provided in the added supplementary figure and described in Section 2.4:

...(Supplementary Figure S5 (A-C)). Here, the CCFv3 section contracts to a similar extent (~ 0.5) in all three mice in areas of the midbrain around the periaqueductal gray matter. In contrast, levels of contraction/expansion across the cortical layers in the primary visual area range from 0.7 – 1.4 across the three mice, indicating differences in geometric shape. In xIV-LDDMM,... (Supplementary Figure S5 (D-G)). Differences in the predicted cell type with highest probability occur, for instance, in the perirhinal and entorhinal areas with excitatory granule cells, excitatory pyramidal neurons, or cortical excitatory neurons predominant in the different mice (Figure 6 (D-F))....measures difference not just in the single cell type with highest probability, but amongst the entire estimated distributions over cell type for each mouse. ...Notably, the area of the medial geniculate nucleus, while exhibiting excitatory neurons as the predominant cell type across mice,

Supplementary figure caption reads:

Comparison of geometry and cell type distribution in 3 MERFISH replicates via estimated geometric transformations (φ) and cell type distributions (π) estimated with xIV-LDDMM. A-C show determinant of the jacobian for estimated diffeomorphism taking CCFv3 section to individual MERFISH coordinates. Green arrows show similarities in geometry across replicates via equal amounts of contraction; red arrows indicate differences in geometry via differing levels of contraction/expansion. D-G highlight similarities and differences in estimated cell type distribution across replicates. D shows mean probability distribution over cell types for each CCFv3 region, computed from those estimated in xIV-LDDMM for each replicate. Highest probability cell type in mean distribution per CCFv3 region shown in E. F shows variance in estimated probability per cell type for each CCFv3 across the 3 distributions estimated via xIV-LDDMM in mapping the CCFv3 section to the 3 replicates. G shows the total variance across estimated cell type probabilities between replicates.

- 4. Real-world Data Demonstrations: The addition of examples using diverse datasets is acknowledged. However, the example of mapping a whole brain atlas to half a brain in spatial transcriptomics seems somewhat simplistic and not entirely realistic.**

We thank the reviewer for their comments. We are currently working extensively on this problem as a central issue that we have discovered is that partial captures of section, coronal or non-coronal, across different technologies is commonplace (e.g. BARseq, Slide-seq). We have several collaborators on the East and West Coasts who have collected censored versions of complete 3D hemispheres as well. Subselecting the plane in a 3D representation, and/or subselecting a multiple gyrus circuit is a problem of tremendous interest. In our own human work, we have to solve the problem in 3D digital pathology of measuring a significant part of the Medial Temporal lobe and matching it to a labelled complete brain specimen.

This presents a formidable challenge for the community, and there are currently no complete solutions. In the figure shown, while we may approximate these captures as hemi-sections, typically the span of tissue captured ranges between 40% to 75% of a full coronal section. To map the entirety of a partial capture to a tissue scale atlas, we have thus elected to use full coronal sections of the atlas in order to maximize the area of overlap, utilizing the full span of tissue captured, and to allow the algorithms to subselect based on the censored target data the region of the whole section that fits optimally.

Our added variability analysis across replicates of MERFISH (Section 2.4) requires us to move the half-brain mapping figure to the Supplementary Figure section. We have added to our discussion of this setting an indication of particular applications as rationale for where partial capture alignment might be used:

...coupled with their use in studying particular subregions or brain circuits of interest has caused...

- 5. Improving Figures and Interpretability:** The inclusion of more comprehensive examples and comparative analyses is a positive development. Yet, the manuscript still lacks an easy-to-understand schematic overview, and the figures come across more as screenshots than as part of a quantitative analysis.

We thank the reviewer for their comment. Please see our responses to reviewer 3's series of subcomments in their comment 2 regarding our reformatting of figures and inclusion of an expanded schematic overview documenting both the optimization scheme and downstream use of estimated transformations in different applications.

- 6. Versatility and Adaptability:** The expanded use of different brain tissue datasets and discussion on the generalizability of the method are appreciated. However, no examples outside of brain tissues are provided, which limits the demonstration of the method's adaptability.

While we agree with the reviewer that application of the method to align datasets in organs other than the brain would be an exciting area to explore, we have to date remained focused on making a contribution to the transcriptomic community in the brain. To confirm our focus on brain tissues, we have added this clarification of scope to the title with the following revision:

Cross-Modality Mapping using Image Varifolds to Align Tissue-Scale Atlases to Molecular-Scale Measures: Application to 2D Brain Sections

We have kept the suggestion of application to other tissue types in our discussion, as previously.

- 7. Computational Requirements Discussion: This section has been well addressed, with detailed computational requirements and links to repositories provided.**

We thank the reviewer for their comment.

- 8. Cross-Modality Mapping: The extension to include DAPI-stained images and digital pathology is commendable. However, the manuscript would benefit from providing novel insights based on this integration, rather than just demonstrating image matching.**

We agree with the reviewer that novel insights could be provided by the mapping of digital pathology to traditional images and to spatial transcriptomics data. Indeed, we are currently collecting spatial transcriptomics and immuno-stained tissue sections to use this method in the future to look at the correlation between tau tangle pathology and gene expression in Alzheimer's disease. Throughout the manuscript, as our focus has been on highlighting the potential uses for our method through demonstration of its efficacy in the setting of different datasets/types, we have worked with datasets of a small sample size (e.g. 1 - 3) and all produced under control conditions, thus precluding the type of significant detection of differences we aim to use our method for in the future.

To address this comment, we have suggested the type of comparison and insight we might glean from using xIV-LDDMM by comparing the gene expression and DAPI intensity across CCFv3 regions by comparing the single coronal section we illustrate here with the addition of a supplementary figure to indicate the mean DAPI signal per region, analogous, to the observed mean probability distribution over genes.

...This facilitates the extraction of measures in a particular subset of regions, that can be directly compared to other types of measures (e.g. gene expression, cell type distributions) as detected by other types of technologies and equivalently localized to these same CCFv3 regions through mappings computed with xIV-LDDMM, as demonstrated here in Sections 2.2, 2.3, and 2.4. For instance, comparison of mean DAPI signal (Supplementary Figure S7) to maximally expressed gene per CCFv3 region, as shown in Figure 1, illustrates positive correlation between DAPI intensity

and *Whrn* expression, particularly in areas bordering the corpus callosum versus negative correlation with *Mdga1* expression in outer cortical layers.

The added supplementary figure caption reads:

Mean normalized DAPI intensity per CCFv3 region following inverse transformation of DAPI image to CCFv3 coordinates. Foreground pixels only selected by thresholding and mean intensity computed per CCFv3 region based on foreground pixels aligning to within each region.

We have also underscored the relevance of using xIV-LDDMM in the context of disease study for gleaning significant correlations between gene expression and accumulation of pathology in our discussion:

...Importantly, this spatial integration will enable correlation of disease signatures at different scales, as measured through different technologies. For instance, we aim to correlate tau tangle and amyloid-beta pathology in Alzheimer's disease to gene expression and imaging biomarkers, as vital for furthering our mechanistic understanding of the disease and developing early diagnostic strategies.

2 Reviewer 2

Thanks for the authors' efforts responding to the questions and concerns. This reviewer has no further comments on the revised manuscript.

3 Reviewer 3

In the modified version of the manuscript titled 'A 2D Cross-Modality Mapping Method using Image Varifolds to Align Tissue-Scale Atlases to Molecular-Scale Measures', the author has made substantial amendments by incorporating a broader spectrum of data analysis (including new datasets such as BARseq and a more comprehensive quantitative analysis), and provided a more lucid explanation of the computational tool xIV-LDDMM. I commend the author's dedication to enhancing the quality of the manuscript. I am of the opinion that the technical part of work is sufficiently innovative and rigorous to warrant publication in Nature Communication. However, I believe there are still a few queries, previously posed by the reviewers, that need to be resolved before it can be deemed ready for publication:

1. While the author has put efforts into restructuring the manuscript, the narrative style remains overly technical. This could potentially impede comprehension for the broad readership of Nature Communications. One example is Section 2.1, which is densely populated with mathematical details. It would

be advisable to relocate some of these extensive technical details to the supplementary information. As a general guideline, the main body of the text should be kept succinct and easily comprehensible.

We thank the reviewer for their comment. First, we have changed the notation throughout the manuscript to emphasize the different components of the representation capturing the physical properties of the tissue, with w representing weight/mass, and the probability distribution over features, denoted p .

Second, we have eliminated mathematical measure theory details in section 2.1. Both changes read:

...where the physical arrangement of measurements is captured by a collection of discrete point (“Dirac”) measures, but where each such point measure is associated to a full distribution over feature values measured in its neighborhood. This collection of point measures is indexed by $i \in I$, with discrete measures, δ_{x_i} , evaluating to 1 at locations $x_i \in \mathbb{R}^2$, and distributions over feature values modeled as weighted probability distributions: $w_i p_i$. Weight, w_i , is representative of total mass (e.g. total mRNA reads or total cells) measured at location x_i , thus enabling an estimate of density over physical space, and probability distribution, p_i , captures the proportions of each feature type within that mass.

The measure of the complete collection is denoted by the sum:

$$\mu \doteq \sum_{i \in I} w_i \delta_{x_i} \otimes p_i. \quad (1)$$

...

- MERFISH gene sections: w_i is total mRNA at location x_i , and p_i is the probability distribution on gene (~ 700).
- BARseq cell sections: w_i is total cells at location x_i and p_i is the probability distribution on cell type (~ 30).
- Allen CCFv3 tissue sections: $w_i = 1$ for location x_i in foreground tissue and p_i is the probability distribution on ontology label (~ 700).

...Importantly, these distributions are not normalized, enabling generation of both a measure of target mass (e.g. mRNA or cells) over physical space (given by w'_i) and conditional probability distribution (p'_i) over target features (e.g. gene or cell type). Both molecular mass and conditional probability distribution are associated to each point measure in the atlas

through the mixture distribution:

$$w'_i p'_i = w_i \sum_{\ell \in \mathcal{L}} p_i(\ell) \pi_\ell, \quad (2)$$

with w_i, p_i denoting the initial weight and probability distribution of the point measure over the atlas feature space and with

$$w'_i p'_i(f) = \begin{cases} \text{mass of cell type } f \text{ at location } x_i \text{ for } f \in \mathcal{F} = \{\text{cell types}\} \\ \text{mass of gene } f \text{ at location } x_i \text{ for } f \in \mathcal{F} = \{\text{genes}\}. \end{cases}$$

The bottom panel in Figure 1 (C,H) exhibits the estimated mRNA density (w'_i) for each location in the corresponding CCFv3 section while the middle column of the middle panel summarizes the estimated probability distribution over genes (p'_i) with the gene with highest probability denoted for each location.

2. Presentation of figures: The figures need to be arranged in a cleaner way:

- (a) **It might be better to label the panels by (a) (b) (c) instead of trying to refer to ‘ith row/jth column of Figure XXX’**

We thank the reviewer for their suggestion. We have updated all of our figures to label the different components, where appropriate, with letters and refer to them by such letters in the text and captions.

- (b) **In certain figures, such as Figure 1, the labels on the scale bars and the spatial axis are quite difficult to read. I would recommend refining these figures to ensure that only the pertinent information is included, thereby enhancing their clarity and readability.**

We thank the reviewer for their suggestion. We have edited Figure 1 with a new schematic. The third panel, which was the original Figure 1 has been edited by (1) replacing axes with a single scale bar, (2) moving colorbars to the bottom of each column to reduce the total number within the figure and make them bigger, and (3) labeling the different panels within this subfigure, according to the reviewer’s first suggestion. We have also removed axes where possible from other figures and replaced them with smaller scale bars to make colorbars larger and more readable (Figures 1,2,3,8).

- (c) **The schematic diagram in Figure 1 does not give a very elaborative view of the tools. Rather than relying heavily on textual descriptions, incorporating illustrative drawings/diagrams could greatly aid readers in grasping the core elements of the computational tool more effectively.**

We thank the reviewer for their comment. In accordance also with reviewer 1’s suggestion, we have updated the schematic in Figure 1,

expanding it to two separate panels. The first showcases the scheme of optimization with the error term between deformed atlas and target updating the parameters governing the geometric and feature transformations. The second panel highlights the use of both forward and inverse transformations. Both panels underscore the central message of two transformations being estimated, related to the geometry of the coordinates versus the gene/cell type distributions.

The new figure caption reads:

Methodology and results of xIV-LDDMM for transforming Allen CCFv3 sections...Top panel shows iterative optimization scheme to estimate geometric transformation (φ) and latent feature distribution (π) by minimizing the normed difference (error) between geometric and feature transformed CCFv3 section to target MERFISH. Middle panel illustrates left to right the application of estimated geometric transformation (φ) to deform CCFv3 atlas to MERFISH coordinates; application of latent feature distribution (π) to generate gene distributions on initial CCFv3 geometry; and application of inverse geometric transformation (φ^{-1}) to deform MERFISH genes to CCFv3 coordinates. Gene with highest probability of expression at each location is shown as target MERFISH feature....Geometric mappings (φ) of CCFv3 sections to MERFISH coordinates...Estimated mRNA density per atlas region (w'_i in (2)), as given by π is shown in CCFv3 coordinates (C,H) and coupled to deformation to MERFISH coordinates (D,I).

We have also described the figure throughout Section 2.1 with the following edits:

Consequently, while they are both applied in the setting of optimization to evaluate alignment of atlas to target (top panel, Figure 1, they can be applied individually as relevant to specific applications including data segmentation, atlas construction, and cross-specimen comparison, as depicted in the middle panel in Figure 1.

Notably, the estimated diffeomorphism can be applied in both the forward and inverse directions, taking tissue-scale atlas to molecular coordinates or vice versa, as shown in the middle panel in Figure 1.

...as necessary to evaluate the similarity function in the optimization scheme (top panel, Figure 1). In the image setting,...

- (d) **Figure 5 needs to be more concise. The quantitative result (2nd and 4th rows of the figures) can be summarized into one panel with the main message getting highlighted. The figures with detailed information could be in supplement information.**

We have updated Figure 5 by reducing the 4 histograms to subsections covering the spread of distances up to 25%. We have summarized the omitted components of the figures in the caption instead. We have also labeled the different components of the figure for readability as suggested by the reviewer's earlier comment. The revisions in the figure caption are:

...Percents of cell markers clipped to 25% in all histograms, with 70%, 50%, 60% cell markers within 10 μm in CA1, CA3, and DG, respectively in F and 95%, 35%, 60% cell markers within 10 μm in CA1, CA3, and DG, respectively in H.

3. **The reviewers' previous suggestions emphasized the need for added quantitative analysis within the manuscript. This revision has addressed some of these concerns, as evidenced by additions of Figure 5. However, the manuscript would benefit from more and further quantitative analysis. This does not necessarily mean incorporating entirely new analyses. For instance, in the third row of Figure 7, there is an opportunity for the author to delve deeper into the details of the figure, explaining the variance differences in a more quantitative way as well as their biological implications.**

We thank the reviewer for their suggestion. We agree with the reviewer that there is further opportunity for quantitative analysis following alignment with our method. For instance, we might consider testing for significant differences in distributions across populations of replicates, where both the spatial variance and variance across replicates within a single population, as highlighted in what was the third row of Figure 7, contribute to our ability to decipher these differences. Here, we note that the small sample size (3) and setting of a single experimental condition (healthy controls) inhibits such further analysis. Indeed, we computed, for instance, a χ^2 test for each region in the CCFv3 atlas, to assess whether there are significant differences between cell type distributions across replicates, and found more than half of the regions to be significantly different in distribution.

To address the implications of these two types of variance, we have added details in our discussion of Figure 6 regarding their effect on detecting differences in populations as captured in classic mixed effects models. We have provided a reference in the discussion specifically to where this is being done with spatial transcriptomics in looking at cell type distributions between control mice and those with binocular enucleation [1]. We have also added a subfigure to the original one highlighting how the use of diffeomorphic transformations in particular compared with rigid+scale only serves to decrease the spatial variance in distribution for each replicate within regions of the cortical layers, which would ultimately increase the power we have, then, to detect differences across groups. These additions are found in Section 2.4 as follows:

The spatial variance in cell type distribution within each atlas region and the variance in distribution between replicates of a given population both influence the statistical power needed to detect biologically-relevant differences in these distributions between populations. Classically, they are accounted for in mixed effects models [10] aimed at detecting differences in group features. For instance, we have looked at differences in atrophy rate of medial temporal lobe structures between control and diseased cohorts in Alzheimer's disease [11, 6]. Here, we observe greater spatial variance across ventral regions than dorsal regions (Figure 6 (I)), with particularly minimal variance in layers of cortex. Importantly, the use of non-rigid geometric transformations (diffeomorphisms) to pull back targets into CCFv3 coordinates reduces this spatial variance in distribution across cortical layers (Figure 6 (J-K)) compared with rigid and scaling transformations only in each of the three mice. Consequently, in settings of comparing groups of replicates under different experimental conditions, we would expect this reduction in variance to facilitate detection of significant differences in cell type distribution per CCFv3 region between them.

Finally, to expand upon the biological nature of the variance highlighted across replicates and over space, we have added indications of which cell types exhibit the most variance across replicates and across space, with designation of the probabilities of each of these cell types. These are included in an added Figure 7 with the accompanying discussion and caption:

Here, the small sample size ($n=3$) of replicates, which are all produced under control conditions, precludes any definitive statistical statement about variations in cell type distributions observed between them and within CCFv3 regions. Nevertheless, summation across all CCFv3 regions of the variance per individual cell type probability between replicates and within each region (Figure 7) elucidates the relative contribution of each cell (sub)type probability to the variances observed in Figure 6 (G,I). We observe, for instance, with both types of variance, a large contribution from the variance in ependymal cell type probabilities (Figure 7 (A,B)). Specifically between replicates, excitatory neurons exhibit the second largest variance in cell type probabilities, as evidenced in the area of the dentate gyrus (C-E). With regard to spatial variance, we see specific astrocyte subtype probabilities with large variance, exhibited particularly in the areas of CA1 (yellow arrow) and the pons (orange arrow) both medially to laterally within the regions and when considering left versus right hemispheres (F-G). In contrast, other astrocyte subtype probabilities (H) are seen to be consistent over space, in line with our assumption of homogeneity within each CCFv3 region.

Figure Caption: Variance in individual cell subtype probabilities across replicates (A) and across space within each CCFv3 region (B), summed across all regions. C-E shows probability of excitatory neuron subtype 2

(star in A) for each of the three mice in CCFv3 coordinates. Yellow arrow highlights area of dentate gyrus with differences in excitatory neuron subtype 2 probabilities. F-H shows probability for astrocyte subtypes 1,2, and 3 (stars in B) in empirical distribution computed from all three mice in CCFv3 coordinates (most likely cell type shown in Figure 6 (H)). Yellow arrow highlights area of CA1 with differences in astrocyte probability medially to laterally in subtypes 1 and 2 but not 3. Orange arrow highlights differences medially to laterally and left and right in areas of pons in astrocyte probability for subtypes 1 and 2 but not 3. Black lines (F-H) indicate boundaries between CCFv3 regions.

4. **Unclear value of the computational tool in biology. In addressing the Comment 2 from Review 3 about the value, the author has highlighted three main applications of the method. However, it will be great if the author can elaborate why these questions can be important and what new biological insights have been obtained from the analysis.**

We thank the reviewer for their comment. In line with our response to the previous comment as well as our responses to reviewer 1's comments 3 and 8, we have included additional insight into what biological questions might be answered through use of our method as follows.

First, we have added references to current and future work in our discussion involving linking datatypes across scales (e.g. imaging biomarkers to distribution of pathology and gene expression) and comparing cell type distributions between control mice and those with binocular enucleation:

Importantly, this spatial integration will enable correlation of disease signatures at different scales, as measured through different technologies. For instance, we aim to correlate tau tangle and amyloid-beta pathology in Alzheimer's disease to gene expression and imaging biomarkers, as vital for furthering our mechanistic understanding of the disease and developing early diagnostic strategies. Additionally, the specific estimation of diffeomorphic mappings compared with rigid+scale transformations decreases the spatial variance in feature distributions across regions, increasing the statistical power for detecting differences in feature distributions (e.g. gene expression, cell composition) not just across replicates within a single group, but between groups of replicates under different experimental conditions. For instance, we are currently looking at differences in cell type and gene distribution following neonatal binocular enucleation as measured with BARseq in four case versus control hemispheres [1].

Second, we have provided more extensive examination of the biological correlations that can be gleaned following alignment in the case of aligning different types of datasets to the same tissue scale atlas (e.g. DAPI signal versus gene-based spatial transcriptomics (Section 2.6)):

...This facilitates the extraction of measures in a particular subset of regions, that can be directly compared to other types of measures (e.g. gene expression, cell type distributions) as detected by other types of technologies and equivalently localized to these same CCFv3 regions through mappings computed with xIV-LDDMM, as demonstrated here in Sections 2.2, 2.3, and 2.4. For instance, comparison of mean DAPI signal (Supplementary Figure S7) to maximally expressed gene per CCFv3 region, as shown in Figure 1, illustrates positive correlation between DAPI intensity and *Whrn* expression, particularly in areas bordering the corpus callosum versus negative correlation with *Mdga1* expression in outer cortical layers.

Third, as described in our response to the reviewer’s prior comment, we have indicated how our reduction in spatial variance via diffeomorphic mapping might provide greater statistical power to decipher significant differences in distributions between groups of replicates (Figure 6), and suggested ways of comparing replicates based on geometric differences, as evidenced by the determinant of the jacobian (Supplementary Figure S5), or feature distribution differences, considering, for instance, the distribution of single feature values (e.g. individual cell subtype probabilities, as in Figure 7) or the variance between replicates’ distributions as a whole (Figure 6).

...(Supplementary Figure S5 (A-C)). Here, the CCFv3 section contracts to a similar extent (~ 0.5) in all three mice in areas of the midbrain around the periaqueductal gray matter. In contrast, levels of contraction/expansion across the cortical layers in the primary visual area range from 0.7 – 1.4 across the three mice, indicating differences in geometric shape.

...Overall, ventral areas exhibit higher spatial variance than dorsal areas (Figure 6 (I)),...The spatial variance in cell type distribution within each atlas region and the variance in distribution between replicates of a given population both influence the statistical power needed to detect biologically-relevant differences in these distributions between populations. Classically, they are accounted for in mixed effects models [10] aimed at detecting differences in group features. For instance, we have looked at differences in atrophy rate of medial temporal lobe structures between control and diseased cohorts in Alzheimer’s disease [11, 6]. Here, we observe greater spatial variance across ventral regions than dorsal regions (Figure 6 (I)), with particularly minimal variance in layers of cortex. Importantly, the use of non-rigid geometric transformations (diffeomorphisms) to pull back targets into CCFv3 coordinates reduces this spatial variance in distribution across cortical layers (Figure 6 (J-K)) compared with rigid and scaling transformations only in each of the three mice. Consequently, in settings of comparing groups of replicates under different experimental conditions, we

would expect this reduction in variance to facilitate detection of significant differences in cell type distribution per CCFv3 region between them.

Here, the small sample size ($n=3$) of replicates, which are all produced under control conditions, precludes any definitive statistical statement about variations in cell type distributions observed between them and within CCFv3 regions. Nevertheless, summation across all CCFv3 regions of the variance per individual cell type probability between replicates and within each region (Figure 7) elucidates the relative contribution of each cell (sub)type probability to the variances observed in Figure 6 (G,I). We observe, for instance, with both types of variance, a large contribution from the variance in ependymal cell type probabilities (Figure 7 (A,B)). Specifically between replicates, excitatory neurons exhibit the second largest variance in cell type probabilities, as evidenced in the area of the dentate gyrus (C-E). With regard to spatial variance, we see specific astrocyte subtype probabilities with large variance, exhibited particularly in the areas of CA1 (yellow arrow) and the pons (orange arrow) both medially to laterally within the regions and when considering left versus right hemispheres (F-G). In contrast, other astrocyte subtype probabilities (H) are seen to be consistent over space, in line with our assumption of homogeneity within each CCFv3 region.

5. Some minor questions about the right part of Figure 3:

- (a) **What is the difference between gene and gene type? Do they refer to the same thing?**

Yes. We have changed the wording in the manuscript from “gene type” to “gene” to reflect the feature of the mRNA as the gene it corresponds to.

- (b) **Why are six gene types shown in the legend while only two gene types are plotted? What is special about these six gene types?**

We thank the reviewer for their comment. We selected six gene types to analyze as spatially varying ones according to Moran’s I score. We chose only to plot the probabilities of two of these for aesthetic purposes to limit the number of total figures while still showcasing the nature of our algorithm’s estimation of probability distributions and not just single maximum values of feature type. However, we have added the probabilities of the other 4 genes for clarity and completeness. We have also updated this figure according to your comments above with panel delimiters and replacement of axes with single scale bars. We have clarified this decision in the text by adding in Section 2.2:

Here, we subsample the entire gene space by selecting a subset of six genes (*Baiap2*, *Slc17a6*, *Adora2a*, *Gpr151*, *Gabbr2*, *Cckar*) as those most spatially varying according to Moran’s I score [12].

The updated Figure caption also reads:

Mapping of CCFv3 section $Z = 675$ out of 1320 to single MERFISH cell-based section with either cell type (A) or gene (B,C) feature spaces. Top row in A,B,C depicts MERFISH target with varying feature spaces: 33 cell types (A) and 6 genes selected from a set of ~ 500 as those with high spatial variance (B,C). Bottom row in A,B,C depicts estimated latent feature distribution for each CCFv3 region on original CCFv3 section geometry. Cell type feature distributions (A) are summarized by cell density and cell type with maximum probability. Gene feature distributions are depicted by probabilities for each gene out of the set of 6 (B,C).

References

- [1] Chen, X., Fischer, S., Rue, M.C., Zhang, A., Mukherjee, D., Kanold, P.O., Gillis, J., Zador, A.M.: Whole-cortex in situ sequencing reveals peripheral input-dependent cell type-defined area identity. *bioRxiv* (2023) <https://arxiv.org/abs/https://www.biorxiv.org/content/early/2023/10/04/2022.11.06.515380.full.pdf>. <https://doi.org/10.1101/2022.11.06.515380>
- [2] Ceritoglu, C., Oishi, K., Mori, S., Miller, M.I.: Multi-contrast large deformation diffeomorphic metric mapping and diffusion tensor image registration. *NeuroImage* **47**(1), 123 (2009)
- [3] Ding, Z., Niethammer, M.: Aladdin: Joint atlas building and diffeomorphic registration learning with pairwise alignment. In: *Proceedings of the IEEE/CVF Conference on Computer Vision and Pattern Recognition*, pp. 20784–20793 (2022)
- [4] Beg, M.F., Miller, M.I., Trouné, A., Younes, L.: Computing large deformation metric mappings via geodesic flows of diffeomorphisms. *International Journal of Computer Vision* **61**(2), 139–157 (2005). <https://doi.org/10.1023/B:VISI.0000043755.93987.aa>
- [5] Wang, J., Zhang, M.: Geo-sic: learning deformable geometric shapes in deep image classifiers. *Advances in Neural Information Processing Systems* **35**, 27994–28007 (2022)
- [6] Kulason, S., et al: Cortical thickness atrophy in the transentorhinal cortex in mild cognitive impairment. *NeuroImage: Clin.* **21**, 101617 (2019). <https://doi.org/10.1016/j.nicl.2018.101617>
- [7] Tang, Z., Luo, S., Zeng, H., Huang, J., Wu, M., Wang, X.: Search and match across spatial omics samples at single-cell resolution. *bioRxiv*, 2023–08 (2023)
- [8] Miller, M., Tward, D., Trouné, A.: Molecular computational anatomy: Unifying the particle to tissue continuum via measure representations of the brain. *BME Frontiers* (2022)
- [9] Miller, M.I., Trouné, A., Younes, L.: Space-feature measures on meshes for mapping spatial transcriptomics. *Medical Image Analysis* **93**, 103068 (2024)

18 REFERENCES

- [10] Nichols, T., Hayasaka, S.: Controlling the familywise error rate in functional neuroimaging: a comparative review. *Statistical methods in medical research* **12**(5), 419–446 (2003)
- [11] Younes, L., Albert, M., Miller, M.I.: Inferring changepoint times of medial temporal lobe morphometric change in preclinical Alzheimer’s disease. *NeuroImage: Clinical* **5**, 178–187 (2014). <https://doi.org/10.1016/j.nicl.2014.04.009>
- [12] Miller, B.F., Bambah-Mukku, D., Dulac, C., Zhuang, X., Fan, J.: Characterizing spatial gene expression heterogeneity in spatially resolved single-cell transcriptomic data with nonuniform cellular densities. *Genome research* **31**(10), 1843–1855 (2021)

REVIEWERS' COMMENTS

Reviewer #1 (Remarks to the Author):

Authors addressed my comments and the manuscript improved significantly. I endorse its publication.

Reviewer #3 (Remarks to the Author):

The author has addressed previously raised issues in this iteration of the manuscript by including further quantitative data analysis and updating the figures. Overall, despite the technical nature of the paper's presentation, it can be considered for acceptance. In my evaluation, the paper has met the standards for publication in Nature Communications.

It is recommended that the legends for Figure 4 (parts E through H) be cleaner for better readability before publication.

Reviewer Responses

Reviewer #1 (Remarks to the Author):

Authors addressed my comments and the manuscript improved significantly. I endorse its publication.

Reviewer #3 (Remarks to the Author):

The author has addressed previously raised issues in this iteration of the manuscript by including further quantitative data analysis and updating the figures. Overall, despite the technical nature of the paper's presentation, it can be considered for acceptance. In my evaluation, the paper has met the standards for publication in Nature Communications.

It is recommended that the legends for Figure 4 (parts E through H) be cleaner for better readability before publication.

We have increased the font size of the legends in each of Figures 4 E-H and reduced the number of decimal points from 4 to 3 for better readability.